

# Comparison between ground-based remote sensing observations and NWP model profiles in complex topography: the Meiringen campaign

Alexandre Bugnard[1], Martine Collaud Coen[1], Maxime Hervo[1], Daniel Leuenberger[1], Marco Arpagaus[1], and Samuel Monhart[1]

[1]Federal Office of Meteorology and Climatology, MeteoSwiss, Switzerland

**Correspondence:** Martine Collaud Coen (martine.collaud@meteoswiss.ch)

**Abstract.** Thermally driven valley winds and near-surface air temperature inversions are common over complex topography and have a significant impact on the mesoscale weather situation. They both affect the dynamics of air masses and pollutant concentration. Valley winds affect it by favoring exchange between the boundary layer and the free troposphere, and temperature inversion by concentrating pollutants in cold stable surface layers. The complex interactions that lead to the observed weather patterns are challenging for Numerical Weather Prediction (NWP) models. To study the performance of the COSMO-1 model anaylsis (KENDA-1), a measurement campaign took place from October 2021 to August 2022 in the 1.5 km wide Swiss Alpine valley called Haslital. A Microwave Radiometer and a Doppler Wind Lidar were installed at Meiringen, in addition to a multitude of automatic ground measurement stations observing meteorologic surface variables. Near the measurement's sites, a low altitude pass, the Brünig Pass, influence the wind dynamic similarly to a tributary. The collected data shows frequent nighttime temperature inversions for all months under study, which persist during daytime in colder months. An extended thermal wind system was also observed during the campaign, except in December and January allowing an extented analysis of along and cross valley winds. The comparison between the observations and the KENDA-1 data provides good model performances for monthly temperature and wind climatologies but frequent and important differences for particular cases, especially in case of foehn events. Modeled nighttime ground temperature overestimations are common due to missed temperature inversions resulting in bias up to 9 °C. Concerning the valley wind system, modeled flows are similar to the observations in their extent and strength, but suffer from a to early morning transition time towards up valley winds. The findings of the present study allow to better understand the temperature distributions, the thermally driven wind system in a medium size valley, the interactions with tributary valley flows, as well as the performances and limitations of a model in such complex topography.

**Keywords.** Complex topography, Remote sensing, NWP, Temperature inversion, Valley winds, Foehn

## 1 Introduction

Over mountainous areas, interactions between the terrain and the overlying atmosphere favor the vertical transport of moisture and pollutants and consequently increase the air masses exchanges between the boundary layer and the free troposphere. Both theoretical studies and experimental campaigns demonstrated that complex topography creates circulation patterns with small





and large space and time pattern. In valleys, the superposition of the various processes leads to a complex vertical layering in the mountainous boundary layer, which is strongly related to the specific conditions ot the surrounding terrain in each studied valley.

For Numerical Weather Prediction (NWP) models, simulation of the atmosphere over complex terrain requires not only dense and accurate horizontal and vertical grids to parametrize the mountainous terrain (Sekula et al., 2019) but also good estimates of vegetation, soil characteristics, net radiation, and speed of the large-scale flow (Adler et al., 2021).

## 1.1 Complex topography specificity and modeling challenges

### 1.1.1 Near surface Temperature inversion

During calm clear nights, the air T in valleys can fall below the T measured across the surrounding hill tops leading to high frequency of cold-air pooling and associated T inversions in mountainous regions (Miró et al., 2018) (Joly and Richard, 2019). T inversions influence fog formation ((Chachere and Pu, 2017)), vertical dilution of pollutants ((Duine et al., 2017; Diémoz et al., 2019)) and the development of the boundary layer during daytime ((Schnitzhofer et al., 2009)). Such inversion are favored in complex topography (Joly and Richard, 2018) and persists longer in deeper valleys, where inversion lifetimes converge to the one over a plain for wide valleys (Colette et al., 2003).

However, the small-scale nature of these near surface stable layers means that they are often poorly represented in even the highest resolution operational NWP models ((Vosper et al., 2013)). The quality predictions for near surface variables during stable conditions depends on locally generated circulations that are controlled by many factors such as turbulence, shortwave and long wave radiation, advection and subsidence. Deficiencies in the parametrization of the fluxes, and especially during stable conditions have already been shown (Hauge, 2006) and thus a finer grid resolution is needed in increasingly steep terrain (Sfyri et al., 2018). Simulations also exhibit high sensitivity to the choice of the vertical grid in the prediction of cold pool formation and suggest that the vertical resolution near the surface is more important than the height of the lowest level (Vosper et al., 2013).

However, the assimilation of measurements, not only of surface data but also of profiling observations ( Crezee et al. (2022)), may improve the NWP performance for surface T inversions (Martinet et al., 2017). (Nipen et al., 2020) even proposes to integrate citizen observations (i.e. measurements from private meteorological stations) into NWP models. Future developments in these techniques may allow addressing the problem of T inversions.

### 1.1.2 Thermally induced valley winds and their interaction with synoptic flows

Thermally driven winds principally occur under fair-weather conditions (Zardi and Whiteman, 2013). They develop due to differential heating of adjacent air masses. They can partially be explained by the topographic amplification factor concept (Whiteman, 1990) and slope-flow-induced local subsidence in the valley center (Schmidli and Rotunno, 2010) leading to a faster heating of the valley than of the plain. The valley–plain T contrast then produces an along-valley pressure gradient that induces strong and deep up-valley wind during the day and more shallow down-valley wind during the night. Slope winds are air mass movements parallel to the slope induced by buoyancy force in the presence of air layers at different T. Slope winds move upward



during the day and downward at night and play an important role in the morning and evening transition. However, slope winds evolve over shorter time scales than valley winds (Serafin et al., 2018).

The transition periods of up and down along valley winds are mostly driven by the sunrise and sunset times. Even though minor changes in the topography can lead to a significant change in the flow regimes (Lang et al., 2015), some common features are observed among the existing studies. In general, the morning transition happens with a certain delay with respect to sunrise

caused by the time required for up slope winds and warm subsidence to erode the nocturnal T inversion. However, wind intensity can be heavily related to tributary valleys (Zängl, 2004) and therefore highly depends on the local topography. As soon as the surface outgoing longwave radiation exceeds incoming shortwave radiation, a layer of cold air forms over all shaded colder surfaces. This air begins to move down the slope and converge in the valley floor, which stops up-valley winds and flips the flow direction. For the evening transition, the wind direction reverses more rapidly at sunset.

Synoptic winds coupled with wind channeling effects can however superpose on the above described thermal mountain winds (Jacques-Coper et al., 2015). This large scale flows present no defined diurnal cycle and are generally stronger than the thermal valley winds. Their effect on the valley wind system is highly variable and depends on the orientation of the synoptic flow with respect to the valley axis (Kossmann and Sturman, 2003). Finally, idealized simulations taking into account the interaction of the purely thermally driven flows with an overlying synoptic flow are rare (Rotach et al., 2015), so that further investigations are

needed.

The capability of mesoscale NWP models to calculate the above described diurnal valley winds in real valleys has been investigated in a few studies (Chow et al., 2006; Langhans et al., 2013; Schmidli et al., 2018; Schmidli and Quimbayo-Duarte, 2023). Globally, a good agreement between modeled and observed valley winds is achieved provided that spatial resolution of the models and surface data (e.g. snow cover and soil moisture) are high enough (Rotach et al., 2015). The size of the valley has

an impact on the accuracy of the modeled winds. Schmidli et al. (2018) showed that the COSMO1-E (resolution 1.1 km) diurnal cycle was well represented in large and medium valleys, but the valley wind speed is underestimated in smaller valleys. The same simulation with lower resolution of 2.2 km shows a larger tendency to diurnal wind speed underestimation. Generally, a closer agreement between the model and measurements was found for the smaller spatial resolution, which allows a better representation of the topography. (Wagner et al., 2014) shows that the grid resolution should be about 10 to 20 times smaller

than the relevant topographic scale to fully capture the different exchange processes. With few exceptions, smaller altitude biases of the model at the ground result in reduced wind speed errors. Hence, increased grid resolution generally improves the performance of numerical simulations, which is even more pronounced if surface and soil model fields are accurately initialized. Precise soil moisture data seems to be an important parameter to obtain good modeling results (Langhans et al., 2013; Schmidli and Quimbayo-Duarte, 2023).

Finally, the performance of models to handle foehn events had been shown to be difficult. Operational analysis data from COSMO-1E exhibit a cold bias over the whole profile bottom part (<1000 m) of valleys (Jansing et al., 2022; Tian et al., 2022). Wind speeds in the model are generally higher, both above crest height and within the valley. Modeling exercises reveal considerable differences in foehn occurrence frequency, which constraints the use of this data for studies at mesocale Saigger and Gohm (2022).





## 1.2 Goals of the paper


Although the surface measurement network is relatively well distributed over the Alps, T and wind profile measurements via remote sensing (REM) instruments are rarely operationally performed in the Alpine valleys. However, a precise knowledge of the T structure of the atmosphere in complex terrain is essential for NWP models and the use of REM observations is a solution to obtain sufficient space/time resolution of the fast varying meteorological conditions in valleys. Moreover few measurement
campaigns provided comparison of modeled and observed profiles of thermally induced winds system (Schmid et al., 2020; Schmidli and Quimbayo-Duarte, 2023; Giovannini et al., 2017; Adler et al., 2021) in the Alps.

The campaign in the Haslital constitutes a unique set of observations including a long period of observation (ten months comprising winter and summer months), a comprehensive measurement program with not only the MicroWave Radiometer (MWR) and Doppler Wind Lidar (DWL) presented in this study, but also a ceilometer and a mobile X-band weather radar, a
location in a short, deep and moderately wide valley, that differs from most of the studies located in long, wide valleys, and finally the presence of numerous ground stations in the Haslital or the auxiliary valleys to support the results.

The first objective of the campaign is to study the seasonal and diurnal cycles of T and wind in the vertical range containing the main topographical features (590-3000m). The analysis is focused on both climatology and isolated events with a deepest interest on T inversion and foehn events. A comprehensive description of along and cross valley winds is performed, including
a detailed analysis of thermal winds during a heatwave event at three stations and two grid cells of the model. The second objective is to evaluate the NWP model performance (KENDA-1) for two model grid cells in the Haslital between the model's ground level and 3000m. Comparisons with the ground-based measurements and the profiling observations allow to assess KENDA-1 performance for both monthly climatology and peculiar events.

## 2  Methods and Data

The campaign took place in Unterbach (MEE, a subsite in the Meiringen municipality) on the Haslital valley floor from October 13, 2021 to August 24, 2022 in so-called complex topography. A MicroWave Radiometer (MWR) was measuring only since end of January whereas the Doppler Wind Lidar (DWL) and data from the NWP model are available during the whole campaign. Both REM instruments sampled the entire winter and summer months.

### 2.1  NWP model COSMO/KENDA-1

The NWP model used in the study is the limited-area non-hydrostatic atmospheric model from the Consortium for Small-Scale Modeling Model (COSMO) (Baldauf et al., 2011) in the operational setup of MeteoSwiss. It uses a horizontal grid size of 1.1 km and 81 vertical levels with spacings from 20 m at the surface, 40 m at 1000 m, to 160 m at 3000 m and coarsening further up to the model top at 22 km. The levels are terrain-following and a smooth level vertical (SLEVE) coordinate transformation is applied (Leuenberger et al., 2010).





The operational COSMO forecasts are initialized by analyses produced by the Kilometre-scale Ensemble Data Assimilation system KENDA-1, similar to that described in Schraff et al. (2016), but using the above described model setup. KENDA-1 uses a 40 members ensemble of 1 hour model forecasts (first guess) and the following observations: SwissMetNet (SMN) ground station measurements (2m T, humidity and surface pressure), aircraft observations (T and wind from AMDAR and MODE-S), radio soundings (T, humidity and wind), radar wind profiler (wind). In addition, radar-based estimates of surface precipitation

are assimilated in every member using the latent heat nudging method (Stephan et al., 2008). Model first guess and observations are combined using the Local Ensemble Transform Kalman Filter (LETKF, Hunt et al., 2007) to obtain the best possible estimate of the current atmospheric state. The KENDA-1 analysis ensemble additionally uses lateral boundary condition perturbations and stochastic physics perturbations to optimize the spread-error relationship. Besides the ensemble analyses, a deterministic analysis member is calculated, which is close to the analysis ensemble mean (Schraff et al., 2016).

**2.2   Site**

The observational site is located in Haslital, an alpine valley within the Swiss Alps in the Bernerse Oberland (Fig. 1). This 30 kilometer long valley extends from the Grimselpass (2164 m) to Lake Brienz (564 m). The upper 15 kilometers are oriented in the SE-NW direction and present a narrow valley floor with steep surrounding slopes. The Haslital is then joined by a tributary valley called Gadmertal (NE-SW) and continues towards NE with a 1.5 km wide valley floor. About 5 km after the junction, it is

joined by the hanging narrow tributary valley of Rychenbachtal (SW-NE) at the Meiringen village. At this point, the valley gradually bends from NW to W and finally from W to SW as it reaches Lake Brienz. Five km before the lake, the Brünig Pass (1008 m) is an important topographic feature that connects the Haslital to the Sarneraatal, a 30 km long valley oriented in the NE-SW direction (B Fig. B1 presents a detailed map of the Sarneraatal and its connection to the Haslital). This pass interrupts the near constant ridge's height (around 2200 m) north to the valley longitudinal axis (B Fig. B2).


This study focuses on data from two particular sites. The measurements provide from both the in-situ SMN station at Meiringen village (MER) and the site of the REM campaign (MEE) facing the Brünig Pass. These two locations are separated by 4 km

and are respectively at 589 and 574 m.a.s.l. The main difference between these two sites are the valley longitudinal axis angle ($\phi_{MER} = 300°$, $\phi_{MEE} = 270°$) and the relative position to the surrounding connected valleys. Finally, the KENDA-1 data are available for both sites according to the existing model 1.1 km grid.

The location of the 1.1 km grid cell in the 1.5 km wide valley has to be considered. Both used cells include part of the

valley north slope, inducing significant differences between the real topography and the model's terrain. The MER and MEE cells present an offset of respectively 109 m and 130 m. This bias is not only observed in these two cells but in the whole study area. Indeed, the ridges and in particular the Brünig Pass are lowered with respect to their real altitudes, whereas the valley floor is globally raised by a hundred meters (Fig. 2).



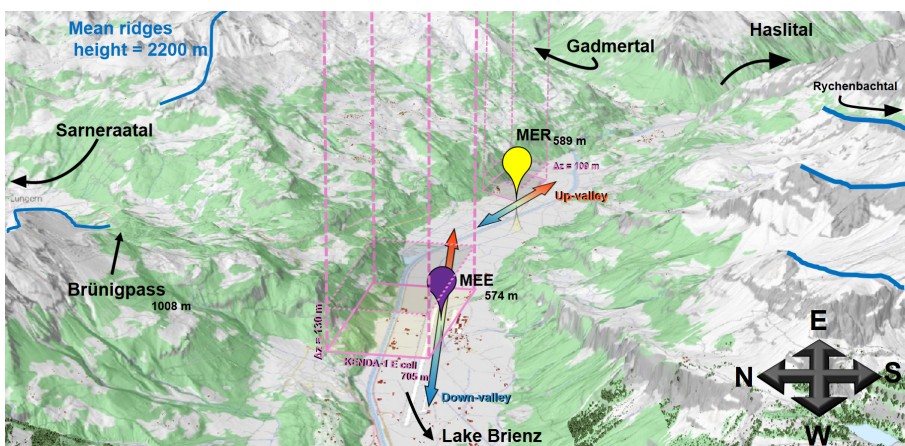

**Figure 1.** Map of the geographical situation in the lower Haslital. The automatic measurement from the SMN in Meiringen (MER) is represented in yellow, the campaign site in Unterbach (MEE) in purple. The two cells of the model used are in pink. Up/Down valley winds are colored respectively in red/blue. The map was download from swisstopo (/https://map.geo.admin.ch, last access: 19.09.2023)



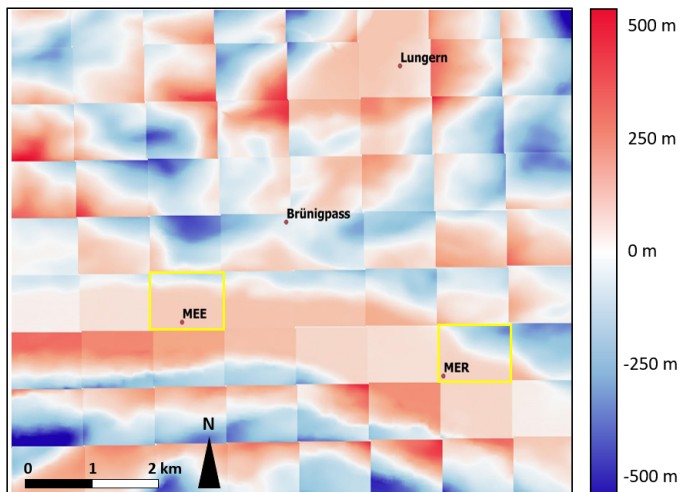

**Figure 2.** COSMO-1 altitude bias [m] relative to the 25 m resolution digital elevation model in the region of Meiringen. The two used cells are highlighted in yellow.

## 2.3 Instrumentation

### 2.3.1 SwissMetNet station MER

The ground measurements in MER are part of the automatic measurement network (SMN) operated by MeteoSwiss (in yellow on fig.1). It continuously measures the main meteorological variables since 1889 and provides near real time data every 10 minutes since November 2011, which allows for making extensive climatologies. The SMN data are composed of T and humidity (2 m a.g.l), surface pressure, precipitation amount, wind speed (mean and gust) and direction (10 m a.g.l.), global radiation, sunshine duration, snow height and an operational foehn index(Dürr, 2008). Surface pressure, T and relative humidity observations of the MER station are actively assimilated in KENDA-1.

### 2.3.2 Additional ground observations in Sarneraatal

SMN also contains a station in Brienz (BRZ) and in the Sarneraatal (B) in the locality of Giswil (GIH). This allows for assessing the influence of the winds originating from this auxiliary valley. MeteoSwiss also cooperates closely Federal Roads Office (FEDRO) that operates its measurement networks. Wind measurements from the Brünigpass (BRU), Lungern (LUN) and Buchholzbrücke (BUC) are therefore also at disposal with similar temporal resolution.

### 2.3.3 Microwave Radiometer

A MWR (HATPRO-G5 produced by RPG Radiometer Physics Gmbh) is used to obtain T profiles and the related mixing layer heights. It collects microwave radiation to infer the T (Rose et al., 2005). It performs a scan every 5 minutes at 11 elevation

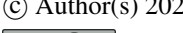



angles and operates in 14 frequencies reception bands in two regions: 22-31 GHz (7 channel filter bank humidity profiler and LWP radiometer) and 51-58 GHz (7 channel filter bank T profiler). The device has an optical resolution of 3.5° (half power beam width) at 22 GHz. The data acquired during rainy conditions are discarded. The spatial vertical resolution increases from 50 m at 250 m.a.g.l. to 300 m at 2500 m.a.g.l. (C) and corresponds to a related T accuracy between 0.25 °C to 1.00 °C, respectively. Löhnert and Maier (2012) found an RMSE between retrieved profiles and radiosonde data between 0.4 and 0.8 K in the lowest

500 m.a.g.l., within 1.2 K at 1200 m and around 1.7 K at 4000 m above ground. However, the performance of an MWR is highly related to the retrieval algorithm and the data used to train the latter (Rotach et al., 2015).

The MWR indirect measurement and the trained retrieval algorithm design for Payerne, the MeteoSwiss reference measurement station can lead to some bias. (Löhnert et al., 2021) showed that obstacles of the surrounding terrain in the line of sight can be a source of external error. The instrument at MER was scanning in the west (275°) direction parallel to the Haslital so that the line of sight of about 10 km should not induce further additional T uncertainty. In simple topography, Hervo et al. (2021) showed

that the HATPRO-G5 can be still biased when compared to radio soundings with a cold bias of 0.5 K around 1500 m altitude.

### 2.3.4 Doppler Wind Lidar (DWL)

DWL can be used to infer wind speeds and direction even in complex topography (Wang et al., 2016). During the campaign, a Vaisala Leosphere Windcube 100S DWL was deployed in MEE to measure wind speeds with a vertical resolution of 100 m.

There are three measurement modes: 120 second zenith scans performed each 10 min to measure vertical wind speed, Range Height Indicator (RHI) scans to measure radial wind speed along and perpendicular to the valley (not used in this study) and Doppler Beam Switching (DBS) scans providing 7 independent wind profiles every 5 min to measured horizontal wind speed. In this analysis the wind profiles were averaged for each 5 minutes interval. Data collected during rain events or/and with confidence level < 90% are discarded. The data availability during the entire campaign is of 80 % at 1000 m a.g.l. and 50% at

2500 m a.g.l.

### 2.3.5 Radio-sounding

The radio-sounding on the 17.11.2022 were performed by a Vaisala RS41, which is a compact lightweight sonde measuring T, wind and humidity profiles with a vertical resolution of 5 m. The three sounding were launched the MEE site at 08:30, 11:00 and 14:00 UTC.

### 2.4 Weather situation during the campaign

During November and December, standard sunshine duration values were recorded, but with only 60% of the precipitation of the 1991-2000 norm in November and 120% in December. These precipitations arrived in form of snow end of November (25 cm of fresh snow) and continued during the first half of December leading to a total snow cover of 40 cm the 15.12.2021. Heavy precipitation occurred then at the end of the month with a snowfall limit recorded at about 2500 m. Consequently, snow

cover remained below the 15 cm mark until the end of the winter. January, February and March were particularly sunny with





strong precipitation deficits in January and especially in March (35 and 15 mm). March experienced frequent foehn events (95 hr determined from the MeteoSwiss foehn index). In April, the weather was changing, with 8 cm of snow precipitation the 01.04. The months from May to August were extremely mild and precipitations were 50% or less compared to the norm, except for June (96%). The first heat wave (6 days) started in mid-June and the second came around mid-July (4 days). Finally, a third heat wave reached Switzerland at the beginning of August. The full evolution of T, precipitation and sunshine duration is aggregated in table A1 and Fig. A1 from A.

### 2.5 Conventions

Unless otherwise stated, the following conventions are valid throughout the rest of the document: data are always reported by the instrument or model name and the site. E.g. MWR/MEE correspond to MWR measurements at MEE and KENDA-1/MER to modeled data from KENDA-1 at the cell comprising MER site, altitude given in meters (m) is equivalent to the altitude above sea level (m a.s.l.), wind speeds are given in km/h and direction in degrees according to north, mentioned times are in UTC format, climatologies are aggregated according to the median hourly values of the studied parameter, median wind speed and direction are calculated by vector averaging the hourly wind vectors, KENDA-1 data refers to the deterministic analysis member, available in hourly time intervals but corresponds to instant values.

### 3 Results

In this section, the measurements of T and wind in the Haslital and its surroundings are presented. The measured climatology for T (6 months) and winds (10 months) is analyzed and then compared to KENDA-1 outcomes. Special cases are treated with particular attention such as surface based T inversion, valley winds and foehn events.

### 3.1 Temperature

### 3.1.1 Climatology

This section focuses primarily on the climatology of measured MWR/MEE temperatures. SMN ground measurements are also used for the description of T inversions.

The evolution of T in MEE from February to July (Fig. 3.a) presents as expected clear diurnal cycle with a vertical extent depending on the season. Layer with higher T develops gradually from sunset to sunrise to reach monthly-related maximal T and height. This layer of warmer air persists during the beginning of the night and then gradually fades out towards sunrise. The time of the T maximum, the persistence of the warm layer and the T range between ground and 2500 m are all enhanced during summer months. Between the mean ridge height and 2500 m, the T remains however relatively constant throughout the day in winter (February). This thermal cycle presents a T rise shortly at sunset from March on, but especially in April and May. This artifact could be caused by the MWR/MEE measurement axis pointed towards West, so that sunshine with low angle at sunset might perturb directly the MWR/MEE T measurement. The maximal temporal T gradient usually follows sunrise and sunset (D



Fig. D1.a) with values up to ± 5 °C and remains below 1500 m whereas vertical negative gradients between -4 and -6.5 °C/km (D D1.b) are coherent with standard values (Lute and Abatzoglou, 2021).

A thermal inversion layer is particularly visible from midnight to sunrise (Fig. 3.a ) for all months in the study. The frequency of occurrence of these T inversions are highlighted by the positive vertical T gradient (D D1.b) near ground (590-1000 m).

In winter, not only nighttime but also daytime near ground T inversions are measured, but they are rare from March apart. A complete analysis of T inversion will be described in section 3.1.4.

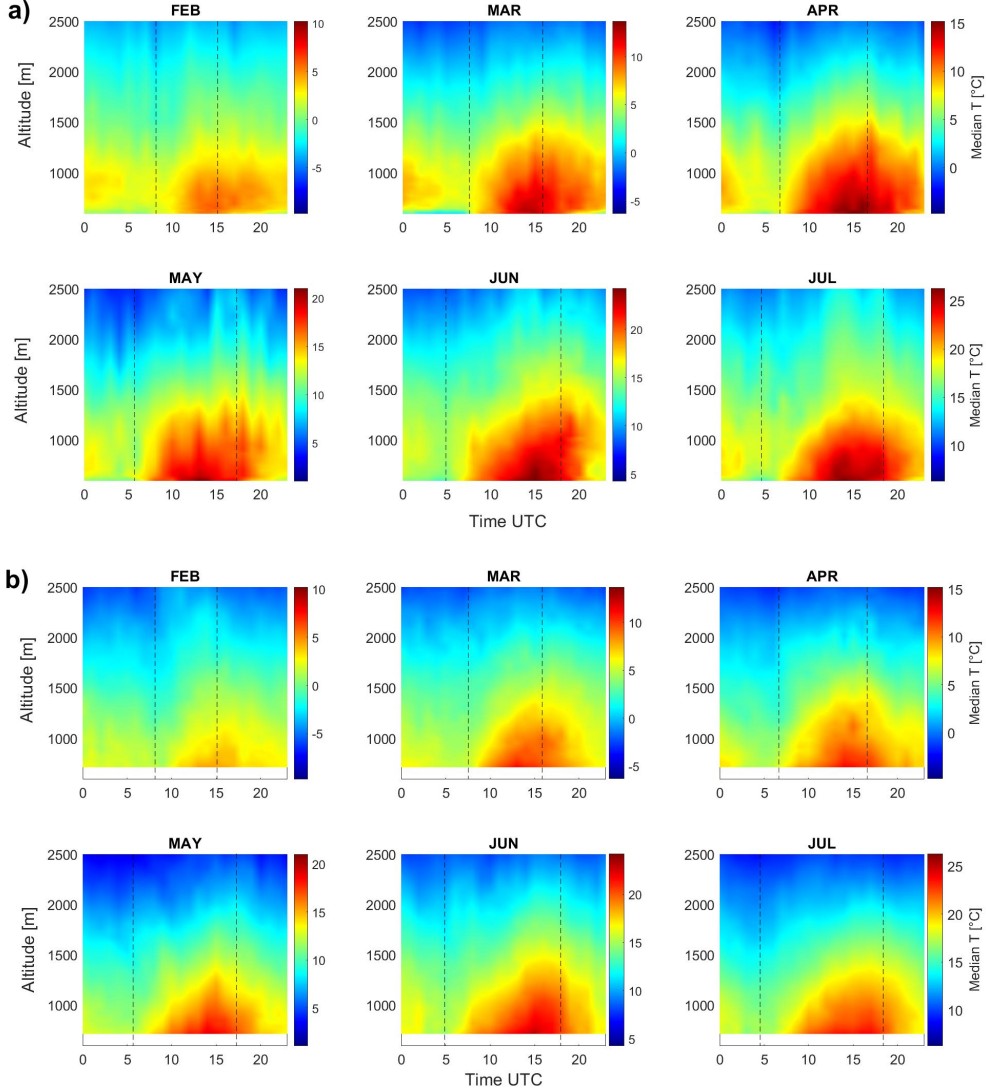

**Figure 3.** a) MWR/MEE T climatology b) KENDA-1/MEE T climatology from February to July 2022 between 590 and 2500 m. Monthly scales with a range of 20 °C but with minimum T based on the MWR/MEE profiles are used.





### 3.1.2 Surface Temperature comparison

In this section, the MWR/MEE ground T at MEE and KENDA-1 ground T at MEE and MER are compared to the measurements
at the MER SMN station, used as a reference.

Ground T measurements from the SMN have a low uncertainty ($\approx 0.2$ °C) and can therefore be considered as the reference
for the comparison with KENDA-1 ground T and MWR/MEE measurements at the first level. The distribution of T differences
between MWR/MEE in MEE and the SMN station in MER (Fig. 4.a) is normally distributed with mean and median close to
zero (-0.07°C) and RMSE equal to 1.45 °C. Extreme values ($3\sigma$) correspond to differences larger than $\pm 4.35$ °C.

     The distribution of ground T differences between KENDA-1/MEE and SMN/MER (Fig. 4.b) is wider than for the MWR/MEE
(RMSE = 2.23 °C) and exhibits a positive skew (median = -0.27 °C and mean = +0.03 °C). Extreme values are significantly
more frequent than for the MWR/MEE measurements, especially in the positive part of the distribution. KENDA-1/MEE
T underestimations occur more often but with lower absolute differences than overestimations, whose differences with the
SMN/MER T reference can reach up to 9 °C. A similar distribution is observed for KENDA-1/MER (Fig. 4.c) with the same
occurrence of extreme T differences (217 hr). Differences under 2 °C represent 71.1 % at MER and 66.0 % at MEE which
explains the slightly smaller RMSE for the cell over the SMN station.

     To check if the altitude differences of respectively 130 and 109 m between the real topography and KENDA-1 ground
level height at MEE and MER could explain the T differences with the SMN station, a standard T correction with a mean
environmental lapse rate (ELR) of -5.25 °C/km (Lute and Abatzoglou, 2021) is applied. This mean ELR is close to the mean
measured MWR/MEE lapse rate of -4.59 °C/km between 590 and 740 m. The distribution of T differences after this correction
(grey in Fig 4.b and 4.c) shifts to higher values leading to a higher mean difference, the right-skewness and standard deviation
being however conserved. The correction leads to higher RMSE at both stations and consequently to a global worse agreement
between the SMN and the model T. The difference in the effect of the ELR correction between MEE and MER is probably
linked to the assimilation of MER measurements by KENDA-1. This type of correction is valid when a large number of values
accounting for various meteorological conditions are aggregated. It is consequently not precise in specific cases (e.g. T inversion),
not applicable to the whole profile and is therefore not further applied in this study. Considering the presented differences
between the SMN and KENDA-1 T as a function of grid cell and ELR correction, we conclude that the horizontal and vertical
distances between the SMN station and the first level of profiles over MEE are not the main causes of KENDA-1 discrepancies
in ground T estimation.

     The T differences between KENDA-1/MEE, KENDA-1/MER, MWR/MEE T and SMN/MER T (Fig. 5) shows a clear diurnal
cycle. KENDA-1 overestimates the T during nighttime (+1.5°C) in both cells and underestimates it during the day (-2°C in MEE
and -1.5°C in MER). The interquartile range (0 to 3.5 °C) and the whiskers (-4 to 8 °C) of the differences are larger during
the second part of the night for KENDA-1 when surface T inversions are more frequent. The presence of this phenomenon
strongly influences the amplitude of the differences and is detailed in the next section 3.1.3. The bias during the day can be
partially explained by the altitude difference between the real topography and the DEM of KENDA-1 since the median T
correction is around +0.65 °C during daytime, reducing the difference by a third. The T difference distribution is similar between





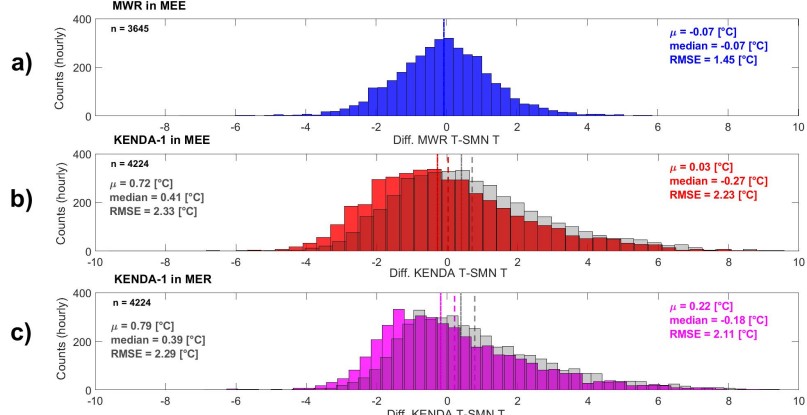

**Figure 4.** Ground T differences distribution for a) MWR/MEE-SMN/MER b) KENDA-1/MEE-SMN/MER, c) KENDA/MER-SMN/MER. The gray distributions are the ground T differences with ELR corrections.

KENDA-1/MER and KENDA-1/MEE during most of the cycle. The differences induced by T inversion during nighttime are the same. The modeled ground T in the cell over MER shows however smaller differences during daytime, which can be explained by the reduced altitude bias or the reinforced assimilation. MWR/MEE also follows a similar cycle with a negative T bias (min. of -1 °C) from 6:00 to 15:00. A slight overestimation is present from 15:00 to 21:00 (max. of + 0.5 °C). The MWR/MEE T

differences present smaller whiskers and interquartile range during the second part of the night than KENDA-1/MEE, but, during daytime, they are similar. Globally, the MWR/MEE first level T are closer to the SMN/MER T than the modeled T.





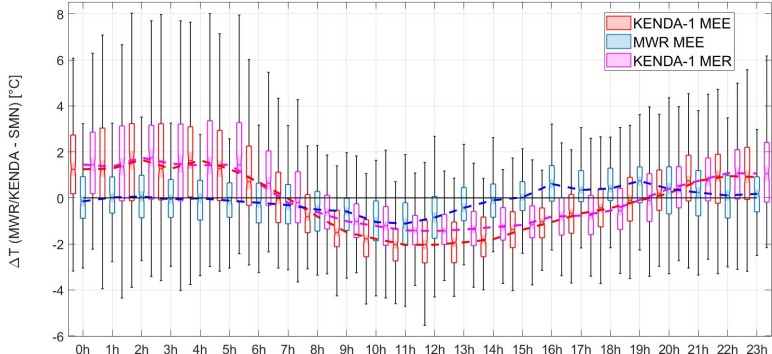

**Figure 5.** Box plots and whiskers of ground T differences between the SMN/MER and the MWR/MEE (blue), the SMN/MERn and KENDA-1/MEE (red), the SMN/MER and KENDA-1/MER (pink) as a function of daytime. The dashed lines represent the median of the different distributions.

### 3.1.3 T profile comparison

The T profile comparisons describes the differences between the MWR/MEE retrieved T profiles and KENDA-1/MEE modelled T between 705 and 2500 m.

The main pattern observed across the different months (Fig. 6) is a general low altitude (< 1500 m) T underestimation from KENDA-1/MEE. In February, this underestimation lasts almost the whole day up to 2500 m, but is larger (< -1 °C) up to 1200-1600 m and below 1 °C difference at higher altitudes. March exhibits the same T underestimation but it is constrained from 15:00 to 05:00 to the lower part of the profile. The underestimation is larger (< -2°C) in the first part of the night due to marked T inversion (section **??** and D). A small T overestimation (< 1 °C) is also observed over the ridges in the morning.

The T underestimation of April extends from midday to late evening. In May and June, underestimations are constrained to nighttime and more important between 700 m and 1500 m. July also exhibits lower altitude (< 1000 m) T underestimation from noon to midnight but also a near continuous T underestimation at ridge level. The latter was, to a smaller extent, already present in June in the afternoon. MWR/MEE and KENDA-1/MEE Ts are however similar between these two underestimated T layers. KENDA-1/MEE mean monthly climatology present lightly lower T than the MWR/MEE T profile with differences

rarely exceeding -2°C.



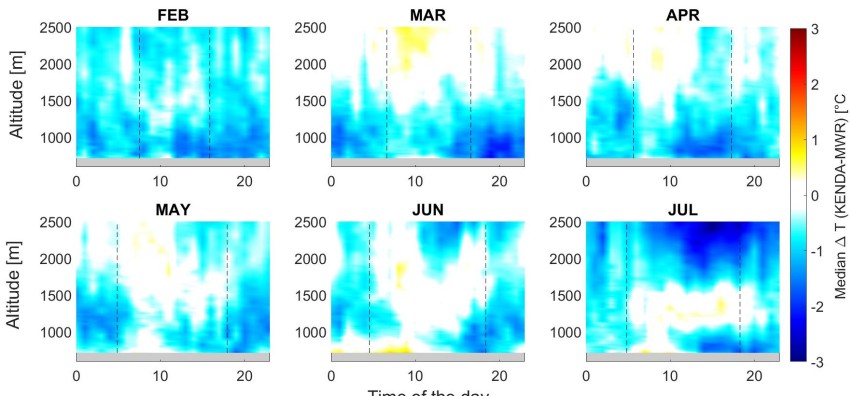

**Figure 6.** Diurnal cycle of the median T profiles difference [°C] between KENDA-1/MEE and MWR/MEE for each month. Sunrise and sunset are given by the dashed lines.

### 3.1.4 Surface T inversion

For the frequency and amplitude comparisons of the T inversions, the data used are ground observations (MER and BRU stations), MWR/MEE profiles and KENDA-1/MEE T.

The analysis of the T profile climatology (Fig. 3), the ground T (Fig. 5) and the T profile differences (Fig. 6) show that
important differences are more common in the presence of T inversion. The analysis of the negative ground T difference between the MER station at 590 m and the BRU station at 998 m (horizontal distance = 3.7 km) shows that near ground T inversions are common during the night for all months in the study (Fig. 7). Their frequency is 60% in December and January (all day long), 40% during the night in spring and 30% during the night in summer. Daytime near ground inversions are rare from March apart and common between November and February (20-60%). The observed T difference follows a seasonal cycle with
enhanced amplitude during winter months reaching up to 4 °C (Fig. 7.b). In summer, this amplitude is reduced to about 2°C and constrained to nighttime. The erosion speed of the T inversion is independent of the month. However, the delay of the erosion onset to sunrise is smaller in summer.

The same analysis between these two similar elevations is performed on MWR/MEE and KENDA-1/MEE T profiles. MWR/MEE shows higher frequencies of T inversion than both the ground stations and KENDA-1/MEE, especially for June and
July. For T amplitude, the maximum difference is relatively constant with ground observation and the model, respectively + 2 °C and + 4 °C. Even if the capability for KENDA-1/MEE to detect the near ground T inversions is enhanced from November to January, their amplitude is always underestimated (Fig. 7.b). Moreover, for March, May and August, the frequency decreases too rapidly after sunrise which can impact the onset time of up valley winds (section 3.2.2). All this leads to both an important overestimation of the T at ground level (Fig. 5) and a slight underestimation of the T just above the T inversion (Fig. 6). To
further illustrate this phenomenon, two full profile's evolution during inversion in winter and summer are available in the appendix (E). The amplitude of the KENDA-1/MEE ground T overestimation is proportional to the amplitude of the T inversion.



As the amplitudes of T inversions are more pronounced during cold months than in summer (Fig. 7.b), March exhibits the most extreme values of ground T differences. It has to be noted that no systematic differences are observed between KENDA-1 T over MEE and MER.

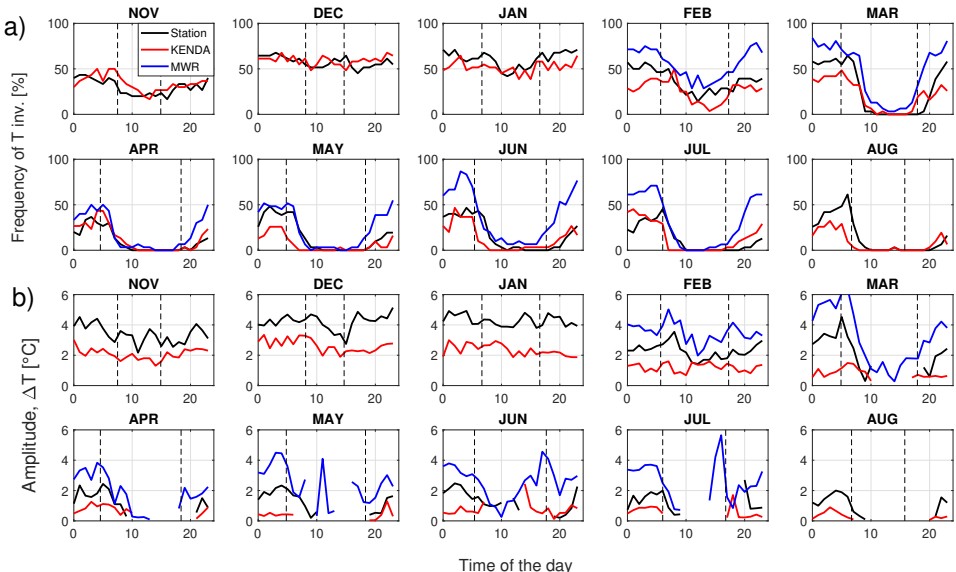

**Figure 7.** a) Diurnal cycle of the hourly T inversion frequency between ground (590m) and 1000 m. T inversion are accounted when $T_{590m} < T_{1000m}$. Observations are retrieved from the differences between SMN T MER and FEDRO T BRU, MWR/MEE data are inferred from the difference between the T at the lowest level (590m) of the profile and 1000 m and the same is done for KENDA-1/MEE (705m-1000m). b) Mean $\Delta T$ for the time where an inversion is detected. Sunrise and sunset are represented by the dotted lines

The analysis of the assimilation process for nights with strong ground KENDA-1 T overestimations shows that the model suffers from a systematic deficiency. During these nights, differences between the model's first guess and observations are mainly around 5 °C and can reach 10 °C in extreme cases (results not shown), so that observations are rejected due to differences exceeding the predefined threshold. The ensemble first guess and its spread as well as the observation error, that is assigned to 2 m temperature observation are the main causes of the assimilation or not of the values (T, humidity, surface pressure) measured

at the surface. During these periods, the SMN T at MER is therefore not assimilated by the model analysis (KENDA-1). Even if the observations are assimilated for some of the KENDA-1 time steps, the assimilation has a very limited effect and allows only minor corrections towards the observations (< 1 °C) during some nights in both MEE and MER.

### 3.1.5   Cloud-topped inversion

A T inversion at higher altitudes was measured on the 17th of November 2021, when three radio soundings (RS) were performed

at MEE. During the three launches at 08:30, 11:00 and 14:00, a cloud-topped inversion was present around 1500 m (F.a-b) and



corresponds to the characteristic fog-top altitude during strong Bise wind (https://www.meteoswiss.admin.ch/weather/weather-and-climate-from-a-to-z/fog-top-height.html). The T-profiles reported by the RS present a sharp inversion that is well reported by KENDA-1/MEE. The inversion is placed at the right altitude during the three launches, but the measured gradient of almost 5 °C/80 m is difficult to grasp by the model due to its too-low vertical resolution at these altitudes. The ability to detect these high altitude inversions may be linked to the fact that the inversion was also present over most of the Swiss Plateau and measured by the RS at Payerne (F.c). The assimilation of RS at PAY probably increases KENDA-1/MEE performance in this case of high atmospheric stability.

## 3.2 Wind

### 3.2.1 Climatology

The climatology presented in this section provide from the DWL/MEE measurement (reference) and KENDA-1/MEE simulations of wind speeds and directions. Ground based 10 m wind compounds were analyzed at the SMN/FEDRO station in MER, BRZ and in the Sarneraatal (BRU, LUN, BUC and GIH). Complete wind profiles (0-3000 m) were only measured at MEE. KENDA-1 profiles are extracted and analyzed over MEE and MER. It is important to take the orientation of the different valleys or valleys' sections into account when analyzing the wind directions at the different stations, especially for the Haslital that bends between MER, MEE and BRZ.

For the climatology of profiles, wind directions are split into two speed categories, below and above 20 km/h, to distinguish between thermally induced valley winds and external synoptic winds, respectively. Fig. 8.a presents the monthly mean wind directions from the DWL/MEE observations at MEE. A clear distinction between winter months (DJF) and the rest of the campaign period is observed:

– Concerning thermally induced winds ($w_s < 20$ km/h), winter months do not show a clear presence of regular direction changes at any altitudes. A predominance of east winds is measured in February and January at low altitudes.

– During March, even though the winds $> 20$ km/h are mainly channeled E winds due to frequent foehn events, the formation of valley winds pattern is already clearly visible. Their time extent (from mid-day to sunset) and vertical extent (up to 1500 m) is similar to November.

– The formation of thermally induced wind is principally visible from April to August ($w_s < 20$ km/h). A classical valley wind pattern is observed at MEE with a simultaneous (for summer months) up valley wind onset and a gradual onset of down valley wind triggered from the ground up to the ridge's height. From 10:00 to mid-afternoon, the direction at low altitudes (800-1000 m) is mainly from W-SW, whereas flows from W-NW are measured in the rest of the profile concerned by up valley winds.

– Simultaneously with up valley winds, low speed N winds are also measured from May to August between 1300 and 1700 m and their intensity increases at warmer months.



- Synoptic winds ($w_s > 20$ km/h in Fig. 8.a) are measured between 2000 and 3000 m from W-SW direction for all months, with a higher variability in January. In December and February, high winds from W-NW are prevalent below 1500 m whereas various directions from N to SE directions are observed for the others months.

- Higher speeds ($w_s > 20$ km/h) N winds are also present from April to November from the ground to 1000-1500 m. These winds originate from Sarneraatal (**??**) during the late morning and several hours after sunset.

- The strong influence of foehn in March leads to winds from E direction up to 2500 m, whereas SW winds are prevalent at higher altitudes.

KENDA-1/MEE wind profiles (Fig. 8.b) are very similar to DWL/MEE observations. One must be aware that the KENDA-1/MEE cell is overlapping the slope towards the Brünig pass so that winds at the junction between Haslital and Sarneraatal can influence the mean modeled wind compounds. The comparison between KENDA-1/MEE and the DWL/MEE observations leads to the further conclusions:

- KENDA-1/MEE models continuous down valley (E) winds ($w_s < 20$ km/h) from December to February between ground and 1000 m. These down valley winds are however not observed by the DWL/MEE in December.

- The main valley wind patterns from March apart are well modeled by KENDA-1/MEE. The main differences concerns the transitions between up and down valley winds. Contrarily to the measurements, the onset of up valley winds during summer months is not simultaneous on the full profile with an earlier onset near the ground. The transition towards down valley wind is also more complex.

- Frequent N flows from the Brünig Pass are modeled between the ground and 900-1200 m with increasing frequency towards sunset, whereas N flows are found at higher altitude (1300-1700 m) in the measurements.

- The influence of the foehn up to 2500 m ($w_s > 20$ km/h) as well as the presence of valley wind below 1200 m ($w_s < 20$ km/h) in March are both well modelled by KENDA-1/MEE.

- For winds above 20 km/h, the match between DWL/MEE observations and KENDA-1/MEE is almost perfect. These flows are more easily modeled since they are mainly influenced by synoptic conditions, which are captured by large grid model inputs and by assimilated measurements (RS, MWR/MEE/DWL/MEE profiles, ...) from the Swiss plateau.

The wind by KENDA-1 at MER is very similar to the results at MEE (Fig. 8.c) taking into account the different orientation of the valley at both stations. In March, the influence of up valley winds is more pronounced in MER with a clear extension up to 2000 m. For the warmer months, the thermally induced wind diurnal cycle is more pronounced and presents with some new patterns. The onset of up valley winds is even more delayed up to higher altitude in spring. Generally, the onset of down valley wind is better defined due to the absence of winds from the Sarneraatal. It has to be noted that up valley winds modeled at MER take almost the same direction (300-310°) as at MEE (290-300°), even if the valley bends ($\approx 30°$) between the two sites, except in the early morning (sunrise to 10:00) when up valley winds come from W at low altitude (from MEE direction). This near





ground direction difference is similar to the observed winds at MEE, but happens earlier (from sunrise) and disappears at 10:00. Modelled down valley winds in MER always follow the main longitudinal valley axis, like in MEE.







**Figure 8.** Monthly median wind direction [°] for a) DWL/MEE, b) KENDA-1/MEE and c) KENDA-1/MER (01.11.2021-23.08.2022). In each case, the data are split according to the threshold wind speed of 20 km/h.





### 3.2.2 Valley winds

To extend the wind analysis, the data from the SMN/MER station, the DWL/MEE and KENDA-1 at MEE and MER, relative to speed and direction, are transformed according to the valley longitudinal axis directions at MER and MEE.

### 3.2.3 Along valley winds

Fig. 9.a shows the diurnal and seasonal cycles of the along valley wind speed at SMN/MER during the campaign, whereas the 10-year climatology (2013-2022) is presented in Fig. 9.b. The occurrence of along valley winds is confirmed by the clear diurnal cycle from February to November. A constant 4 hours delay between sunrise and the onset of up valley winds (> 10 km/h) is observed in the 10-y climatology whereas this delay decreases to 3.5 hours in the summer months 2022 during the campaign. February 2022 shows some early up valley wind but their origin is more linked to synoptic flow intrusions. The transition to down valley winds is closely related to sunset and exhibits a maximum delay of 1 hour in the climatology. In 2022, it occurs 1 hours before sunset in March and June, at sunset in April and May, July and less than hour after sunset in August. The maximum median wind speeds are between 15-20 km/h and occur during daytime. Down valley winds are weaker with a median maximum speed of 2-7 km/h reached within the 2 to 3 hours after sunset.

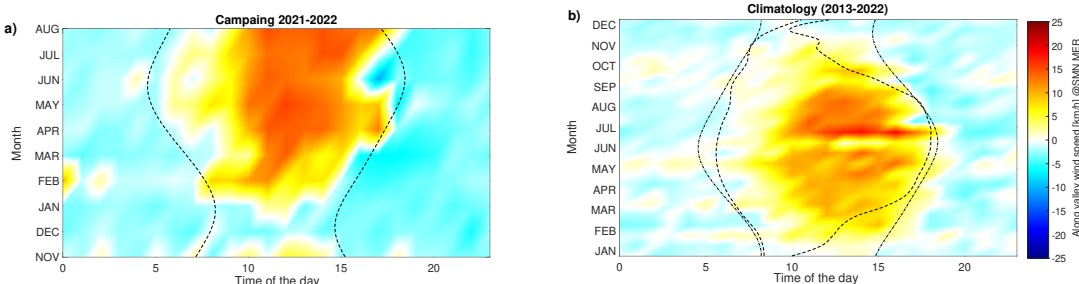

**Figure 9.** a) Monthly evolution of along-valley wind speeds [km/h] at the SMN station in MER during the campaign. Sunrise and sunset are represented with dashed lines. b) 10-year climatology of the monthly evolution of along-valley wind speeds [km/h] at the SMN station in MER. True sunrise and sunset taking shading into account are represented with the dashed-dotted line.

During the campaign, similar diurnal and seasonal cycles of the valley wind at ground levels are measured by the DWL/MEE at 800 m (Fig. 10). The onset of the up valley winds occurs with the same delay to sunrise during the summer months but their speed is of reduced amplitude (10-15 km/h). At 800 m, the up valley wind intensity is also less regular than at ground with maximum speed around noon for May to August. The strongest down valley winds are also measured in the first part of the night, with a higher wind speed (5-10 km/h) than at ground at MER. Additionally, during August, down valley winds occurring 2 hours before sunset are measured by the DWL/MEE whereas the onset to down valley winds occured after sunset at SMN/MER (Fig. 9.a). At 1000 m, the DWL/MEE reports less pronounced diurnal cycles with similar speeds as at 800 m, but the daytime wind direction does not stay constant even during summer months. This might be related to potential turbulence in valley wind regime (Krishnamurthy et al., 2011), especially when synoptic flows interact with thermally driven flows or to influence of flows from





the Sarneraatal. The maximum down valley wind speed also occurs 2-3 hours around sunset. At 1500 m the effect of synoptic winds becomes significant for cold months (November-March). In spring and summer, the up valley winds are stronger and more uniform than at 1000 m and persist longer in the afternoon probably due to the influence of the synoptic winds. At 2000 m, the impact of synoptic winds extends to the warmer months except for July and August where down valley winds are still observed. Finally, at 2500 m, just over the mean ridge's height, the impact of synoptic wind is prevailing for all presented months.

Overall, KENDA-1/MEE shows similar results as the DWL/MEE (Fig. 10.b). The modeled valley winds evolution are consistent with the measurements, including the presence of turbulence leading to daytime varying wind direction. The main differences are a slightly higher up valley wind speed, an underestimation of the down valley winds speed, an earlier onset of up valley winds. KENDA-1/MEE also overestimates the influence of the synoptic winds leading to the absence of along valley wind in winter, higher up valley wind speed, and, e.g. an earlier onset of down valley wind in August at 2000 m. The foehn influence in March up to 2500 m is well modeled.

The difference between the modeled (Fig. 10.c) and measured SMN (Fig. 9.a) data at MER indicates an underestimation of thermally induced along valley wind by KENDA-1/MER, which leads to the absence of diurnal cycle in November and December. Even in summer months, the along valley wind diurnal cycle is less pronounced due to the presence of light up valley wind in the second part of the night. The modeled data at MER and MEE also present marked differences. The along valley diurnal cycle is more pronounced at MER with more constant up valley wind direction at 1000 and 1500 m and higher down valley wind speed, particularly in the morning at 1500 m. KENDA-1 also modeled a lower influence of synoptic winds at MER than at MEE, which is visible at 1500m for cold months and at 2000 m for warm months. These modeled differences between the two cells can be explained by the different orientation of the valley at both sites and by a reduced influence of flows from the Sarneraatal at MER.

The mean monthly diurnal cycle as a function of altitude observer over MEE (Fig. 11.a) allows a better visualisation of the vertical extent of valley winds. First, the height of the down valley wind determines the limit of the influence of SW synoptic winds. Thermally induced wind height increases with temperature, reaching only 1000 m in February, 1800 m in May and up to 2000-2200 m in July and August. Second, the onset of up valley wind occurs simultaneous (about 3-4 hours after sunrise) over the full valley wind extent, whereas the onset of down valley wind is delayed from ground to its maximal extent. (Fig. 11.a) shows that up valley wind can persist until 1-3 hr after sunset at the ridge's height. Finally, down valley wind speed decreases with altitude and with time after sunset.

The same representation for KENDA-1/MEE (Fig. 11.b) shows that the vertical extent of the modeled valley wind is comparable to the observation ($\pm$ 100 m) expect in April. The main differences are first, an underestimation from ground to 1600 m of the down valley wind speed until midnight in summer and, second, a too short delay (1-2h) after sunrise of the onset of up valley winds between the ground and 1200 m. These statements hold from April to July but not for August. In winter, KENDA-1/MEE overestimates the presence of constant down valley winds below 1200 m. However, in November, the modeled profiles show continuous up valley winds between 1000 and 1700 m where the DWL/MEE measures mostly down valley winds.




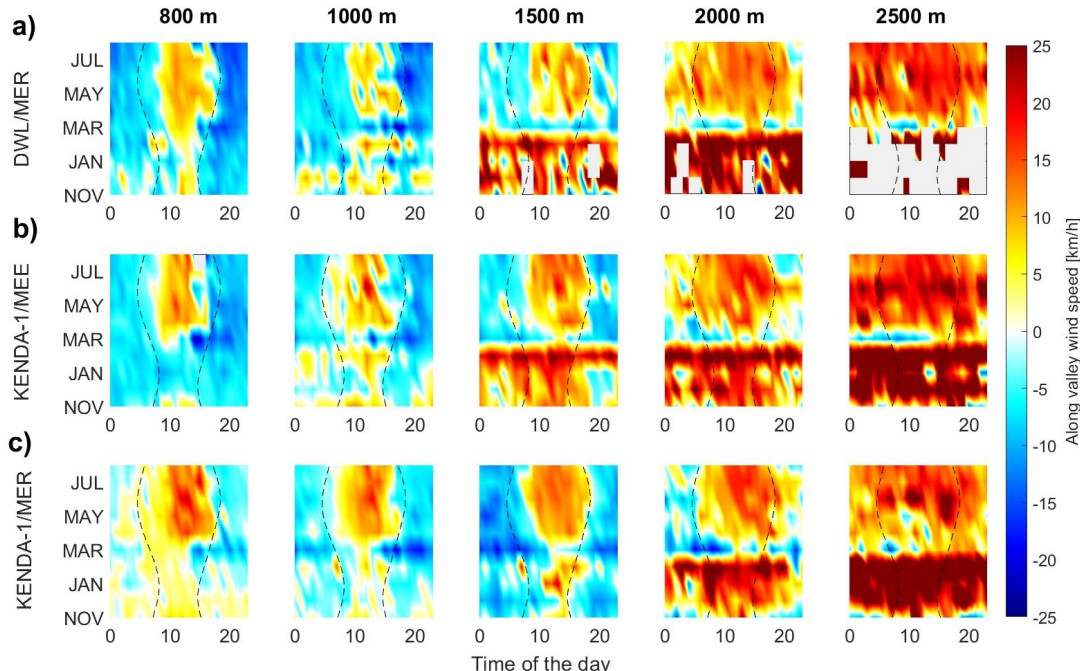

**Figure 10.** Monthly evolution of the along-valley wind speeds [km/h] for a) the DWL/MEE, b) KENDA-1/MEE and c) KENDA-1/MER. From left to right: along-valley wind speeds at 800, 1000, 1500, 2000 and 2500 m. Sunrise and sunset are represented with dotted lines.

### 3.2.4   In-depth analysis of selected clear summer days

A closer look at the SMN and DWL/MEE wind speeds during a series of clear warm days in July with low cloud amounts (Fig. 12) shows some particularity relative to the previous analysis of monthly median values.

In MER (Fig. 12.a), a clear diurnal pattern of thermally induced winds is measured. The onset of up valley winds occur at 10:00 and the wind speed strengthens during the day (approximately +4 km/h per hour) to reach a maximum of 25 and 30 km/h at 15:00-16:00. The onset of down valley winds occurs at 19:00. During night, down valley winds are constant in direction and drop to 0-5 km/h. It has to be mentioned that the direction of up valley winds at MER gradually shifts from the longitudinal axis of the Haslital towards an enhanced northern component on the 10 and 11 July during the afternoon.

In MEE, DWL/MEE measurements are reported at 200 m.a.g.l. Up valley wind is only measured on the 10 July around 08:00 and 12:00 (Fig. 12.a). at 13.00-14:00, the wind direction switches to N and the wind speed increase gradually to reach 40 km/h at 20:00. The wind then weakens until midnight and changes direction afterward with a down valley wind direction that persists sometimes (e.g. on the 12 July) during the morning. Along valley wind patterns following the valley longitudinal axis (W/E) are only observed between 1300 m and 2000 m, namely higher than the Brünig pass altitude. They present then a standard diurnal cycle with up valley wind measured from 09:00-10:00 to 16:00-17:00 with wind speeds between 15 and 20 km/h.





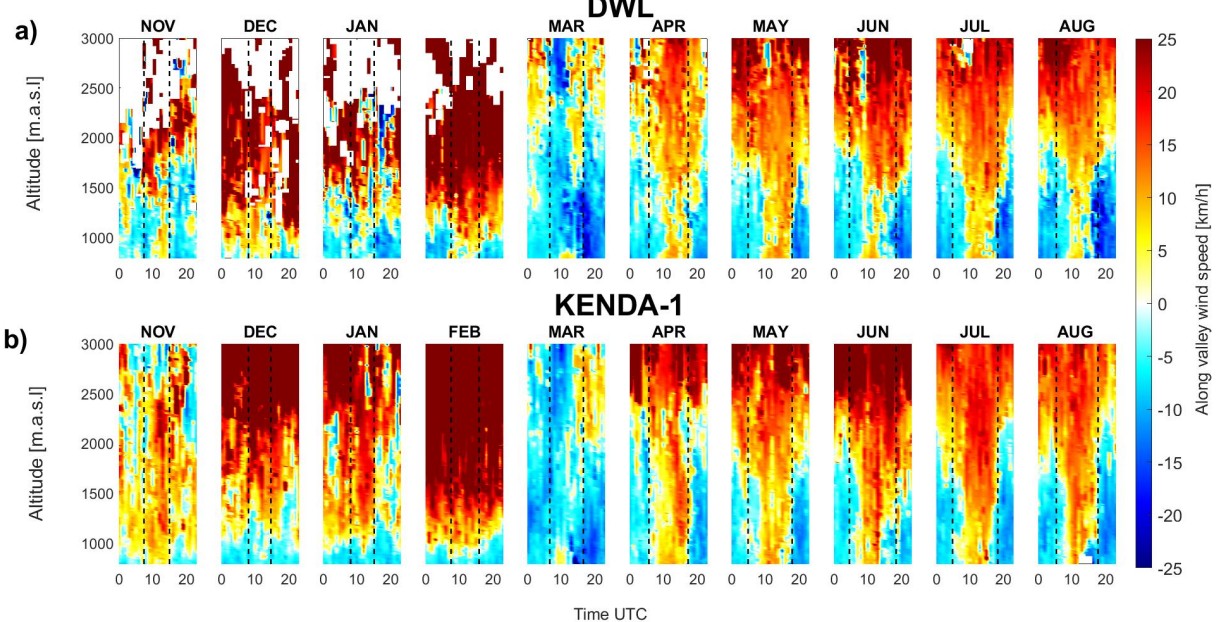

**Figure 11.** Monthly mean diurnal cycle of the along-valley wind component [km/h] as a function of altitude for a) the DWL/MEE observation and b) the KENDA-1/MEE data. Sunrise and sunset at ground level are given by dotted lines.

In BRZ, the wind pattern is very different for the 3 selected days (Fig. 12.a). July 10 and 12, up valley wind begins at 8:00
and last until 14:00 with low wind speeds between 5 and 10 km/h. At 14:00, the wind direction switches towards down valley winds (17-19 km/h), which last until 20:00. Small direction change towards WSW occur during the night. July 11, there is no up valley wind phase; the directions remain between NE and E corresponding to down valley wind. The wind speeds are lower in the morning and strengthen to 20 km/h in the afternoon before to drop at 21:00.

The strong influence of the thermal winds from the Sarneraatal over the Brünig pass is evidenced by this analysis of the wind
at the three stations during hot summer days. An analysis of ground measurements from the BRZ, BRU, LUN, BUC and GIH (G) automatic stations shows that flows measured at the Brünig pass switch towards the Haslital (SSW) 2 to 3 hours earlier (5:00-6:00) compared to other stations of the Sarneraatal (08:00-09:00) and last much longer after sunset, up to 21:00-22:00. The low altitude difference between the Brünig pass and the Haslital floor (400 m) explains the wind diurnal cycle measured at low altitude at MEE, that is characterized by N wind from the Sarneraatal during the afternoon, the early evening and also sometimes in the morning (e.g. on the 11 of July). These winds also strongly influence the diurnal cycle at BRZ leading to the onset of
down valley winds in the early afternoon or even by suppressing the occurrence of up valley winds (July, 11). Their influence at MER is however weak with only a slight shift of the wind direction towards N in the late afternoon. During these summer days, a standard thermal wind diurnal cycle is then observed in MER and in MEE at altitudes higher than the Brünig pass.

Concerning the modelled data (Fig. 12.b), KENDA-1/MEE also take into account the influence of the Sarneraatal thermal
winds so that the differences between the two analyzed sites (MER/MEE) are important. At MER, the wind speeds and direction



follow a clear thermally related valley wind cycle with an onset of the up valley wind from 09:00 apart, a maximum wind speed of 30-35 km/h around 14:00 and an onset of down valley wind at 18:00 with decreasing wind speeds (3-5 km/h) until the next morning. At MEE the modeled winds show a relatively stable wind direction from NE during nighttime and NNE during daytime. Speeds are always equal or higher than over MER with weaker diurnal cycle. The major differences compared to

the observations (Fig. 12.b and (H Fig. H1.a) are a too strong influence of the valley winds from the Sarneraatal leading to no modeled down valley winds at MEE during the night and the morning as well as a wind direction shifted toward N at MER. The differences of the wind speed diurnal cycles at MER and MEE are well modeled by KENDA-1, but the wind speed is almost constantly overestimated at both sites with differences up to +30 km/h.

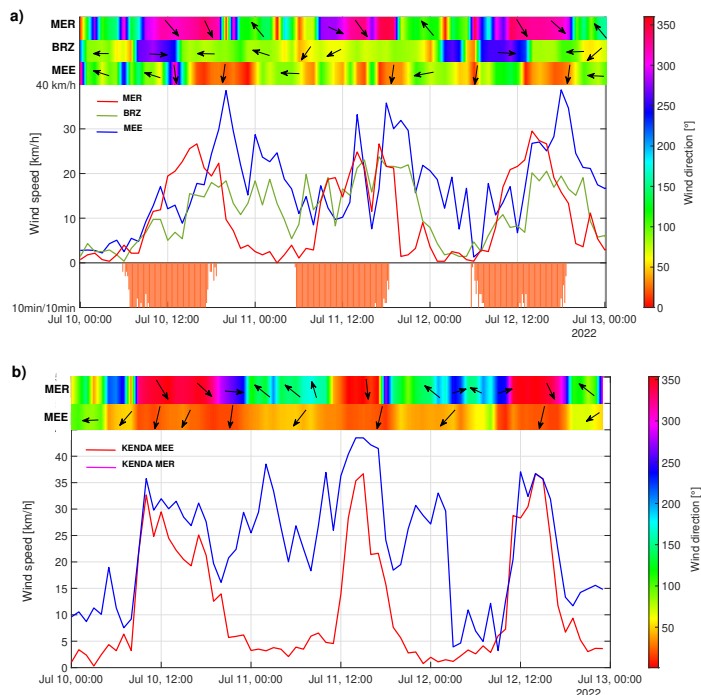

**Figure 12.** a) Measured b) modeled wind speeds (solid lines), wind direction (colored bands and arrow) and sunshine amount for a) the DWL/MEE (800m), the SMN/BRZ, the SMN/MER and b) KENDA-1/MEE (800m) and KENDA-1/MER (800m).

### 3.2.5 Cross valley winds

The cross-valley winds at MEE can originate from thermally induced slope winds in the Haslital or from valley winds from the Sarneraatal passing over the Brünig pass. Slope winds are however more likely to be detected by KENDA-1/MEE cell overlapping the south facing slope (Fig. 1) than by the DWL/MEE located in the middle of the valley. Fig. 13.a shows the mean monthly diurnal cycle of the cross-valley wind measured by the DWL/MEE. During winter, the data are scarce and no particular pattern is visible except the presence of Brünig pass N winds from 800 m to 1500 m in January and February. Cross valley



winds originating from Brünig pass are measured up to 1500 m during all other months with a wind speed of 20-25 km/h. These
N winds start between midday and sunset and stop around midnight. They are generally first measured near the ground and
reach 1200-1500 m after sunset. Intense north-facing slope winds ($> 25$ km/h) are also observed between 1400 and 2000 m
during some hours around sunset with a much lower intensity in May. This suggests a circular motion with North updraft winds
(median vertical velocity of 1 km/h) that cross the valley at a low altitude, rise against the north facing slope and come back at
higher altitude with a South downdraft component (median vertical velocity of -2 km/h).

KENDA-1/MEE also shows cross valley wind patterns (Fig. 13.b) with strong winds from Brünig pass from March to August.
These N winds also develop progressively from ground to 1400 m and stop around midnight. They are however modelled earlier
than measured, at the time (10:00) of the onset of up valley winds in the Sarneraatal (G). The absence or weakening of up valley
winds before sunset in both observations and modeled data (Fig. 10 and 11) are somehow related to these strong cross valley
winds at MEE. Winds from the north facing slopes between 1400 and 1800 m are never modeled by KENDA-1/MEE despite
their intensity and being systematically measured.

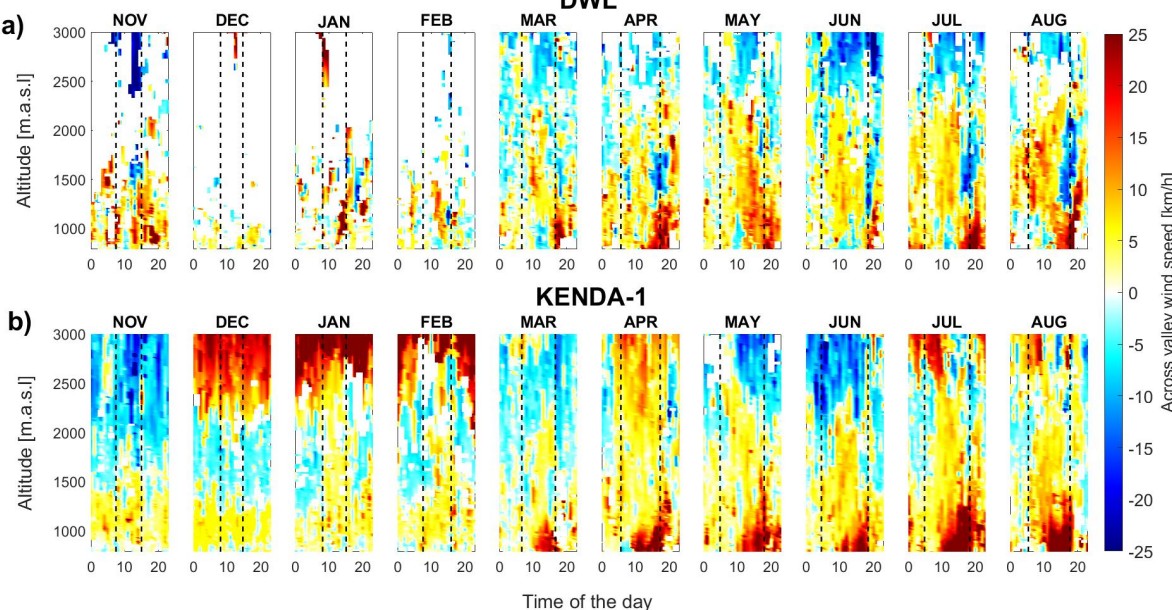

**Figure 13.** Evolution of the diurnal cycle of the across-valley wind component [km/h] as a function of altitude for a) the DWL/MEE
measurement and b) the KENDA-1/MEE. Winds coming from the south-facing slopes take a positive value (red), for the north-facing slope
wind speeds values are negative (blue). Sunrise and sunset at ground level are given by dotted lines.

### 3.3 Foehn events

Foehn in a katabatic wind bringing generally strong warm and dry downdraughts associated generally with clear weather
(foehn window). The following case study combines all the periods where foehn was measured at the SMN station in MER,

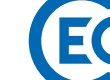

according to the foehn index. The selected data represent 117 hours mainly occurring in March. Three events have been selected (10.03-16.03/19.03-22.03/26.06-27.06) corresponding to clear weather in March, while the April and June episodes presented a slightly overcast sky (50-70% of maximum global radiation).

### 3.3.1 Temperature

14.a shows that the MWR/MEE tends to measure lower T than the SMN/MER. However, the 0.5 to 1.5°C differences can be
explained by the small size of the data set as well as the different locations and altitudes between both T measurements. In contrast, a significant T underestimation (-2 to -4 °C) by KENDA-1 is observed during foehn events, without the mean diurnal cycle measured during the whole campaign (Fig. 5). Furthermore, the differences categorized according to wind speed (Fig. 14.b) show that larger wind speeds (> 20 km/h) induce larger median T underestimations. Saigger and Gohm (2022) performed simulations in the Inn valley with the Weather Research and Forecasting model and observed similar bias at low altitudes during
an intensive foehn event. Additionally, Tian et al. (2022) also report significant cold and moist biases are found in the model during foehn hours. Note that the KENDA-1/MER is in better agreement than KENDA-1/MEE with SMN/MER (not shown), which can indicate significant differences in the foehn influence at the two stations.

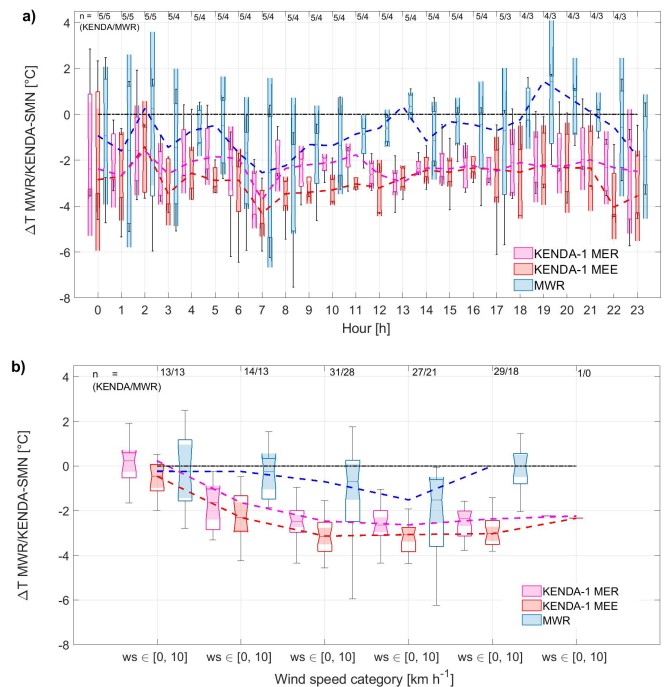

**Figure 14.** Box plots and whiskers of ground T differences between the SMN/MER station and the MWR/MEE (blue), the SMN/MER station and KENDA-1/MEE (red), the SMN/MER station and KENDA-1/MER (pink) as a function of a) the hour of the day and b) the 10 m measured wind speed at SMN/MER. The dashed lines represent the median of the different distributions.



The comparison of T profiles during foehn events (I, Fig. I1.a) shows that KENDA-1 at both MEE and MER underestimates the T not only at the surface but up to 900-1400 m depending on the event. In some cases, KENDA-1 missed the T increase due to foehn but in other cases, KENDA-1 follows the T evolution but with a smaller T gradient. The mean T bias of 2-4°C is similar along the profile (mettre reference fig ap 26) and at the surface and is reinforced when a T inversion missed by KENDA-1 precedes the foehn event. The T increase due to the foehn breakthrough measured by the MWR/MEE is delayed by less than one hour compared to the SMN/MER detection. Similar time delays of about one hour are modelled by KENDA-1, with shorter delay at MER than at MEE as expected by the orientation of the Haslital and the provenance of foehn.

### 3.3.2 Wind

DWL/MEE measurements (Fig. 15.a) shows the extend of the higher wind speeds induced by the foehn from ground to the mean ridge's height (1800-2000 m). The foehn breakthroughs are nearly simultaneous at ground (SMN/MER) and up to 1000-1500 m at MWR/MEE for the events of March 11 and April 23. However, an important delay of $\approx$ 3 hr is measured between 800 and 1300 m for March 20. Foehn winds are however measured from 1300 m up to the ridge at the same time as at the SMN/MER station. The maximal measured wind speeds at 800 m in MEE (60-75 km/h) are higher than at the SMN/MER (45 km/h), especially for the event of March 11.

During the first selected episodes (11.03) the foehn arrival is modeled 2 hr too early by KENDA-1/MEE (Fig. 15.b) with strong winds (60 km/h) from SE between 800 and 1000m. At 11:00, the measured foehn arrival, the KENDA-/MEE wind direction is coherent with measurements but speeds are overestimated between ground to ridges with differences up to 15-30 km/h (J, Fig J1). This overestimation lasts for the first 4 hr and then turns out to a KENDA-1/MEE underestimation of wind speed. This happens for 2 hr and then finally KENDA-1/MEE is in accordance with measurements for the rest of the event. For KENDA-1/MER (Fig. 15.c), the same delay in the foehn breakthrough is observed. It happens over a larger extent (800 to 1200 m) with even higher wind speeds (> 100 km/h). This overestimation is constant during the entire event.

For March 20, the 3h delay between SMN/MER and DWL/MEE measurement is modelled by KENDA-1 (Fig. 15.b and Fig. 15.c) but, contrarily to DWL/MEE, up the ridge height. The KENDA-1/MER wind speeds tend to be overestimated (+15 km/h) from ground to 1100m during the entire event. From 1100 m to the ridge's height the wind is underestimated (-30 km/h) during the first 5 hours. At the very end of the event (March 20 at 23:00), KENDA-1 keeps showing strong foehn winds between 900 and 1500 m while nothing is measured by SMN/MER and DWL/MEE. At MER, KENDA-1/MER modelled again wind speeds up to 100 km/h with a foehn breakthrough at the same time as the SMN/MER.

To summarize, the modeled data show a better representation of the temporal extent of the foehn events over MER than over MEE. Despite the good temporal representation of foehn arrival, the speed up to 110 km/h from ground level to 1500 m is twice as big as what is measured at MEE, 5 km further down in the valley. Even though the Haslital is narrower just before MER, such wind speeds difference is subject to a discussion about a potential large overestimation of the winds at this location. The three analyzed events exhibit some similarities but also large differences. Literature (Jansing et al., 2022) cites three main foehn types related to distinctively different synoptic situations for Switzerland. During the most typical deep foehn situation occurring 90%



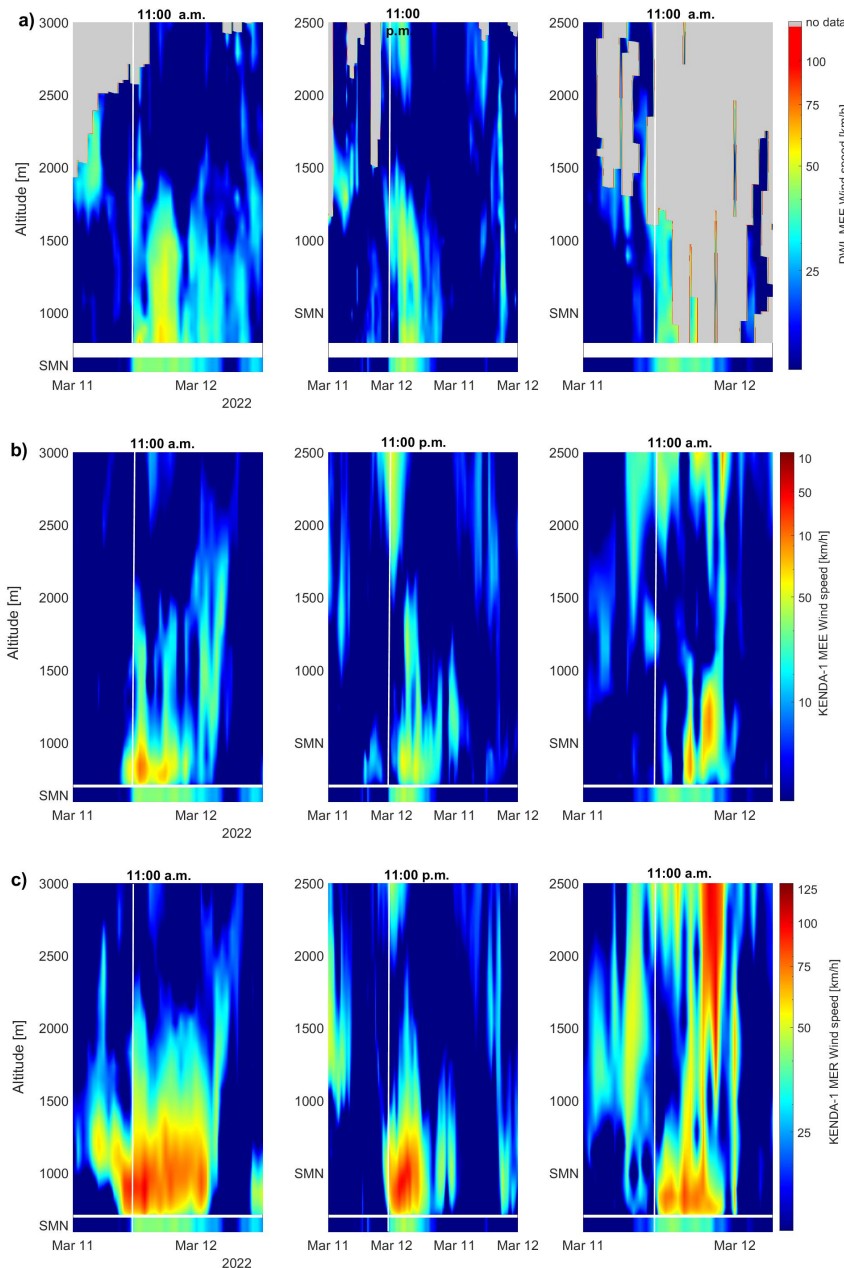

**Figure 15.** Wind speed profiles [km/h] time series from a) DWL/MEE, b) KENDA-1/MEE and c) KENDA-1/MER during a selection of 3 foehn events: left 11-12.03.2022, middle 19-22.03.2022 and right 23-24.04.2022. Wind speeds [km/h] from the SMN/MER are given in the lower part of each figure. The solid line represents the foehn breakthrough.





of the time, the foehn flow is accompanied by a deep layer of southerly or southwesterly winds. The 3 selected cases can be characterized as deep valley foehn.

The simultaneous wind speed overestimation and the T underestimation by KENDA-1 during foehn events are difficult to explain. A stronger foehn should allow for a greater T increase. A moist bias of the model is a hypothesis for the observed differences.





## 4 Discussion

### 4.1 Topographical and methodological challenges

The difficulty to model the physical phenomena taking place in complex topography comes first from the complexity of the atmosphere itself, which can be composed of several layers with thermal structures and wind regimes inducing enhanced mass and energy exchanges. Additionally, in regions with complex topography, the terrain affects the distribution and exchange of energy between air masses. Finally, the Haslital has several peculiar topographical characteristics along its path and particularly near the campaign site (Figs. **??**). Its junction with the Sarneraatal, 4 km downstream, via the Brünigpass links the two valleys with an angle of ≈ 90°, 400 m above the valley floor. Considering this small altitude difference, this pass can even be considered as a tributary inlet. It allows the winds from the Sarneraatal to easily reach the Haslital with a cross-valley wind component, similarly to down slope winds and to disturb its along valley wind system (Fig. 10). The location of MEE just under the Brünig pass has to be taken into account for comparison between MEE and MER results. Based on numerical simulation in the Alpine Inn Valley, Zängl (2004) suggests that variations in wind intensity are mainly related to tributary valleys, which increase or decrease the mass flux in the main valley. As such low pass can have similar effects as tributaries.

The difference between the location of the observations and the extension of the model cell over MEE needs to be taken into account. Indeed, a part of this cell overlaps the slopes towards the Brünigpass while the observations over MEE are characterized by a single point located approximately in the middle of the valley floor, SW to the pass 1. The model cell only gives an averaged value of T/wind within it, which will induce bias when compared with high horizontal resolution measurements. Consequently, differences between the model and the measurements cannot only be categorized as model errors. For instance, KENDA-1 cell lies on the pass area implies that wind originating from there are completely comprised in the cell while along valley wind from the Haslital bottom part are only partially comprised in it.

Additionally, the change in orientation between MER and MEE implies that the same valley side faces different orientations along the Haslital and consequently different heating time by the incoming solar radiation of the system (Fig. 1), slope at MER being more orientated towards E than at MEE. The presence of large lakes covering the entire valley floor in its lower part, 5 km after MEE, modifies the heat exchange between the surface and the atmosphere due to their high thermal inertia. The presence of lakes can influence the T profiles along the valley, and consequently the pressure difference, as well as the time, extent and strength of the thermally induced valley winds. When comparing observed phenomena with similar studies, the combination of the above mentioned peculiar features gives explanatory hints for the observed differences.

Finally, most of the data representation are using the monthly median value of hourly data. No filtering of data relative to particular weather type (except for precipitation periods) is done. It allows to get ride of special meteorological cases in a climatological way but does not allow to evaluate particular cases. The effect of averaging has to be taken into account for the analysis of maximum valley wind speed or onset of valley wind. This is especially the case for wind directions: the aggregation





can lead to angles that are never observed. Therefore, this study does not allow to make prediction of model performance for forecasting.

## 4.2   Comparison of observed phenomena with other studies

### 4.2.1   T inversion

T patterns in MER follow a classical diurnal and seasonal cycle. The most important feature in the context of this study is the
presence of frequent ground T inversion (Fig. 3.a, 7.a). According to a 3 years study in the French Jura performed over 16 station pairs (Joly and Richard (2019)), T-inversions are equally common in winter and summer (60% of the time), but with a larger amplitude (3°C) in winter than in summer (2°C). Additionally, a study conducted by Rupp (2020) examined 13 years of hourly air temperature data in the Cascade Range, USA, at comparable altitudes. The findings reveal that temperature inversions occurred more than 50% of the time, with the formation and dissipation of inversions consistently lagging behind local sunset
and sunrise times by approximately four hours. Finally, a 56-year climatology in the Austrian Alps (Hiebl and Schöner, 2018), shows that T inversions occur throughout the year with a frequency of about 30%, from October to January and 15% from April to August. Their intensity, magnitude and thickness follow a similar seasonal pattern as observed in the Haslital. Inversions are more frequent in eastern Austria and less frequent in the large western valleys and basin of Austria and almost vanishing in high-Alpine summit area. This study in the Haslital show similar near ground T inversion occurrence (7.a). During the campaign
(Nov-Aug), T inversions were present 30% of the time between the two ground station (MER-BRU) and for 40% of the time in the MWR profile. Amplitudes are similar to the results from Joly and Richard (2019) with slightly higher number during winter month (+ 1°C). The seasonality of the phenomena is mainly characterized by the frequency of T inversion along the day in winter and the onset of the erosion process. Similar number than (Hiebl and Schöner, 2018) are found concerning the evolution of the phenomena throughout the months: 14.3% from April to August and 46% from November to February.


### 4.2.2   Valley winds

Previous works have been done on diurnal valley winds in alpine valleys. The result obtained are compared with studies performed in the Rhone (Length = 140km, Floor Width = 4-5km, Ridge to Ridge Width = 15km, Schmid et al. (2020)), in the Adige
(L = 140km, BW = 2-3km, RRW = 8km, Giovannini et al. (2017)) and in the Inn valleys (L = 140km, BW = 4-5km, RRW = 20km, Adler et al. (2021)). The valleys of these 3 studies are relatively long and wide compared to the Haslital (L = 30km, BW = 1.5km, RRW = 5km, Fig. B2), which can induce differences in the thermal valley wind systems. All three studies make a selection of valley wind days by using threshold on minimum global solar radiation or up valley wind speeds and selected global weather type.

Similarly to the observations in the Haslital, the wind direction change in the Rhone valley (Schmid et al., 2020) occurs for altitudes up to about 2 km a.g.l. with diurnal pattern undergoing significant changes during the course of the year. During



warm weather, mean maximum up valley velocities of 30-35 km/h are found above the Rhone valley between 15:00 and 16:00 at altitudes around 200 m.a.g.l. Similar timing for maximum up valley winds are found at both MER and MEE, but with reduced speed both at ground in MER (20-30 km/h) and at 200-300 m a.g.l. over MEE (15-20 km/h). The applied monthly median,

without selection of days favorable to thermal valley winds, can also explain the lower maximal up valley wind speed in the Haslital, but other causes bounded to the valley size and to further topography features cannot be excluded. At MEE, in fact, the highest wind speeds of 30 to 45 km/h are found later on, at 18:00 and 19:00, between 800 and 1400 m. They correspond to valley winds from the Sarneraatal with a direction of 50° and contribute to the main down valley wind component. They are also observed at the same time at LUN and BRU (Fig. 13 and G1) and are strong enough to flow into the Haslital. The N-S

orientation of the Sarneraatal with west facing slopes could allow this persistence but no past study mention this phenomenon. These winds from the pass explains the difference between the along valley wind speeds pattern at MER (Fig. 9) and at MEE (Fig. 10.a). This flows from the Sarneraatal can also explain the time and altitude differences of maximum measured wind speeds with the Rhone valley, even if inlet of tributaries are also present near the campaign site at Sion. Due to the low altitude of the Brünigpass, the Sarneraatal can be considered as a tributaries, but with effect at higher altitude. Concerning down valley wind

speeds, Schmid et al. (2020) report their presence between 500 and 1000 m.a.g.l with a speed of about 15-20 km/h. They occur in the second part of the night in spring and summer, and during the entire night in winter. Several differences are observed in the Haslital: 1) down valley winds reach ground even in summer (Fig. 9) and extend up to 800-1000 m.a.g.l., 2) their speed gradually decrease around the night with almost no wind between 00:00 and the new onset of up valley winds for all the studied period and 3) at MEE, maximum down valley wind speeds are measured from March to July at the same altitude as in the Rhone

valley but with lower wind speeds (10-15 km/h). If the last difference can be also explained by the applied monthly average, the timing and extent of the down valley winds probably relates to topography differences.

The work of Giovannini et al. (2017) in the Adige valley in the Italian Alps in May-August shows similar results as found in the present study. The wind speed measured at ground show maximum up valley wind speeds between 15:00 and 16:00 being

stronger near the valley outlet (20-30 km/h) that then gradually weaken towards the highest valley parts (8-10 km/h) situated 100 km further up. Surface down valley wind speed appears to be very weak, between 0 and 5 km/h, and nearly constant in the entire valley. However, their mean onset is delayed to 00:00 which is not the case both in the Haslital, for the entire climatology (Fig. 9) or for selected valley wind days (Fig. H1.a), and in the Rhone valley. Wind profiler data from the outlet of the Adige valley show that the strongest up valley winds are recorded between 1000 and 1400 m.a.g.l. at 18:00. Similar late, maximal

up valley wind speeds are also recorded at MEE (Fig.11.a) if Brünigpass winds are accounted for up valley winds (Fig. 13). Contrarily to both Schmid et al. (2020) and this study, the down valley winds of the Adige valley gradually weaken towards higher altitudes at 00:00 and 02:00. For the rest of the night, stronger wind are also found between 500 and 1000 m.a.g similarly to the observation in the Rhone valley (Schmid et al., 2020).

Finally concerning the onset of up valley wind, both the time and the pattern of the transition are similar between the Rhone, the Adige and the Haslital valleys. The onset occurs 3-4 hours after sunrise with flows that move almost simultaneously





between 0 to 1500 m.a.g.l (Fig. 8.a) from June apart. Indeed, the same is observed by Schmid et al. (2020) at the same season and by Giovannini et al. (2017) for which almost all cases are in spring and summer. The fact that morning transition occurs at the same time at all heights while the up-valley wind starts weakening from the bottom due to progressive cooling of the

lowest atmospheric layer (as Zängl (2004)) and down wind starts at ground and thickens during the night. Note that, Schmid et al. (2020) reported a delayed onset as a function of altitude in autumn but unfortunately, no data were acquired in the Haslital during this period.

    The CROSSINN campaign (Adler et al. (2021)) was performed from August to October in the lower part of the Inn valley and focused on cross valley winds. During two days of September, the wind field in the vertical plane across the valley show

mean subsidence around 13:30 and 14:30 without any particular cross valley wind direction above the valley floor center. In the second part of the afternoon (15:00-17:00), the valley atmosphere presents an enhanced cross valley wind circulation. Over the south facing slope of the valley, subsidence prevails, while over the north facing slope upward motion is measured. This flow pattern form a closed circulation cell with a clear cross-valley component with a northerly component in the lower 700 m.a.g.l. and a southerly component above. Similarly to the Inn valley, the Haslital at MEE lies in the E-W direction (slope however

reversed). A cross-valley phenomena is also measured in MEE from March to August (Fig. 13.a), but especially during Summer. In the Haslital, the separation between north and south facing wind lays between 700 and 1000 m.a.g.l. The particularity in MEE is that the lower part of this cycle (winds from the south facing slope) is probably mainly due to valley winds from the Sarneraatal. However the upper part of the cycle is a good clue that this type of circulation occurs in the Haslital at sunset.

### 4.3    Model performance

According to the different monthly climatologies presented, KENDA-1 is generally able to present accurate results regarding the evolution of studied variables even in complex topography. However, some phenomenon specific to mountainous regions and/or particular synoptic conditions can also lead to large modeling errors.

### 4.3.1    Temperature

The analysis of the daily cycle, averaged over the entire campaign shows that the majority of ground T differences with respect

to measurements of the nearby ground station lays between $\pm-3$ °C (Fig. 5) with a nighttime overestimation and a daytime underestimation by KENDA-1. Two one-year analysis over the COSMO-2 (Voudouri et al., 2018) and COSMO-1E (Voudouri et al., 2021) domains (Alpine Arc and particularly Switzerland and northern Italy) find a similar daily cycle in ground T mean error, but of reduced amplitude. COSMO-2 show a -0.8 °C bias during daytime and unbiased T during night while COSMO-1E has a -0.5 °C bias during day and a +0.5 °C bias during night. An additional study (not shown) performed on COSMO-2 (v.5.03)

show the same diurnal cycle in T difference over a full year. According to the complex topography around MER and the induced elevation bias, the modeled climatology of ground T is satisfactory, even if differences up to 8° C are found. The main explained source of ground T differences is caused by missed surface T inversion. The frequency of this phenomenon is partially missed by KENDA-1 from March to August (7.a) and its amplitude is underestimated for all month under study. It is especially the case at the end of March, when sharp inversions form due to enhanced night time radiative cooling and important global solar





radiation. The observed amplitude difference are mainly due to an underestimation of T at ground (Fig. 6 and 5). A work carried by Sekula et al. (2019) on the nonhydrostatic model CY40T1 AROME CMC (2km horizontal resolution) showed the same general overestimation of the minimum T in valleys bottom. The largest differences were measured during strong high-pressure systems which favors cold air pools formation leading to e.g. T overestimations of up + 7 to 9 °C during 10 days in March.

A preliminary analysis on KENDA-1 behaviour during this strong T inversions (not shown) show that the observed differences

are probably due to a too low model first guess ensemble spread. The model is too much trusted in the model-observation weighting scheme and measured T at MER are therefore not used in the model assimilation step. Another hypothesis is that a too large observation error is assigned to the station of MER (1.17K end of March). Additionally, at this period, the model predictive capacity concerning ground relative humidity is variable. During day, the observation and KENDA-1 data are within $\pm$ 5% relative humidity (RH) interval during day (not shown) but at night the model is heavily drier (-20 to -30 % RH). According to

Westerhuis et al. (2021), artifacts from the NWP are to be expected under conditions favourable to surface T-inversion. The COSMO-1E vertical coordinates follow the terrain. Therefore, over complex topography, T-inversions and the surface of the vertical grid used by the model intersects which can produce numerical artifacts. The systematic T underestimation during night can also be driven by errors of the model cloudiness. An overestimated model cloudiness could prevent the model surface to cool down due to too low out-going long-wave radiation. Further investigation have to be done using either the ceilometer or

the DWL to estimate the cloud cover over the Haslital valley and to compare with the model estimation. Finally, an ongoing turbulent mixing in the model can be present while in reality a cold pool might be formed and a full or partial decoupling from the flow above could also cause the observed differences with observations.

For profile comparison, MWR T is considered as the reference, but the MWR T reliability, especially at higher altitude, has to be taken into account in the evaluation of KENDA-1 results. Löhnert and Maier (2012) performed a MWR-RS comparison and

showed that random error inherent to the measurement principle can be important in some cases. Random errors range grows up to 1.7 K at 4 km height, due to a 95% influence from the used apriori profile. KENDA-1 and MWR T profiles differences are constrained to $\pm$ 1 °C for all altitudes between 1400 and 2200 m both day and night except in June and July (Fig. 6). Differences up to -3 °C can occur near the ground in winter or at ridge level in July. This near overall negative bias can be explained by several way. First, the MWR is susceptible of errors especially for higher altitudes with RMSE between 1 and 1.5 °C (Liu

et al., 2022). KENDA-1 T remains mainly inside the uncertainty. Moreover, the Meiringen radiometer has been trained with profiles from Payerne on the Swiss Plateau, so that the difference in altitude (+100 m) and in atmospheric conditions could induce a larger RMSE or even a bias in the MWR measurements. The direct influence of topography (Löhnert et al., 2021) can be discarded, since the instrument has been placed in order to have to obstale in the line of sight. Despite this uncertainties, the T differences up to -3 are probably a clear underestimation of KENDA-1 Ts. The hypothesis of of cloud amount overestimation,

mentioned for the problem of T inversions, can also explain this T profile bias.

### 4.3.2    Wind

The valley wind monthly climatology reveals a very good performance of the model. Up and down-valley wind are well modeled from March to July and, to a lesser extent, in November if compared to the observations. KENDA-1 is also able to get this



seasonal evolution of the vertical extent of the valley wind system. The onset of valley winds is however predicted with a larger
inaccuracy. KENDA-1 places the transition to up-valley wind too early after sunrise (Fig. 8 and 11). This 1-2 hours difference
with the observations is partially explained by the absence of surface T inversion in the model (section **??**), so the time allowing
an erosion of the stable layer is not taken into account.

The capability of COSMO models to estimate the diurnal along-valley winds in real valleys has been investigated by Schmidli
et al. (2018) for 3 summer weeks with weak synoptic forcing and intense solar heating. The model results are compared to
measurements at the MeteoSwiss ANETZ stations, the automatic monitoring network preceding the present-day SMN. They
showed that the wind diurnal cycle was well represented by COSMO1-E in large valleys such as the Rhine Valley at Chur
(base width of 3 km and width at half height of 8 km), and medium valleys (e.g. the Rhone Valley at Visp a with base width of
1 km and width at half height of 4 km). For smaller valleys, e.g. the Maggia Valley in Cevio (base width of 500 m, width at
half-height of 3 km), the valley wind amplitude was underestimated. Despite an underestimation of the maximal valley wind
speed, the onset of up and down valley winds was correctly modeled. The results of the modeled wind speed and direction at
MEE are comparable to the analysis in Visp (Fig. 11). However, the onset of up and down valley winds is less well modeled in
the Haslital. The valleys at both sites have a similar cross-sections but the Rhone valley is four time longer than the Haslital.

The differences between KENDA-1 and the observed cross-valley wind climatology (Fig. 13) can be interpreted as a too
strong influence of the Sarneraatal thermal winds or as an effect of the grid cell overlap on the north-facing slope. The presence
of strong down slope winds at the Brünig pass may have a direct influence on the along valley wind diurnal cycle. In a more
recent study in the Rhone valley at Sion, Schmidli and Quimbayo-Duarte (2023) reports an inadequate representation of the
morning wind reversal by COSMO-1E whereas the evening transition is correctly modeled. They showed particularly poor
performance in along valley wind simulation for certain days in the Rhone valley. The study focuses on results above Sion and
on the interaction of along and cross valley flows. Like in the Haslital (Fig. 13), too strong modeled cross-valley wind reaching
the valley floor interrupt the formation of the up-valley flows. At Sion, the cross-valley flow is restricted to upper levels so that
the stronger lower valley atmosphere stratification protects the up-valley flow.

According to (Schmidli et al., 2018), the key factor is not the resolution of the grid. The horizontal resolution required for a
good along valley wind representation is moderate. As long as the resolution of the valley base cross section is 1 or 2 grid points
width, the results obtained are good in most cases. A more important feature is the altitude bias of the model at the ground. For
the MER station, the width of the valley can contain 1.5 grid cells (Fig. 1) but the fact that no cell is superimposed on the valley
floor only leads to this disfavouring altitude bias. Indeed, the two cells overlapping on the valley floor also overlap part of the
steep ascending slope. Concerning the problem of stratification that favors the influence of cross valley winds, surface moisture
is a key factor. Simulations performed by Schmidli and Quimbayo-Duarte (2023) show that a 30% increased soil moisture
relative to KENDA-1 data leads to better along valley wind modeling. Even though stronger smoothing of the topography
improves the stratus cloud simulations, it also decrease the quality of forecasts of valley winds and orographically induced
convection (Westerhuis et al., 2021).

Finally, despite the fact that KENDA-1 proposes good climatologies, the case-by-case analysis shows important differences
with measurements, whether they are particular events or not. Non-systematic differences are observed in most profiles. Even



thought these differences show regular patterns in the case of foehn or valley winds, it is common that unpredictable behavior
affects the model (Fig. J1, H1).

## 5 Conclusion

The analysis of the data from the MER campaign, in the Haslital, yields valuable information on the climatology of wind profiles and T-profiles. The MWR and DWL installed, as well as the nearby SMN station, allowed to determine the particularities of this valley between November 2021 and July 2022. This extensive measurement campaign provided data that was not available 765 in this region and that are rather sparse in mountainous regions. The main conclusions that can be drawn from the Meiringen campaign are the following:

- Nighttime T inversions are common during all the months under study with bigger amplitudes during December and January. They persist during daytime from November to February.

- The valley wind system is distinguishable in the monthly aggregated values for all the months under study except in 770 December and January. In addition, this thermal wind system can be influenced by external factors such as synoptic wind intrusions or perturbation from from adjacent valleys wind system. Here are the peculiar characteristics observed:

  - This diurnal flow patterns develop in a more distinct way for the summer months (June to August). The vertical extent of down-valley winds after sunset increases from March to August: from 1000 m.a.g.l. to 1600 m.a.g.l. respectively.

  - The periods of transition between up and down valley wind are related to sunrise and sunset. The morning transition on the ground is delayed by about 3-4 hours compared to sunrise and is near simultaneous for the rest of the profile.

  - The evening transition to down valley winds happens on a shorter time scale both at ground and in the DWL profile. The onset of down valley winds happens less than an hour before sunset and propagtes from ground to ridge height.

  - The influence of the tributary valley (Sarneraatal) on the cycle transitions is important below 600 m.a.g.l. even 780 during calm clear days. These strong winds can easily exceed the Haslital along up valley winds.

  - A cross valley circulation is measured around sunset (19:00-20:00). A separation between north and south facing wind lays between 700 and 1000 m.a.g.l. which suggests that flow pattern form a closed circulation cell. This mechanism is influenced by the strong wind from the Sarneraatal.

  - Foehn is able to rapidly (<1 hr) destroy the T inversion if present. The delay between the T increase due to the 785 foehn breakthrough can be of 1-2 hr between the SMN/MER ground measurement and the MWR/MEE lowest level. Maximum wind speeds are homogeneous between ground and 1500 m.a.g.l. which corresponds to mean ridge's height (valley channeling).



In parallel to these observations, the data of two cells of the KENDA-1 assimilation model has been analyzed. The results in MER and MEE show similar T profiles. At MEE, the classical valley wind pattern is significantly affected by the tributary valley of the Sarneraatal. Additionally, during foehn events, wind speeds at MER are almost two times higher than at MEE.

The comparison with observations show that KENDA-1 was able to simulate climatologies in close agreement with those obtained from REM instruments. T profiles climatologies mostly show T difference <1 °C. Moreover directions and speeds of the diurnal valley winds are in agreement with the measurements. The vertical extent of the thermal winds, the onset time of down valley winds and the interaction with synoptic winds are also appropriately modeled. However, some phenomena are not properly captured by the model in assimilation mode:

- The frequency of occurrence and the amplitude of the surface T inversion are both underestimated in the T profiles of KENDA-1. This results in a systematic overestimation of the ground T during the presence of surface based inversions. In extreme cases it reaches up to 9 °C. Moreover, the discrepancies between the model first guess prevents the SMN station of MER to be assimilated.

- A T underestimation of -2 to -3 °C under 1500 m, more frequent during nighttime. In July, it is also seen between 2000 and 2500 m during the full daily cycle, except in the early morning.

- KENDA-1 shows a too early onset of up-valley winds due to the absence of the near surface stable layer caused by the nighttime inversion. The onset of up-valley winds between the ground and 1200 m occurs therefore 1-2 hours earlier than measured.

- Contrarily to climatologies, the analysis of single profiles show important differences with measurements. This is particularly true during foehn events with a systematic cold bias, a delayed time of foehn breakthrough and the associated overestimated wind speeds.

  - In the case of southern foehn, a near systematic ground T underestimation by both MWR (-0.2 to -1.5 °C) and KENDA-1 (-2 to -4 °C). Concerning the T profiles, KENDA-1 show a negative T bias between 2.5 and 4°C from the foehn breakthrough until maximal wind speed is reached, the difference then gradually decreases.

  - Wind speeds simulation during foehn show significant difference over MEE and MER: the KENDA-1 MEE profile show a good match up to 1000 m.a.g.l. whereas the winds over MER are reported to be twice as high (120 km/h).

The results obtained in this study allowed to deepen consensual knowledge about atmospheric phenomena in complex topography and to identify processes specific to the studied valley. Complex interactions between the Haslital and the tributary valley of the Sarneraatal have been observed and could explain some differences observed with the literature. However, many observed phenomena are not yet satisfactorily observed and modeled and require further investigation. A better understanding of the exchange processes in complex topography and the ability of the model to take them into account is an essential condition to improve the prediction capacity of NWP in complex topography.



*Data availability.* Data are available on request

*Author contributions.* AB did the analysis, AB and MCC prepared the manuscript. MH and SM operated the instruments during the campaign. DL and MA provided the model data. All co-authors contributed to the manuscript online.

*Competing interests.* The authors declare that they have no conflict of interest.

*Acknowledgement.* This work was supported by the the Swiss Federal Office for Meteorology and Climatology



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





## Appendix A: Weather conditions

| | Temp. [°C] | | Precip. [mm] | | % max sunshine | |
|---|---|---|---|---|---|---|
| | *Abs.* | $\Delta T_{norm}$ | *Abs.* | *% Norm* | *Abs.* | *% Norm* |
| **Nov** | 3.1 | -0.6 | 57.8 | 61.03 | 38 | 90 |
| **Dec** | -1.5 | -1 | 119 | 121.93 | 30 | 90 |
| **Jan** | -2 | -0.9 | 35 | 42.42 | 54 | 130 |
| **Feb** | 2.2 | +1.8 | 90.5 | 125.52 | 54 | 120 |
| **Mar** | 6.4 | +1.4 | 17.6 | 20.71 | 70 | 150 |
| **Apr** | 8.8 | -0.4 | 74.4 | 82.85 | 54 | 110 |
| **May** | 15.3 | +2.1 | 47.2 | 34.43 | 48 | 110 |
| **Jun** | 18.7 | +2.2 | 142 | 95.95 | 53 | 110 |
| **Jul** | 20.4 | +2.5 | 72.5 | 45.26 | 62 | 130 |
| **Aug** | 19.7 | +2.3 | 89.7 | 51.49 | 61 | 120 |

**Table A1.** Absolute values and differences/ratios to the climatological norm (1991-2020) for monthly temperature, total precipitation and proportion of max. sunshine duration in the SMN station from MER.

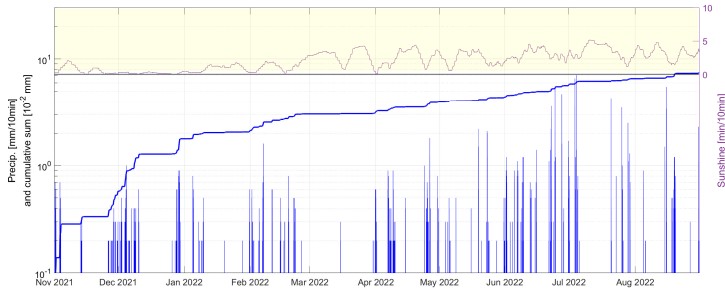

**Figure A1.** Timeseries of precipitation intensity [mm/10mm] (blue bar), cumulative precipitation [mm] (blue line) and mean sunshine duration [min/10min] (5 days moving average)




**Appendix B: Geographical description**

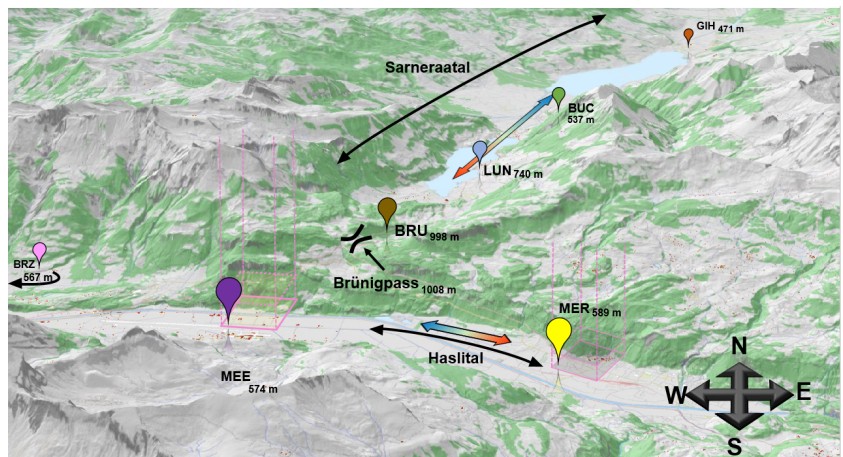

**Figure B1.** Map of the geographical situation in the lower Haslital (E-W and the Sarneraatal (S-N). The automatic measurement from the SMN are: Meiringen (MER), Brienz (BRZ) and Giswil (GIH). Stations from the FEDRO are also depicted: Brünig (BRU), Lungern (LUN) and Buchholzbrücke (BUC). The last represented site is the campaign site in Unterbach (MEE). The map was download from swisstopo (/https://map.geo.admin.ch, last access: 19.09.2023)

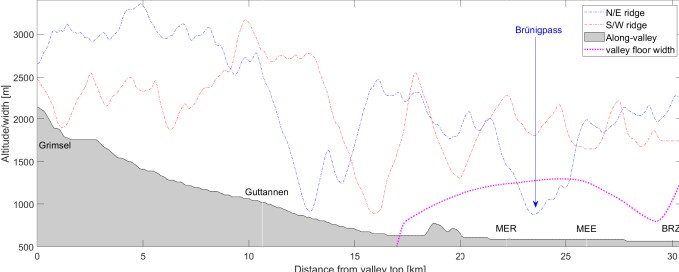
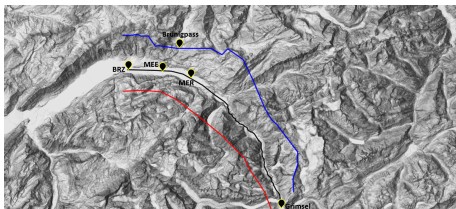

**Figure B2.** Along-valley altitude variation for the two crests (red/blue) and valley floor (shadowed). Valley floor width is depicted with the dotted pink line. The abscissa indicates the distance from the top of the valley (Grimselpass) following the valley lowest point. The map was download from swisstopo (/https://map.geo.admin.ch, last access: 19.09.2023)



**Appendix C: MWR vertical resolution**

| Range [m.a.g.l.] | Vertical resolution [m] | RMSE [°C] |
|---|:---:|:---:|
| 0-250 | 50 | 0.25 |
| 250-500 | 75 | 0.25 |
| 500-800 | 100 | 0.50 |
| 800-1200 | 150 | 0.50 |
| 1200-1600 | 150 | 0.75 |
| 1600-2200 | 200 | 1.00 |
| 2200-3000 | 300 | 1.00 |

**Table C1.** Vertical resolution and RMS error (given by the manufacturer) of the MWR temperature according to its different ranges.





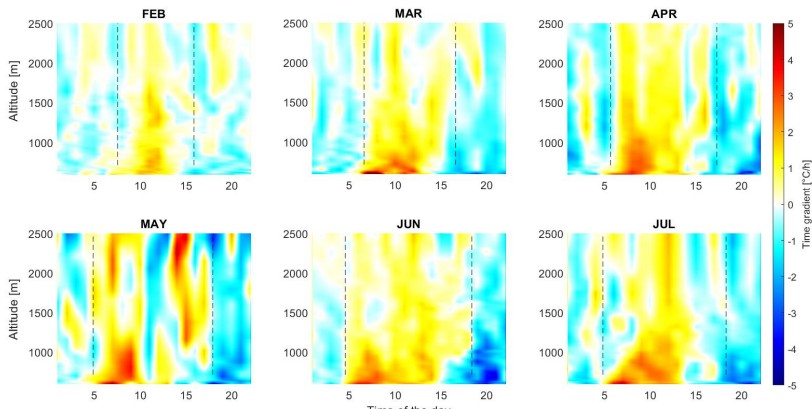

**Figure D1.** MWR temporal T gradient climatology [°C/hr]. Sunset and sunrise are depicted with the dotted lines.

**Appendix D: T gradient climatology**





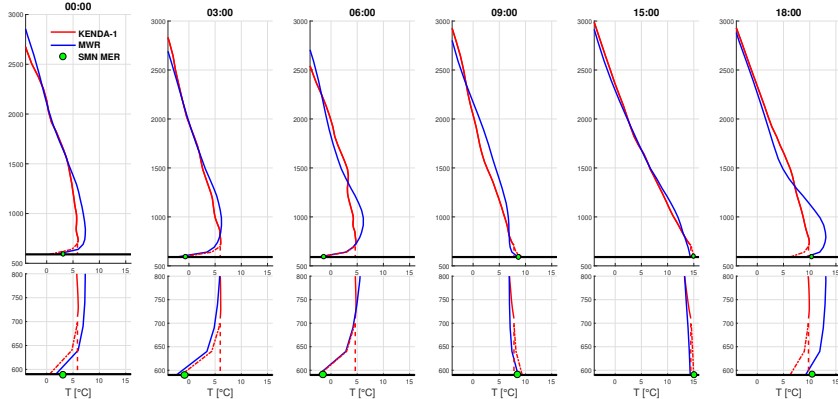

**Figure E1.** T-profile from MWR (blue) and KENDA-1 (red) for the 22.03.2022. The dashed line indicates the raw value of KENDA-1 as it is used in the ground T comparison. The dash-dotted line extends the profile to the ground with the same gradient as the MWR measurement.

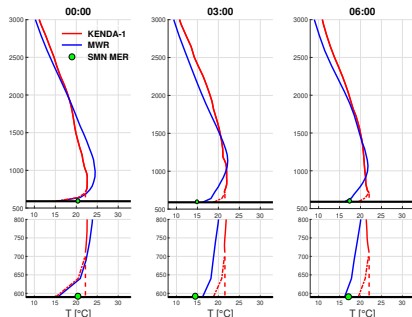

**Figure E2.** T-profile from MWR (blue) and KENDA-1 (red) for the 19.07.2022. The dashed line indicates the raw value of KENDA-1 as it is used in the ground T comparison. The dash-dotted line extends the profile to the ground with the same gradient as the MWR measurement..

**Appendix E: T-inversion at the end of March**



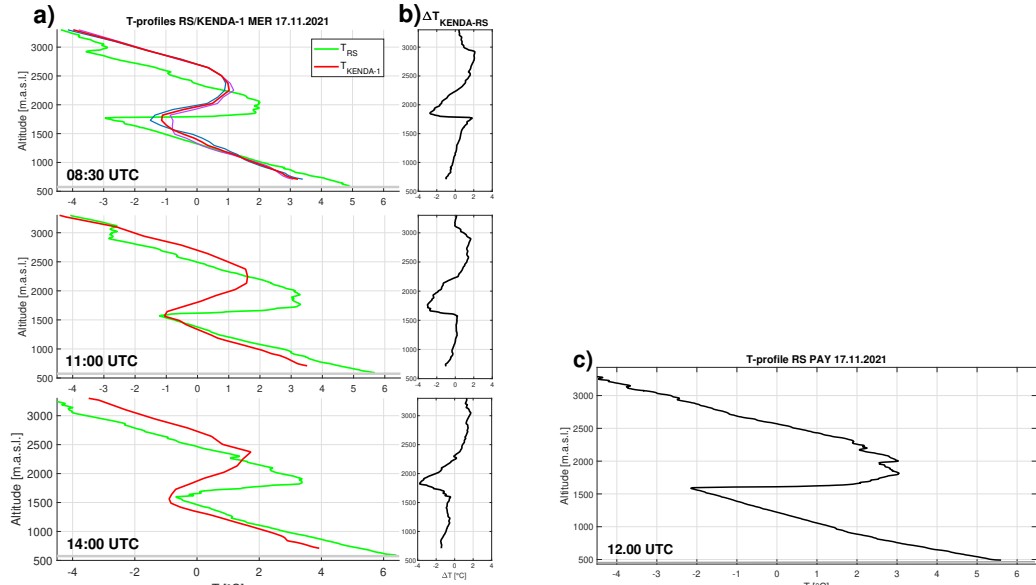

**Figure F1.** a) T-profiles of RS (in green) and KENDA-1 (in red) at MER during the day of 17.11.2022 at 08:30, 11:00 and 14:00. For the top plot, the T-profile of KENDA-1 at 08:00, 09:00 and their average are plotted in purple, blue and red, respectively. b) shows the differences between KENDA-1 and the RS. The height of the 3 main T inversions is also reported by dashed lines: bottom inversion at 08:30 (1760 m) and 11:00/14:00 (1560 m) and finally the upper inversion from 08:30 at 2920 m. c) T-profiles of RS in PAY at 12:00

**Appendix F: RS 17.11.2021**



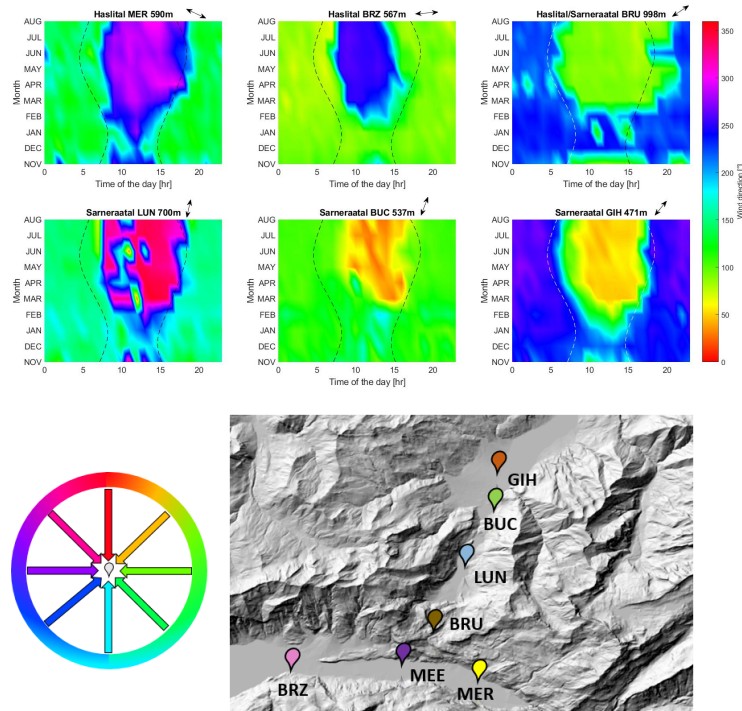

**Figure G1.** Monthly median wind direction [°] at ground level for the different automatic measurement station of the Haslital (top row) and Sarneraatal (bottom row). The color code is presented on the color wheel. The map was download from swisstopo (/https://map.geo.admin.ch, last access: 19.09.2023)

## 950 Appendix G: Wind pattern at SMN stations

All these stations show valley wind patterns from March to August and in November at some stations (MER, BRU, LUN and GIH). The up valley daily period at BRZ und BUC is less extended than few kilometers upstream in MER and LUN/GIH, respectively. This reduced time extents are probably due to the location of these stations near the slopes and not in the center of the valleys. The vicinity of the lake ($<$ 100 m) can also cause this difference at the BRZ station due to its higher thermal inertia.





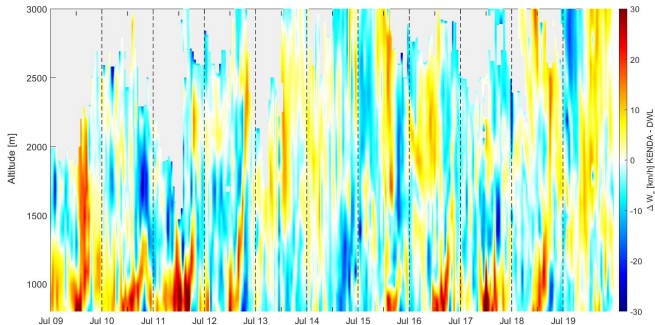

**Figure H1.** Wind speed profiles comparison [km/h] timeseries between KENDA-1/MEE and DWL/MEE during a selection of 10 clear days middle of July (July 15th is overcast). Sunshine and sunset are represented by the dashed line.

**Appendix H: Valley winds during clear days**




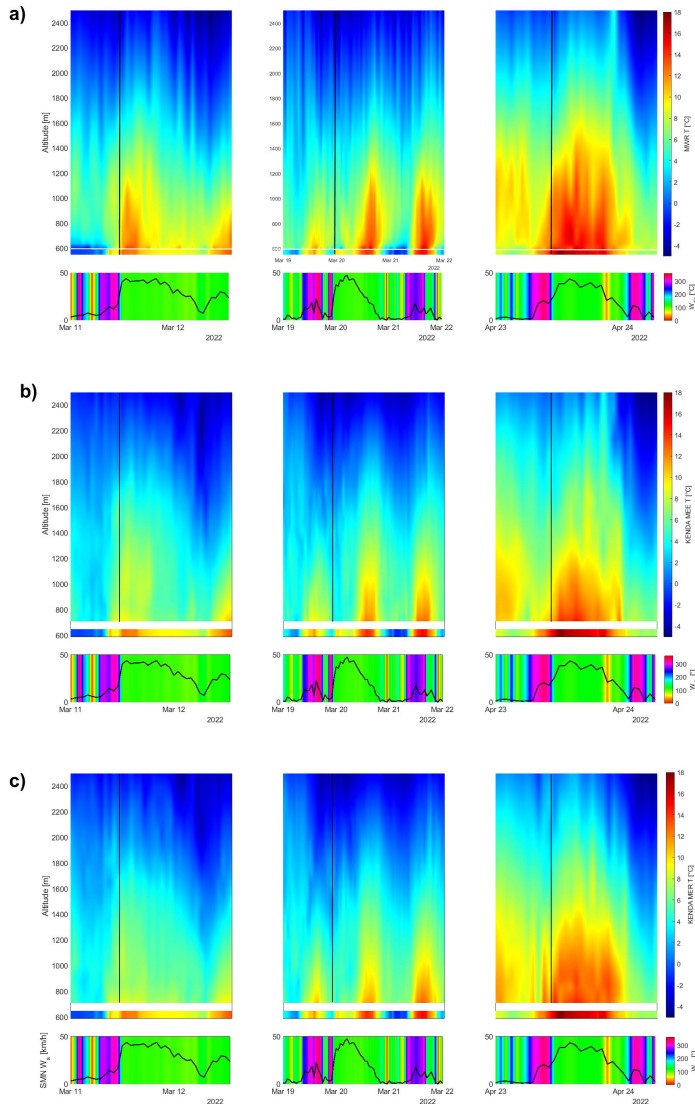

**Figure I1.** T profiles [°C] timeserie from a) MWR b) KENDA-1 MEE and c) KENDA-1 MER. during a selection of 3 foehn events during the campaign: left 11-12.03.2022, middle 19-22.03.2022 and right 23.04.2022. T, wind speed [km/h] and direction [°] from the SMN MER are given in the lower part of each figure. The solid line represent the begining of the foehn event.

## Appendix I: T comparison during foehn

The effect of foehn on T is variable: during March 20, the foehn event started at 23:00 and inhibited the formation of the T inversion by destroying the stable layer built so far. This induced mild T during the night compared to previous days (I, Fig. I1.a). On the other hand, for the event of March 11 and April 23, the foehn events start around 11:00, and the T increase is not as



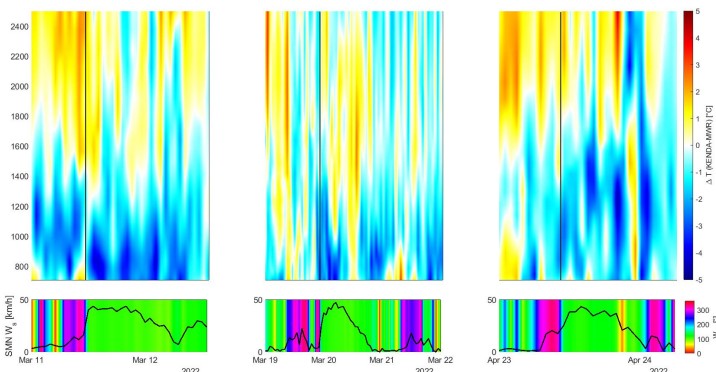

**Figure I2.** T profiles comparison [°C] between KENDA-1/MEE and MWR/MEE during a selection of 3 foehn events during the campaign: left 11-12.03.2022, middle 19-22.03.2022 and right 23-24.04.2022. Wind speed [km/h] and direction [°] from the SMN MER are given in the lower part of each figure. The solid line represent the begining of the foehn event.

important. For March 11, the foehn breakthrough happens at 11:00 with a T increase of 8 °C between 08:00 and 09:00 and a second T increase of 3 °C between 11:00 and 12:00. The MWR measures the same T evolution. With a delay of less than an hour, KENDA at MEE and MER (Fig. I1.b-c) show the same temporal T evolution as at SMN. However, the temporal T gradients are smaller because of the missed inversion of the previous night and the underestimated T during the foehn event. The two modeled T evolution exhibits a delay of one hour. KENDA-1/MER T increase happens before that from MEE which

is coherent with the orientation of the Haslital and the provenance of foehn. For the rest of the profile, the rapid T increase is observed between the ground and 1200 m and nearly constant T are measured in that part of the profile. KENDA-1 shows the same pattern with a negative T bias of 2 °C that grows up to 4°C when maximal wind speed is reached. For March 20, at ground, the foehn breakthrough at 11:00 breaks the inversion around midnight and maintains mild T until the arrival of the sun where the T further rise according to a classical daily cycle. KENDA-1 ground T at MER and MEE does not show any T increase during

the night and therefore underestimates the T. During daytime, the T underestimation is the same as for days without foehn. For the rest of the profile, the T are better modeled than for the 11.03. The T underestimation (2-3 °C) again extends from ground to 1200 m but this time it remains only in the first 4 hr of the event. Afterward, this underestimation is constrained under 900 m. Finally for the 24th of April, at ground, MWR and SMN T are the same until the foehn breakthrough at 11:00. The two measurements then show a 2 °C difference during the foehn event (wind speeds > 25km/h). KENDA MEE and MER are similar

to the MWR measurements. For the rest of the profile, the model is again underestimating the T but higher up, between 1200 and 1400 m.

**Appendix J: Wind speed and direction comparison during foehn**

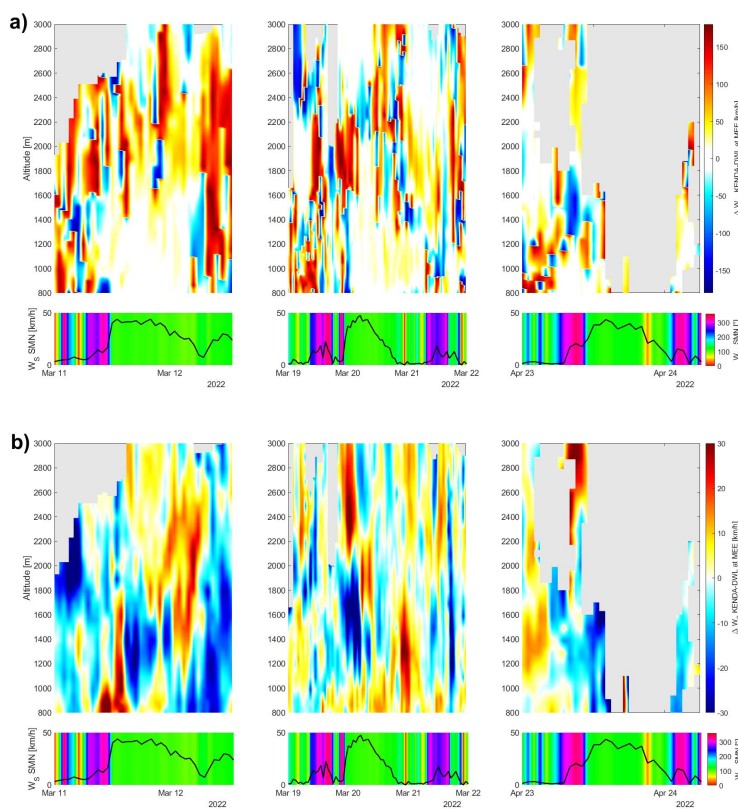

**Figure J1.** a)Wind direction comparison [°] and b) wind speed profiles comparison [km/h] timeseries between KENDA-1 MEE and DWL during a selection of 3 foehn events during the campaign: left 11-12.03.2022, middle 19-22.03.2022 and right 23-24.04.2022. Wind speed [km/h] and direction [°] from the SMN MER are given in the lower part of each figure.