# Peer review of "Comparison of temperature and wind between ground-based remote sensing observations and numerical weather prediction model profiles in alpine complex topography: the Meiringen campaign"

_EGUsphere, 2023_

## Referee Comment (RC2)

Review of egusphere-2023-1961

*'Comparison between ground-based remote sensing observations and NWP model profiles in complex topography: the Meiringen campaign"*

by Alexandre Bugnard et al.

Paper in review for Atmospheric Measurement Techniques

**Summary**

The manuscript presents a comparison of campaign observations from the Meiringen Campaign in a narrow Swiss Alpine valley with the high-resolution 1-km KENDA analysis. The comparison focuses on temperature and wind profiles measured by a microwave radiometer and Doppler wind lidar for ten months during 2021/2022. It is shown that observed and modeled seasonal climatologies of temperature and wind profiles agree well, although for specific situations, such as for example temperature inversions or foehn events the differences are relatively large. The manuscript also links the complex topography to thermal wind system and presents cross- and along valley flow systems observed during the campaign period.

The manuscript presents valuable observations from an Alpine site and provides new insights in the quality of the high-resolution analysis in complex terrain and shows examples of how specific terrain-induced flow features can influence the differences between observations and the analysis. In general, I support publication of the manuscript, but I have several comments and questions to the authors that should be addressed prior to publication and which are primarily related to the general state of the manuscript, the selection of results and the storyline.

**1   General comments**

1 General state of the manuscript

Generally, the research results in the manuscript are well presented, however, the overall structure as well as the overall "state" of the manuscript should be improved. For example, (i) citations are frequently not correctly used (e.g., double brackets, missing references, incorrect citation style), (ii) references to Sections and Figures are frequently missing, (iii) the presentation of Figures could be improved, and (iv) the Appendix should be substantially shortened to only include the additional information that is absolutely necessary for the manuscript (see also General Comment 4). Moreover, please check if all abbreviations are correctly introduced when first mentioned (e.g. "T" for temperature is not introduced, l. 31, p. 2). Thus, while I find the content of the manuscript interesting, the manuscript needs further polishing prior to publication.

2 Structure

The overall structure of the manuscript with introduction, methods, results, etc. is good, however, in my opinion the Results Section is missing a coherent storyline. I would suggest to re-structure and streamline this section with a strong focus on relevant synoptic features and important differences/agreements between observations and the analysis. Some specific suggestions for potential improvements are listed below. Generally, the manuscript is (unnecessarily) long (in total more than 50 pages), and focusing on a coherent storyline will likely help to shorten the manuscript and convey the key results in a concise way.

3 Figures
Many figures contain a large number of panels and show the results from the analysis and the observations. In my opinion, the figures should be optimized (i) by minimizing white space between panels and (ii) by showing the result from the analysis or observation and directly the difference between both as sub-panels. This would help to combine the synoptic conditions and associated errors and remove redundancies when analysis and observations are very similar. It also emphasizes differences between observations and analysis. The manuscript includes a relatively large number of figures. I believe that the figure number could be reduced by carefully selecting the relevant ones and combining figures.

4 Appendix
The appendix includes too many figures. I would ask the authors to carefully select only those of primary relevance for the manuscript. Moreover, similarly to General Comment 3, the information content of many figures can probably be condensed to fewer figures. E.g., Figures 1, B1, B2, and G1 all show a map of the measurements sites. I suggest their content can be summarized in 1-2 panels.

5 Consistency
I would ask the authors to double-check the consistency of used abbreviations and naming conventions. E.g., it is explicitly stated that data are presented with instrumentation/site, however, this is often not consistently applied (in particular in Results Section). Moreover, several different data sets and sites are compared with each other. When overestimations / underestimations are mentioned, please check that it is specified which data/site are compared.

**2 Specific comments**

1. **Introduction**
   The introduction is well written, cites relevant literature, and the goals of the study are clearly presented. As a minor adjustment, I would suggest to remove the sub-sections in the introduction.

2. **Methods**
   In the results, bias and errors of the analysis comparison to MWR are shown. How strongly does this result depend on the quality of trained retrieval algorithm? Is it possible that the MWR measurements itself are biased? I would appreciate if the authors could comment on the error

magnitude of MWR-retrieved temperature profiles and relate this to the shown bias and error magnitudes compared to the analysis.

3. I appreciate the 3D map (Fig. 1), however, it would suggest to use the "northing". Moreover, it is very similar to Fig. B1, although B1 contains some added useful information. I would suggest to replace Fig. 1 by Fig. B1, and also include some information from Fig. G1 (specifically, I would find it a lot easier if wind arrows would depict the median wind direction instead of coloring the stations accordingly).

4. The different instrumentation and sites are well described. Due to different durations of employment I would appreciate an overview table of instrumentation, available data, sites, and the measurement period.

5. l. 86: COSMO-1E: Please introduce this abbreviation.

6. l. 136 "Five km before the lake": I would suggest to write "Five kilometers".

7. l. 186: "perpendicular to the valley (not used in this study)": To streamline the manuscript I would suggest to only describe the measurement setup that was actually applied during the campaign.

8. l. 197: "These precipitations arrived in form of snow": I'm not sure if precipitation is commonly used in plural.

9. Section 2.4
I appreciate a description of the weather during the campaign, although I do not fully understand why the authors focus on precipitation, snow, and sunshine duration while the focus of the study is on wind and temperature profiles and circulation features. I think this section could more strongly focus on the relevant aspects for the analysis.

10. Section 2.5
I believe that this section is not necessary as a separate section, but the information should be included in the previous paragraphs, e.g., where the sites, instrumentation, KENDA, etc. are described, respectively.

11. l. 590: "Therefore, this study does not allow to make prediction of model performance for forecasting." Can the authors please elaborate on this, I do not fully follow the reasoning and relation to forecasts here.

12. **Results**
I would suggest to streamline the Results Section (see also General Comments above).

13. I would suggest to use more informative titles in the Results Section (e.g. "3.1.1 Climatology" is only based on observations, which cannot be deduced from the title).

14. The overall section structure could be improved. It is not very intuitive to show (i) temperature, (ii) wind, and (iii) Foehn (with wind and temperature), while other atmospheric features were explicitly discussed in (i) and (ii). I would suggest to define a storyline to follow in the Results Section and focus on the key results.

15. Generally I would be careful with the word "climatology" as here only a few months and not several years of data are analysed.

16. Section 3.1.1
I would suggest to streamline this section and focus on relevant features. E.g., the presence of a diurnal cycle and temperature increase after sunset are expected features and their description could be streamlined.

17. l. 230f: Please correct units: temporal gradient of 5C/?.

18. l. 235: I struggle to see the daytime temperature inversions. Could these features be outlined in the figures (e.g., through contours)?

19. Section 3.1.2
Which differences are analysed in this section? Is it hourly values?

20. l. 255 f: "The difference in the effect of the ELR correction" Which difference? At both stations RMSE increases? Please clarify.

21. Section 3.1.3
Would it make sense to place this sub-section before Section 3.1.2 as profiles have already been described Section 3.1.1 (Fig. 6 fits better to Fig. 3)? Is Fig. 3b required? If I'm not mistaken it is not referenced in the text. The comparison of temperature profiles and respective KENDA biases assumes the MWR retrieval does not include a bias. Given some uncertainty in the retrieval algorithm, could the authors comment on this issue (see also comment above)?

22. Section 3.1.4
l. 308 f: "All this leads to both an important overestimation of the T at ground level (Fig. 5) and a slight underestimation of the T just above the T inversion (Fig. 6)." Both figures compare different data sets, i.e. a direct comparison of temperature differences at different heights is difficult. Moreover, I struggle to see temperature overestimations at the lowest level in MWR-KENDA comparisons in Fig. 5. Can the authors please clarify?

23. l. 309-311: Please either elaborate on this or remove.

24. l. 315 - 322 I find this very interesting and would like to see some results, as this observation rejection is linked to some of the surface temperature differences reported in the study.

25. Section 3.1.5
Personally, this very short sub-section interrupts the storyline which strongly focuses on MWR and surface measurements. Did the authors also compare MWR profiles with the RS profile? Please note that Fig. F1 is not referenced. Please also elaborate on the influence of the RS from Payerne. Was the additional RS/MEE not assimilated? Please double-check the spelling of radiosounding.

26. Section 3.2.1
This section appears unfinished and I think it needs some more work. The writing style with bullet points differs substantially from the style used above for temperature. Moreover, I would ask the authors to improve

Fig. 8. It is very difficult to identify relevant features in a 60 panel figure. Would it be an option to, e.g., show differences in panel b)?

27. Section 3.2.2
This section should be removed.

28. Section 3.2.3
I appreciate the comparison of allong valley winds during the campaign period to the 10-year climatology, however, this interrupts the storyline. In my opinion, it would be sufficient to mention the good agreement and move the figure to the appendix.

29. Fig. 10a: Typo in axis label: "DWL/MER"

30. Section 3.2.4
This section provides a very detailed description of the circulation evolution at different sites. I would ask as the authors to streamline this section and highlight the important circulation features. Figure B1 could also be referred to for clarification.

31. Section 3.2.5
l. 493-495 Could the authors please elaborate on this? How was the vertical velocity estimated?

32. Section 3.3
General comment: In my opinion, much of the comparison between KENDA and observations has already been described above. I would suggest to restructure to avoid repetion and potentially include foehn events as subsections in 3.1 and 3.2.

33. l. 505: Please define the foehn index and provide a reference.

34. l. 505: Is the subsequent analysis (e.g., Fig. 14) performed for three events only or for "all the period" with foehn? Please clarify.

35. l. 506: Which April episode? Fig. 15 shows again different time periods? Please clarify.

36. l. 516: I would expect a better agreement of KENDA and observations if both data are taken from the same site (compared to different sites). I would appreciate if the authors could explain the reason for comparison of KENDA/MEE with SMS/MER (instead of KENDA/MEE). Such comparison are frequently performed throughout the manuscript, and to some extent it is difficult to follow all different comparisons.

37. l. 521: "mettre reference fig ap 26"?

38. l. 537: Does KENDA/MER really show a delay in foehn onset? To me it rather looks like a too early onset (similar to KENDA/MEE)?

39. l. 541-544: In particular in this section it is very difficult to see where wind speed is over-/unterestimated, i.e., showing differences to DWL might be helpful.

40. l. 545: Based on Fig. 15 (which is the main figure discussed in this paragraph), I find it difficult to follow the conclusion that the representation at MER is better than MEE. Improved visualization may help.

41. l. 554: Please elaborate on the link between wind bias, temperature bias and specific humidity bias.

42. Fig. 14: Please specify what is shown in the figure. What do the numbers on top represent? What is shown in the x-axis in b)?

43. Fig. 15: Are the dates correctly shown in all panels? I cannot find any figures for 19-22 March nor 23-24 April. Are data shown only at 11 am and 11 pm?

44. **Discussion**
The manuscript includes an extensive discussion of the results, which I appreciate. However, I would suggest to condense the information and streamline the Discussion Section. It may also be helpful to include a short summary and/or discussion after the respective Results Sections, respectively.

45. l. 680: The daily cycle of temperature underestimation/overestimation is not apparent in the MER observations (Fig. 6). Could the authors please elaborate on this?

46. l. 681 f: Comparisons are made to different versions of COSMO. Please either specify the versions /differences or remove.

47. **Conclusions**
The Conclusions provides a bullet point summary of the key results. I would suggest to formulate continuous text for the conclusions with distinct paragraphs instead of bullet points.

48. l. 772 - 787. The important circulation features are listed here. I would appreciate a figure/sketch similar to Fig. B1 where all the identified flow features are outlined and summarized.

---

## Author Comment (AC1)

**Answers to the reviewer 2 comments on**

**"Comparison of temperature and wind between ground-based remote sensing observations and NWP model profiles in complex topography: the Meiringen campaign"**

First of all, we would like to thank the reviewer for the valuable, in-depth comments to our manuscript. We would also like to apologize for the numerous typos, wrong links to figures and not complete references that have sorrily lengthened the review process. As supposed, the manuscript was written very rapidly and the latex implementation to the AMT formal leads to unexpected problems. Second, the appendix was really designed as a supplement but just not submitted in a separate file. The revised version produces now two distinct files. Finally, according to both reviewers' request, the manuscript was largely shortened (35 pages instead of 42) and contains now only 12 figures.

The answers to the comments and questions are written in italic thereafter. When modifications of the manuscript are cited, the numbering of the figures correspond to the one of the new manuscript. The explanations themselves cite the numbering of the figures in the submitted manuscript in accordance to the lines' numbers of the comments.
* * *
—

1 General comments

 General state of the manuscript

Generally, the research results in the manuscript are well presented, however, the overall structure as well as the overall "state" of the manuscript should be improved. For example, (i) citations are frequently not correctly used (e.g., double brackets, missing references, incorrect citation style), (ii) references to Sections and Figures are frequently missing, (iii) the presentation of Figures could be improved, and (iv) the Appendix should be substantially shortened to only include the additional information that is absolutely necessary for the manuscript (see also General Comment 4). Moreover, please check if all abbreviations are correctly introduced when first mentioned (e.g. "T" for temperature is not introduced, l. 31, p. 2). Thus, while I find the content of the manuscript interesting, the manuscript needs further polishing prior to publication.

_We apologize for the numerous typos. Citations are now correctly done, abbreviations and references to sections and figures were checked. Figures were improved, the appendix were transformed to a supplement and both the manuscript and the supplement were shortened._

2  Structure

1 The overall structure of the manuscript with introduction, methods, results, etc. is good, however, in my opinion the Results Section is missing a coherent storyline. I would suggest to re-structure and

streamline this section with a strong focus on relevant synoptic features and important differences/agreements between observations and the analysis. Some specific suggestions for potential improvements are listed below. Generally, the manuscript is (unnecessarily) long (in total more than 50 pages), and focusing on a coherent storyline will likely help to shorten the manuscript and convey the key results in a concise way.

*The structure of the manuscript was revised and modified to improve the streamline and to shorten the paper .*

3 Figures

Many figures contain a large number of panels and show the results from the analysis and the observations. In my opinion, the figures should be optimized (i) by minimizing white space between panels and (ii) by showing the result from the analysis or observation and directly the difference between both as sub-panels. This would help to combine the synoptic conditions and associated errors and remove redundancies when analysis and observations are very similar. It also emphasizes differences between observations and analysis. The manuscript includes a relatively large number of figures. I believe that the figure number could be reduced by carefully selecting the relevant ones and combining figures.

*Figures were improved and their number in the manuscript was reduced to 12 and some of them were moved to the supplement.*

*Figures of the differences between the observation and the model are useful for the T and wind speed analysis (see Fig. 6). They are however more difficult to interpret in the case of monthly medians of wind direction. Moreover, the analysis of the wind direction differences between MEE and MER is complex due to the bending of the valley between both sites. We estimated then that the present figures are more adapted to the wind analysis.*

4 Appendix

The appendix includes too many figures. I would ask the authors to carefully select only those of primary relevance for the manuscript. Moreover, similarly to General Comment 3, the information content of many figures can probably be condensed to fewer figures. E.g., Figures 1, B1, B2, and G1 all show a map of the measurements sites. I suggest their content can be summarized in 1-2 panels.

 *Some figures of the supplements were also removed or condensed (e.g. Fig. 1, B1, B2 and G1 about the topography) . Other figures were moved from the paper to the supplements.*

5 Consistency

I would ask the authors to double-check the consistency of used abbreviations and naming conventions. E.g., it is explicitly stated that data are presented with instrumentation/site, however, this is often not consistently applied (in particular in Results Section). Moreover, several different data sets and sites are compared with each other. When overestimations / underestimations are mentioned, please check that it is specified which data/site are compared.

*A scrupulously proofreading of the second version of the manuscript hopefully corrected all the mentioned typos, including also reference to figures and sections and full and complete references.*

2 Specific comments

1. Introduction

The introduction is well written, cites relevant literature, and the goals of the study are clearly presented. As a minor adjustment, I would suggest to remove the sub-sections in the introduction.

*The subsections were removed and, as requested by the first referee, the introduction was shortened without removing important notions relevant for the study.*

2. Methods

In the results, bias and errors of the analysis comparison to MWR are shown. How strongly does this result depend on the quality of trained retrieval algorithm? Is it possible that the MWR measurements itself are biased? I would appreciate if the authors could comment on the error magnitude of MWR-retrieved temperature profiles and relate this to the shown bias and error magnitudes compared to the analysis.

*The MWR bias and errors and their potential effects on the comparison between the model and the observations are already described in sections 2.3.2, 3.1.2 and 4.3.1. We concluded that (sect 4.3.1) "The near overall negative bias can mainly be explained by two factors: first, the MWR is susceptible of errors especially for higher altitudes with RMSE between 1 and 1.5 °C (Liu et al., 2022) and, second, the 620 MWR/MEE has been trained with profiles from Payerne, so that the difference in altitude between both stations (+100 m) and in atmospheric conditions could induce a larger RMSE or even a bias in the MWR measurements. Despite these uncertainties, the T differences up to -3 °C are probably a clear underestimation of KENDA-1 Ts."*

*A closer estimate e.g. of the error induced by the training with the sounding of Payerne would necessitate a complete study including sounding at Meiringen at different periods of the year and with different weather type patterns. As stated in the first version of the manuscript, we were able to perform only three soundings and, sorrily, the weather conditions during that day (constant high-altitude inversion) did not allow to draw a preliminary conclusion regarding the differences between T profile over PAY and over MEE. Moreover, the MWR was not yet measuring by that time. Several reasons impede further RS observations at MEE during the campaign.*

3. I appreciate the 3D map (Fig. 1), however, it would suggest to use the "northing". Moreover, it is very similar to Fig. B1, although B1 contains some added useful information. I would suggest to replace Fig. 1 by Fig. B1, and also include some information from Fig. G1 (specifically, I would find it a lot easier if wind arrows would depict the median wind direction instead of coloring the stations accordingly).

*The corrected manuscript contains only one figure with all necessary information and the northing view. The colors of the stations on Fig. G1 did not correspond to wind directions but were just added to allow an easier description in the text. The stations are now depicted on Fig. 1 that only concern topography without any wind information. We hope that the colors will no more cause misinterpretation in the corrected manuscript.*

4. The different instrumentation and sites are well described. Due to different durations of employment I would appreciate an overview table of instrumentation, available data, sites, and the measurement period.

*The required table is introduced in the supplement in order not to lengthen the manuscript.*

5. l. 86: COSMO-1E: Please introduce this abbreviation.

*As requested by the first referee, the introduction was shortened so that the abbreviation "COSMO-1E" only appears in the experimental section, where it is introduced.*

6. l. 136 "Five km before the lake": I would suggest to write "Five kilometers".

> *Done*

7. l. 186: "perpendicular to the valley (not used in this study)": To streamline the manuscript I would suggest to only describe the measurement setup that was actually applied during the campaign.

*This is a wise advice. We suppressed the description of the not used scanning mode.*

8. l. 197: "These precipitations arrived in form of snow": I'm not sure if precipitation is commonly used in plural.

*Changed in singular*

9. Section 2.4

I appreciate a description of the weather during the campaign, although I do not fully understand why the authors focus on precipitation, snow, and sunshine duration while the focus of the study is on wind and temperature profiles and circulation features. I think this section could more strongly focus on the relevant aspects for the analysis.

*The description of the weather during the campaign was largely shortened. First the T features are summarized. Then a sentence explains the importance of snow cover and precipitation before to summarize the precipitation patterns during the campaign. Finally, a sentence was also added to inform that the wind features during the campaign are described in the results section. As suggested the explanation of the weather situation no more constitutes an individual section but was added as a heading to the results' section: "During the campaign, the mean T was ~1°C below the 1991-2000 norm in December and January but clearly above the norm (1.5 to 2.5°C) in February, March and from May to August. Three heat waves occurred, the first one lasting 6 days in mid-June, the second lasting 4 days around mid-July and the third one reached Switzerland at the beginning of August. Snow cover and precipitation are important parameters since the surface albedo and the soil moisture affect the development of cold pool with T inversion, subsidence, the atmospheric boundary layer development and consequently thermal valley winds. Only 60% of the precipitation of the 1991-2000 norm were observed in November, but 120% in December. Snow covers the valley's floor from the end of November to mid-December. Heavy precipitation reduced then the snow cover to less than 15 cm until the end of the winter. Strong precipitation deficits happened in January and especially in March (35 and 15 mm). March experienced frequent foehn events (95 hr determined from the MeteoSwiss foehn index (Dürr, 2008)). Precipitation from May to August was 50% or less compared to the norm, except for June (96%). The full evolution of T, precipitation and sunshine duration is aggregated in the Supplement (Tab. S2 and Fig. S3) and the wind features are fully described in the results section."*

10. Section 2.5

I believe that this section is not necessary as a separate section, but the information should be included in the previous paragraphs, e.g., where the sites, instrumentation, KENDA, etc. are described, respectively.

*The subsection 2.5 was deleted and the related information dispatched at the beginning of the section or in the subsections.*

11. l. 590: "Therefore, this study does not allow to make prediction of model performance for forecasting." Can the authors please elaborate on this, I do not fully follow the reasoning and relation to forecasts here.

*We simply wanted to highlight first the potential artifacts and bias bounded to the used of monthly medians, second the focus of this analysis on climatology and not forecasting skills of the COSMO-1E. In that sense, a good accordance of median values between the model and the observations does not allow any predictions of COSMO-1E performance as a forecast model. The analysis of special cases such as foehn events underlines this point. Following your remark and a request of the other reviewer, this § was shorten: "Finally, this study is based on monthly median values, so that the averaging artifacts has to be considered, e.g. for the analysis of maximum wind speed, the onset time of valley wind or wind directions. In that sense, this analysis focused on climatology and not on the forecast skills of COSMO-1E."*

12. Results

I would suggest to streamline the Results Section (see also General Comments above).

13. I would suggest to use more informative titles in the Results Section (e.g. "3.1.1 Climatology" is only based on observations, which cannot be deduced from the title).

*The titles were modified to include all necessary information in order to correctly describe the section content.*

14. The overall section structure could be improved. It is not very intuitive to show (i) temperature, (ii) wind, and (iii) Foehn (with wind and temperature), while other atmospheric features were explicitly discussed in (i) and (ii). I would suggest to define a storyline to follow in the Results Section and focus on the key results.

*The structure of the result section was largely modified to have a better storyline. For example, the section on wind comprises now*

> *3.2.1 Seasonality of wind profiles at MEE*

> *3.2.2 Along valley winds*

> *3.2.3 Cross valley winds*

*The foehn is however a very specific meteorological event that is particularly difficult to model. We prefer then to keep the analysis of foehn event as a separated subsection comprising both the T and wind analysis during foehn events.*

15. Generally I would be careful with the word "climatology" as here only a few months and not several years of data are analysed.

*We do agree that the use of "climatology" for a 10 months analysis is partly usurped, so that it was removed in the entire manuscript and replace by monthly values/medians/averages or by seasonality.*

16. Section 3.1.1

I would suggest to streamline this section and focus on relevant features. E.g., the presence of a diurnal cycle and temperature increase after sunset are expected features and their description could be streamlined.

*The structure of section 3.1 was modified and comprise now only three subsections (Seasonality of T profiles at MEE, Surface T comparison and Surface T inversions) and it was also shortened. I hope that these modifications improve the reading and understanding of the results.*

17. l. 230f: Please correct units: temporal gradient of 5C/?.

*Done: it is °C/km*

18. l. 235: I struggle to see the daytime temperature inversions. Could these features be outlined in the figures (e.g., through contours)?

*Daytime T inversions occurs in winter but this results is not visible on Fig. 3a but on Fig. 7a. The sentence was then deleted so that the analysis of surface T inversions is only described in the related section.*

19. Section 3.1.2

Which differences are analysed in this section? Is it hourly values?

*Yes, Figs. 4 and 5 are done with hourly averages from the whole campaign. It is now specified in the figures' caption.*

20. l. 255 f: "The difference in the effect of the ELR correction" Which difference? At both stations RMSE increases? Please clarify.

*As also stated by the first referee, the difference is not obvious. This sentence was deleted.*

21. Section 3.1.3

Would it make sense to place this sub-section before Section 3.1.2 as profiles have already been described Section 3.1.1 (Fig. 6 fits better to Fig. 3)? Is Fig. 3b required? If I'm not mistaken it is not referenced in the text. The comparison of temperature profiles and respective KENDA biases assumes the MWR retrieval does not include a bias. Given some uncertainty in the retrieval algorithm, could the authors comment on this issue (see also comment above)?

*Yes, it makes sense and sections 3.1.1 and 3.1.3 are now merged into one single section. A new figure (Fig. 2 in the corrected manuscript) comprises Fig. 3a and Fig. 6.*

22. Section 3.1.4

l. 308 f: "All this leads to both an important overestimation of the T at ground level (Fig. 5) and a slight underestimation of the T just above the T inversion (Fig. 6)." Both figures compare different data sets, i.e. a direct comparison of temperature differences at different heights is difficult. Moreover, I struggle to see temperature overestimations at the lowest level in MWR-KENDA comparisons in Fig. 5. Can the authors please clarify?

*Figure 5 does not directly compare MWR and KENDA-1, but compare MWR and KENDA T at the first level with SMN/MWR T observations. Fig. 5 clearly shows an T overestimation by KENDA-1 during nighttime (positive difference from 20h to 7h) where no averaged difference is found between MWR and the SMN/MWR observations (blue dashed line). We can conclude that KENDA-1 overestimates the ground T at both MEE and MER but not MWR/MEE. The KENDA-1/MEE underestimation from 850 m to ~1200-1500 m is visible in Fig. 3b and occurs mostly during nighttime, even if daytime underestimation is also present. Fig. S5 in the Supplement presents individual MWR/MEE and KENDA-1/MEE profiles allows a better understanding of the described phenomena.*

23. l. 309-311: Please either elaborate on this or remove.

*Since a systematic analysis of this effect was not done, we chose to remove this result.*

24. l. 315 - 322 I find this very interesting and would like to see some results, as this observation rejection is linked to some of the surface temperature differences reported in the study.

*We only analyzed visually a period with strong T inversions and large error in the modeled T in March 2022. As you can see from the figures below, COSMO-1E did not assimilate the SMN/MER T due to too large differences reaching up to 10°C in some cases. During the same period, the humidity at 2 m was largely underestimated during nighttime by the model. This is only a first rapid study and a complete analysis of the causes of the model deficiencies in case of T inversion in middle size and narrow valleys would be very interesting and will perhaps be the focus of a next study.*

[Figure]

*Figure 1: Ground T during the end of March 2022 with SMN/MER observations in black, COSMO-1E first guess in blue and model analysis in red. Red vertical bars denote times when the observations are rejected.*

25. Section 3.1.5

Personally, this very short sub-section interrupts the storyline which strongly focuses on MWR and surface measurements. Did the authors also compare MWR profiles with the RS profile? Please note that Fig. F1 is not referenced. Please also elaborate on the influence of the RS from Payerne. Was the additional RS/MEE not assimilated? Please double-check the spelling of radiosounding.

*This result does not bring any relevant explanation for the T and wind results of this study. It was consequently removed. The MWR was not yet measuring in November 2021, so that the comparison was not possible. We cannot elaborate more on the influence of the RS from Payerne on KENDA-1 due to the absence of further radio-sounding measurement in complex terrain. No try to assimilate this isolated radio-sounding was done.*

26. Section 3.2.1

This section appears unfinished and I think it needs some more work. The writing style with bullet points differs substantially from the style used above for temperature. Moreover, I would ask the authors to improve

*As already explained before, the entire result section was reorganized and modified, including section 3.2.1. Bullet points were removed, the redundant content with section 3.2.2 (along valley wind) was suppressed.*

Fig. 8. It is very difficult to identify relevant features in a 60 panel figure. Would it be an option to, e.g., show differences in panel b)?

*As explained as an answer to the general comments about figures, the difference of median monthly wind direction are difficult to interpret so that we will keep the present representation.*

27. Section 3.2.2

This section should be removed.

*The structure of the paper was entirely revised and this title was removed.*

28. Section 3.2.3

I appreciate the comparison of allong valley winds during the campaign period to the 10-year climatology, however, this interrupts the storyline. In my opinion, it would be sufficient to mention the good agreement and move the figure to the appendix.

*To shorten the paper and improve the story line, the climatology of along- alley wind at SMN/MER was moved to the supplement.*

29. Fig. 10a: Typo in axis label: "DWL/MER"

*Done*

30. Section 3.2.4

This section provides a very detailed description of the circulation evolution at different sites. I would ask as the authors to streamline this section and highlight the important circulation features. Figure B1 could also be referred to for clarification.

*This section was moved after the along and cross valley description and merged with results about the differences between MEE and MER. Its title is now "Heterogeneity of wind patterns in the Haslital valley". Moreover, the content was simplified and shorten. We hope that these modifications increase the manuscript readiness.*

31. Section 3.2.5

l. 493-495 Could the authors please elaborate on this? How was the vertical velocity estimated?

*The observation of the vertical velocities is part of the DWL scanning procedure, so that these were not estimated but measured by the DWL. In order to better describe this vortex, an example of the radial winds perpendicular to the valley axis was added to the supplement.*

[Figure]

[Figure]

*Fig: DWL/MEE adial speed perpendicular to the valley axis on the 11 July 2022 at 19h18. A Vortex is clearly visible.*

32. Section 3.3

General comment: In my opinion, much of the comparison between KENDA and observations has already been described above. I would suggest to restructure to avoid repetions and potentially include foehn events as subsections in 3.1 and 3.2.

*As already station under question 14, we prefer to keep the foehn description as an individual section but the text was revised to avoid unnecessary repetitions.*

33. l. 505: Please define the foehn index and provide a reference.

*The foehn index is mentioned with a reference in the experimental section, so that this information will not be repeated here.*

34. l. 505: Is the subsequent analysis (e.g., Fig. 14) performed for three events only or for "all the period" with foehn? Please clarify.

*The analysis of the T modeled performance is done on all the periods with foehn, whereas the analysis of the wind model performance is done on only the three mentioned cases. The text has been modified for a better comprehension: "Foehn is in a katabatic wind bringing strong warm and dry downdraughts associated generally with clear weather. The study of the T during foehn events combines all the periods where foehn was measured at the SMN station in MER, according to the foehn index. The study on the wind is however performed on only three selected events (10-16.03.2022/19-22.03.2022/23-24.04.2022) representing foehn 117 hours during clear weather in March, while the April and June episodes presented a slightly overcast sky (50-70% of maximum global radiation)."*

35. l. 506: Which April episode? Fig. 15 shows again different time periods? Please clarify.

*There was some incoherence between the text and the figures and some further typos in the x label of Fig. 15. The three foehn episodes represented in Figs. 15, 25, 26 and 27 correspond to the events of the 10-16.3.2022, 19-22.3.2022 and 23-24.4.2022. They comprise then the April event mentioned in the text.*

36. l. 516: I would expect a better agreement of KENDA and observations if both data are taken from the same site (compared to different sites). I would appreciate if the authors could explain the reason for comparison of KENDA/MEE with SMS/MER (instead of KENDA/MEE). Such comparisons are frequently performed throughout the manuscript, and to some extent it is difficult to follow all different comparisons.

*It was not possible to install the REM instruments at MER during the campaign, so that they were put in MEE. The frequent comparisons between MEE and MER are due to this campaign setup comprising ground data with the smallest uncertainties that are measured at SMN/MER whereas the REM profiles were at MEE. The second reason is that, mostly regarding wind profiles, we found marked differences between both sites such as very different thermal valley wind diurnal cycles or wind speed during foehn events. These differences were not expected prior to the campaign but they effectively lead to complex comparison between MEE/MER, ground/profiles and observed/modeled data. Anyhow this complex setup allows having a good representation of the influence of the complex topography on the T and wind patterns.*

37. l. 521: "mettre reference fig ap 26"?

*Done*

38. l. 537: Does KENDA/MER really show a delay in foehn onset? To me it rather looks like a too early onset (similar to KENDA/MEE)?

*Yes, there is also a too early foehn onset modelled by KENDA-1/MER. The word "delay" was wrongly used here as a time shift that could be positive. The manuscript was corrected: "the same too early onset of the foehn breakthrough is observed …"*

39. l. 541-544: In particular in this section it is very difficult to see where wind speed is over-/unterestimated, i.e., showing differences to DWL might be helpful.

*I think that the difficulties to see wind speed misestimation comes principally from a mistake at L540, where KENDA-1/MER is mentioned instead of KENDA-1/MEE. The text was however also improved: "During the second episode, KENDA-1/MEE models correctly the 3h delay between SMN/MER and DWL/MEE measurement (Fig. 12.b) but extends it up the ridge height contrarily to the measurements. The KENDA-1/MEE wind speeds tend to be overestimated (+15 km/h) from ground to 1100 m during the entire event and underestimated from 1100 m to the ridge's height (-30 km/h) the first hours following the breakthrough. KENDA-1/MER modelled again wind speeds up to 100 km/h with a foehn breakthrough at the same time as the SMN/MER (Fig. 12.c)."*

40. l. 545: Based on Fig. 15 (which is the main figure discussed in this paragraph), I find it difficult to follow the conclusion that the representation at MER is better than MEE. Improved visualization may help.

*We agree that KENDA-1/MER cannot be considered as better than KENDA-1/MEE. The text was modified: "To summarize, the three analyzed events exhibit some similarities but also large differences. The foehn breakthrough is often observed some hours later by DWL/MEE than by SMN/MER and not always simultaneously in the entire profile. The wind speed at the DWL/MEE first level is usually similar to the one at SMN/MER. KENDA-1 tends to model the foehn arrival and end with positive or negative time shifts at both stations. The most critical point concerns the very high KENDA-1/MER modeled speed up to 110 km/h from ground level to 1500 m that is twice faster than the DWL/MEE observation, 5 km further down in the valley."*

41. l. 554: Please elaborate on the link between wind bias, temperature bias and specific humidity bias.

*Relation between moist bias and T are obvious, since warm air is able to contain more humidity and more humid air need more energy to be heated. Anyhow there is no hypothesis about a relation between a moist bias in the model and the wind speed overestimation. This sentence was then deleted.*

42. Fig. 14: Please specify what is shown in the figure. What do the numbers on top represent? What is shown in the x-axis in b)?

*The numbers n on top are the number of cases in each category. This information was added in the figure caption. The x-axis labels corresponding to wind categories were corrected.*

43. Fig. 15: Are the dates correctly shown in all panels? I cannot find any figures for 19-22 March nor 23-24 April. Are data shown only at 11 am and 11 pm?

*No, the dates are not correct and Fig. 15 was revised.*

44. Discussion

The manuscript includes an extensive discussion of the results, which I appreciate. However, I would suggest to condense the information and streamline the Discussion Section. It may also be helpful to include a short summary and/or discussion after the respective Results Sections, respectively.

45. l. 680: The daily cycle of temperature underestimation/overestimation is not apparent in the MER observations (Fig. 6). Could the authors please elaborate on this?

*Fig. 6 allows to compare the observed and modeled profiles at MEE and not the ground T at MER, which is represented in Fig. 5. This sentence refers to the ground T comparison at MER, that presents a diurnal cycle of overestimation during nighttime due to the missed T inversion and underestimation during daytime. This cycle of the T difference is much less visible at the lowest levels of the profiles (Fig. 6), due to the difference in elevation and in site as well as to the larger uncertainties of REM instruments compared to ground observations.*

46. l. 681 f: Comparisons are made to different versions of COSMO. Please either specify the versions /differences or remove.

*COSMO-1E is the only used version in this paper since KENDA-1 is the analysis mode of COSMO-1E. The nomenclature was harmonized in the whole manuscript.*

47. Conclusions

The Conclusions provides a bullet point summary of the key results. I would suggest to formulate continuous text for the conclusions with distinct paragraphs instead of bullet points.

*Done*

48. l. 772 - 787. The important circulation features are listed here. I would appreciate a figure/sketch similar to Fig. B1 where all the identified flow features are outlined and summarized

*Certainly, but up to now we didn't have found a good way to simply skitch the diurnal cycle of the various wind compounds. We will try a representation and incorporate it to the manuscript or the supplement if we find it helpful.*

---

## Author Comment (AC2)

**Answers to the reviewer 1 comments on**

**"Comparison of temperature and wind between ground-based remote sensing observations and NWP model profiles in complex topography: the Meiringen campaign"**

First of all, we would like to thank the reviewer for the valuable, in-depth comments to our manuscript. We would also like to apologize for the numerous typos, wrong links to figures and not complete references that have sorrily lengthened the review process. As supposed, the manuscript was written very rapidly and the latex implementation to the AMT formal leads to unexpected problems. Second, the appendix was really designed as a supplement but just not submitted in a separate file. The revised version produces now two distinct files. Finally, according to both reviewers' request, the manuscript was largely shortened (35 pages instead of 42) and contains now only 12 figures.

The answers to the comments and questions are written in italic thereafter. When modifications of the manuscript are cited, the numbering of the figures correspond to the one of the new manuscript. The explanations themselves cite the numbering of the figures in the submitted manuscript in accordance to the lines' numbers of the comments.

Reviewer 1:

**General comments**

While the research framework is well-designed and current literature well referenced, the manuscript is apparently written too quickly (missing references, typos, erroneous labels in figures, etc.) and has not yet reached the minimum quality standard for publication. Notably, the manuscript is very long (53 pages, too many considering that it is not a review paper) and more synthesis is required. Indeed, there is a lot of interesting scientific material, but too much detail is to the detriment of the overall focus of the paper and the reader's attention.

I suggest that the authors revise the form of the manuscript and try to shorten it. The (10!) appendices could be transformed into a Supplement, where also a part of the text could be moved. More suggestions are provided hereafter.

*We apologize for the numerous typos in the first version of the manuscript and hope to have corrected all of them. We revised and reorganized the manuscript in order to synthesize the results and discussion sections and to shorten the text. The appendix was also shortened and transformed into a supplement.*

**Specific comments**

- Title: temperature and wind could be cited in the title, as they represent useful keywords

  *This is a very good suggestion. The title was modified to "Comparison of temperature and wind between ground-based remote sensing observations and NWP model profiles in complex topography: the Meiringen campaign"*

- Introduction: multiple sub-sections in the introduction are a bit unusual. I would suggest that the authors shorten this section and limit the information to the most relevant topics for the paper

  *The sub-sections were removed and the introduction was shortened.*

- Sect. 1.2: here the authors should better highlight that the analysis of the model "performances" does not focus on the forecast skills, but on the ability by the model to represent the average general patterns. I also suggest that they replace the word "climatology" (which would need a much longer period) with "statistical analysis" or "average weather patterns" or a similar expression

  *It is now specified that the model was only used in analysis mode for monthly averages and some specific events.*

  *The word climatology is no more used and replaced by seasonality in the titles and by monthly median/average in the text.*

- Sect. 2: the authors could move the description of the site (now Sect. 2.2) and the weather situation (now Sect. 2.4) at the beginning to allow the reader to better follow the description of the instruments. I would suggest that the numerical (now Sect. 2.1) and experimental (Sect. 2.3) tools could be brought closer (as 3rd and 4th subsections).

  *The experimental section was reshaped with first the description of the station, then the description of the model and finally of the used instruments. The weather situation during the campaign was also shortened and moved as an introducing paragraph at the beginning of the result section.*

- Sect. 2.2, l. 134: if these are important geographical features, please include a figure with these references

  *The Gadmertal is not an important feature and is not further mentioned in the paper. It is visible on Fig. 1.a but not explicitly mentioned to avoid overloading of the map.*

- l. 149-153: is the altitude bias depicted in Fig. 2 only due to inclusion of the slopes in the model cells (as written at l. 149-150) or is this bias due to different DEMs (KENDA-1 and the 25 m elevation model)? In the first case, I cannot explain why the average bias in some cells is far from zero. As a further note, this paragraph could be moved in the model description

*Fig. 2 represents the difference between the cell mean altitude and the 25 m digital elevation model at a 25 m resolution. L. 149-153 just mean that cells containing both valley floor and slope have a more important bias than cells situated only in the valley floor. The text was modified: "Data from the two grid cells containing the MER and MEE stations were used. Both cells include part of the valley's north slope, inducing differences of 109 m and 130 m between the real topography and the model's terrain, respectively. The modeled valley floor is globally raised by a hundred meters (see Supplement, Fig. S2), whereas the ridges and the Brünig Pass are lowered with respect to their real altitudes. The altitude difference between the valley floor and the crests is thus reduced of several hundred meters and, in particularly, the Brünig Pass is only 200 m higher than the valley floor."*

*It was also moved at the end of the model section. Moreover, in order to shorten the manuscript, Fig. 2 was moved to the supplement.*

- Fig. 1: a map over a wider area, such as the one in Fig. B1 should be reported in Fig. 1 (e.g., as a subfigure) to help the reader better understand the geography of the valley

*Fig. 1 was modified in order to present a broader view of the geography in the region of the Haslital and comprises now a) a map over a wider aera comprising the complete Haslital, both lakes of Brienz and Thun and the Sarneraatal, b) a view of the terrain elevation along the Haslital and the ridges heights and the valley flood width and c) a detailed view of the Haslital and the Sarneraatal including all used stations and the used KENDA-1 cells.*

[Figure]

- Sect. 2.3.3: can you shortly explain what the MWR "training" implies?

*The retrieval algorithm employs simulated brightness temperatures at specified frequencies and elevation angles, which are obtained through radiative transfer calculations using Payerne radiosonde data. A multi-linear regression is conducted, establishing a relationship between the forward-modeled brightness temperatures and the atmospheric temperature and humidity measured by the radiosonde at a defined height level. This training was performed by the manufacturer RPG and is described in Lohnert and Maier (2012) based on Crewell and Lohnert (2007) methodology.*

*https://amt.copernicus.org/articles/5/1121/2012/amt-5-1121-2012.pdf*

*https://ieeexplore.ieee.org/abstract/document/4261043*

*S. Crewell and U. Lohnert, "Accuracy of Boundary Layer Temperature Profiles Retrieved With Multifrequency Multiangle Microwave Radiometry," in IEEE Transactions on Geoscience and Remote Sensing, vol. 45, no. 7, pp. 2195-2201, July 2007, doi: 10.1109/TGRS.2006.888434.*

*The manuscript was also modified: "During the Meiringen campaign, the retrieval of Payerne was used (Lohnert and Maier, 2012). This retrieval uses Payerne's radiosonde data to perform the multi-linear regression leading to potential further uncertainties."*

- Sect. 2.3.4: is the vertical component of the wind velocity used anywhere in the study? Also, there is no mention to a blind (low) zone in the DWL measurements, however the DWL plots start higher than ground level. For the same reason, I guess that the T inversion is not detectable from the wind fields (DWL)? Is it possible to detect any turbulent mixing phenomenon (e.g., development of PBL) at the bottom of the valley from DWL, overlapping to the slope/valley circulation?

*The vertical component of the wind velocity is used to calculate the horizontal wind speed with the DBS algorithm. The blind zone is of 200 m and the manuscript specify the limits of the DWL measurements from 200 to 12000 m a.g.l. The altitude of the DWL first level is now mentioned explicitly and correspond to the altitude of MEE stations (574 m) plus the depth of the blind zone (200 m).*

*The DWL derives wind direction and speed from aerosol backscattering observations. It is consequently not able to measure T profiles and to detect T inversions.*

*The development of the PBL can be estimate not only by the DWL but also by the MWR and the ceilometer. The automatic PBLH detection from the DWL was tested at Payerne, compared with other detection methods and found not to be completely reliable.We did a first analysis of the PBL height from the MWR data at Meiringen leading to interesting results that still have to be compared with the Ceilometer PBL estimates. Even if the inclusion of these analysis could bring further explanatory variables, we estimate that it would lengthen the manuscript so that we did not include these preliminary results in the paper.*

*The observation of the overlapping to the slope/valley circulation needs a much more comprehensive setup. A good example is given by the CROSSIN campaign (https://journals.ametsoc.org/view/journals/bams/102/1/BAMS-D-19-0283.1.xml) in the Inn Valley using three DWL performing synchronized continuous coplanar RHI scans. This setup*

*allowed the retrieval of the two-dimensional cross-valley wind vectors in the scanning plane. The setup during the Meiringen campaign does not allow such a sophisticated analysis.*

- Sect. 3: it may be a matter of taste, but as a reader I would be more comfortable if the results were split between "Climatology" and "Case studies" rather than "Temperature" and "Wind" (each one with analyses on both the long- and short-term). That would first provide an overview on how the model performs, then the focus could be on specific episodes

  *The paper could also have been organized as proposed. Anyhow, the case studies include one subject only related to T (surface based T inversion), one subject only related to wind (heterogeneity of wind along the Haslital valley) and finally the special case of foehn events where both the T and the wind compounds are important. We prefer then to keep the present structure and we hope that the present modifications have improved the manuscript and make the reading easier.*

- Fig. 3: what is the reason for the "erosion" in the temperature field below about 1000 m in May and June at the end of the day?

  *We do not observe a peculiar erosion in the T field below 1000 m in May and June at the end of the day. Anyhow, the maximum of the color scales is adapted for each month (15°C in April, 20°C in May and 24°C in June) leading to shift, e.g., of the yellow color from 8°C in April to 14°C in May and 16°C in June and perhaps giving the impression of a T erosion.*

- l. 259: it is stated that the environmental lapse rate correction is "not precise in specific cases". However, it is not even precise on average. More generally, I would remove the whole paragraph at lines 251-264 and just mention that tests using a fixed lapse rate determined that horizontal/vertical distances between the station and the KENDA-1 cell are not the reason of the observed T discrepancies

  *We do agree that tests with the environmental lapse rate correction were important to reject the hypothesis of large differences due to the horizontal and vertical distances, and that they are not very relevant for the rest of the study. Following your proposition, the text was largely shortened:*

  *"To check if the altitude differences between the station and KENDA-1/MER-MEE first levels could explain the T differences with SMN/MER, a standard T correction with a mean environmental lapse rate (-6.5 °C/km (Lute and Abatzoglou, 2021)) close to the mean measured MWR/MEE lapse rate (-4.59 °C/km between 590 and 740 m) was applied to the modeled profiles. Considering the remaining T differences after the correction (grey in Fig 5.b and 5.c), we conclude that the horizontal and vertical distances between the SMN/MER station and the first level of KENDA-1/MEE are not the main causes of discrepancies in ground T estimation."*

- l. 264-265: do these differences present a seasonal cycle? Is there a figure similar to Fig. 5 for each month?

  *No such figures for each month were done. Anyhow it was shown that T overestimation by KENDA-1 is the largest at low T (< 5°C) and T underestimation the greatest at T>20°C. There is then probably a clear seasonality that follows the climatology of T as well as the T inversion seasonal*

*cycle (see sect 3.1.3). We do not think that this information would enhance the quality of the paper so that we will not incorporate this potentially new figure in the revised manuscript.*

- l. 304: could the difference in the frequencies of T inversion between MWR and ground station be an effect of the surface, i.e. due to the fact that DWL measurements are in the free atmosphere and the station is on the ground?

    *This is a very interesting but very complex question. Differences due to the direct influence of the ground could relate to:*

    - *The presence of a shallow fog layer that would affect both ground stations without extending to the middle of the valley. Fog usually forms lakes in the valley so that the influence of fog is probably not a reliable potential explanation.*
    - *The moisture state of the ground could also influence T inversions since water is a very good IR emitter. A complete radiative budget should be perfomed in order to make some reliable assumptions.*
    - *Snow has, on the contrary, very good insolate properties. Different snow coverage as a function of altitude would lead to larger daytime heating and nighttime cooling in the valley floor than at higher altitude. Such cases are however not relevant during very cold days as well as during late spring and summer.*
    - *The more simple explanation for large differences between the temperatures at the ground and in the free atmosphere would be nighttime down slope winds that would cool only a few tens of meters above ground.*

    *All these explanations are, however, quite speculative and cannot be solved in the framework of the Meiringen campaign.*

- l. 309: in March and April, the differences between MWR and KENDA-1 are not "just above" the T inversion

    *The sentence at L 309 ("All this leads to both an important overestimation of the T at ground level (Fig. 5) and a slight underestimation of the T just above the T inversion (Fig. 6).") aims to underline that KENDA-1 overestimates T at ground level (Fig. 5 is the comparison between KENDA-1/MER and SMN/MER) but slightly underestimates T above the T inversion, as can be seen in the profiles of Fig. 6. This behavior is obvious if individual T profiles are analyzed (Fig. S20 et S21). Anyhow we do agree that the T underestimation by KENDA-1 is not restricted to the altitudes just above the T inversion but to the greatest part of the profiles between the first MWR level (625 m) and 1500 m. To avoid any confusion, the manuscript was modified: "The missed T inversions by KENDA-1/MEE leads to both its important overestimation of the T at ground level (Fig. 4) and its slight T underestimation between ~850-1200 m (see Supplement Fig. S5 for detailed examples)."*

- Sect. 3.1.5: this section is very short and maybe not too relevant. Can it be removed?

    *Since this result is effectively not relevant for the rest of the study, it was removed.*

- Sects. 3.2.1, l. 341-342: this classification method sounds a bit naive, and some similarity between the w<20 km/h and w>20 km/h diagrams (Fig. 8) are visible, i.e. no clear boundaries are found between synoptic and thermal circulations. Could the authors further elaborate on that?

*We do agree that this classification method relay on the arbitrary threshold of 20 km/h to separate synoptic winds from thermal valley winds. A separation between wind speeds smaller 10 km/h, wind speeds comprised between 10 and 20 km/h and wind speeds higher than 20 km/h clearly showed that valley winds are already observed for speed < 10 km/h but are much better defined if speed until 20 km/h are considered, whereas different clear features are visible for speed > 20 km/h. The maximum up valley speed measured in the alps (30-35 km/h see Schmid et al., 2020, Adler et al., 2021 and Giovannini et al., 2017)) in case of clear sky conditions leads to lower monthly median speeds for all weather conditions, i.e. no selection of days favoring thermal valley winds. The foehn have usually high wind speeds reaching 70-140 km/h ([https://www.meteoswiss.admin.ch/weather/weather-and-climate-from-a-to-z/foehn.html](https://www.meteoswiss.admin.ch/weather/weather-and-climate-from-a-to-z/foehn.html) ) and Bise events also have speed > 40-50 km/h over the Swiss plateau. The westerlies are usually less strong and they can be observed in the categories with speed < 20 km/h. However, the main purpose was to exclude strong foehn events from thermal winds. This goal is reached since thermal valley winds can be observed in March even if this month is the most strongly affected by foehn events.*

- l. 396-397: is the difference between 3.5 and 4 hours significant (and relevant)? Also, is the +/-1h offset described at l. 399-400 significant?

  *We do agree that these differences are not significant. The whole paragraph was modified:" Fig. 7.a shows the diurnal and seasonal cycles of the along valley wind speed at SMN/MER during the campaign. The occurrence of along valley winds is confirmed by the diurnal cycle in November and from February to August. A 3-4 hours delay between sunrise and the onset of up valley winds (> 10 km/h) is observed. February shows some early up valley wind, but their origin is more linked to synoptic flow intrusions. The transition to down valley winds occurs 1 hours before sunset in March and June and around sunset otherwise."*

- l. 453: the wind is defined as "Up valley", but has E/SE direction

  *This is a typo. The right time of up-valley wind measured by DWL/MEE is on the 10 of July at 13:00-14:00, namely the second reported time in the sentence. The sentence was corrected: "At the DWL first level (200 m a.g.l.), up valley wind is only measured in DWL/MEE on the 10 July at 13.00-14:00 (Fig. 10.a, color bar). The wind direction switches thereafter to N and the wind speed increase gradually to reach 40 km/h at 20:00."*

- l. 457: "only observed between 1300 and 2000 m", but it looks like there is a positive along-valley at 800 m in Fig. 10

  *Fig. 10 refers to the monthly median wind speed whereas section 3.2.4 analyzes three clear-sky days in July 2022. The result described in L. 457 is not shown since we did not report another figure with complete DWL profiles for these selected days. We mentioned now in the manuscript that this result is not presented.*

- l. 464-470: a "3D" figure with the winds depicted as arrows would be very beneficial for the readers not familiar with the Swiss geography

*The wind regime is constantly changing in the Haslital valley during these three days, so that no clear and explicit figure with the various configuration of the wind could be produced. We did a skitch of a small film with the wind flows during several days that can however not be included in such a paper.*

- l. 478: "from NE", this seems to contradict Fig. 8b ("green colour" of the wind direction during the night)

  *As in L 457, L 478 refers to the in-deep analysis of three days in July and not to the monthly median directions presented in Fig. 8b.*

- l. 578-584: is the presence of a lake really discussed in the study? Can you better explain why the model would not be able to deal with a lake?

  *This paragraph does not explicitly mention known problems of models in complex topography but aimed to describe the particularities of the Haslital valley. These peculiarities could explain differences between the results in MEE and in MER as well as different thermal wind behavior than reported in other studies as mentioned at the end of the last sentence.*

- l. 596: "equally common", is this the case? The last sentence of Sect. 4.2.1 lets me think the opposite

  *This sentence describes the results of Joly and Richard (2019) and not the present study, where T inversions are less frequent in summer than in winter.*

- l. 602: is the "thickness" of the T inversion analysed here?

  *Yes. It is now better specified in the manuscript:" The intensity, magnitude and thickness of these surface T inversions follow a similar seasonal pattern as observed in the Haslital."*

- Sect. 4.3: can qualitative concepts such as "accurate results" (l. 675), "large modeling errors" (l. 677), "satisfactory" (l. 686), "very good performance" and "well modeled" (l. 722), etc. be quantified in a more precise way, i.e. based on some performance targets.

  *We have reformulated the mentioned text and put the results of this study into perspective of the standard verification against radiosondes and surface observations regularly done at MeteoSwiss. This verification is done averaged over the whole domain and comprises complex and non-complex terrain.*

- Conclusions should be better synthesized. So many "bullets" and detail are not common in the conclusions and do not help the reader get the overall idea of the outcomes.

  *The conclusion has also been extensively reworked. Only the main conclusions remain, results of observations and about Kenda-1 performance are now grouped for T and wind and all the bullets were removed. We hope that the most important results are now easier to grasp.*

**Technical remarks**

- Please, read the "Manuscript composition" guidelines (https://www.atmospheric-chemistry-and-physics.net/submission.html) and the LaTeX template, if necessary. Correct the bibliographic references, section/table/appendix/figure references, date formats, etc.

  *See general answer to the reviewers' comments. We corrected all the mentioned types of typos.*

- l. 2: why do you cite the "mesoscale" and not the local scale?

  *The manuscript was modified: "Thermally driven valley winds and near-surface air temperature inversions are common over complex topography and have a significant impact on the local and mesoscale weather situation."*

- l. 3: "it" or "them"? Vertical exchange is explained here, but horizontal transport should also be mentioned (also at lines 21-22)

  *The manuscript was modified:*

  *L3: "Valley winds affect them by favoring horizontal transport and exchange between the boundary layer and the free troposphere, whereas temperature inversion concentrates pollutants in cold stable surface layers."*

  *L21-22:" Over mountainous areas, interactions between the terrain and the overlying atmosphere favor the horizontal and vertical transports of moisture and pollutants and consequently increase the air masses exchanges along the valleys and between the boundary layer and the free troposphere."*

- l. 8: "measurement's" --> "measurement"

  *Done*

- l. 9: "influences"

  *Done*

- l. 16: "too"

  *Done*

- l. 22: bibliographic references can be added (e.g., anticipated from the next lines) after "troposphere"

  *References to de Wekker et al, 2015 and Rotach et al., 2022, two papers describing general phenomena leading to exchanges with the troposphere in complex terrain, were added.*

- l. 31: "air T" --> "air temperature (T)"

*Done*

- l. 51: "slope-flow-induced local subsidence", please rephrase

  *The sentence was changed:" They can partially be explained by the topographic amplification factor concept (Whiteman, 1990) and local subsidence in the valley center induced by upslope-flow (Schmidli and Rotumno, 2010) leading to a faster heating of the valley than of the plain."*

- l. 121: differences between Schraff et al. (2016) and the "current setup" are not clear

  *We have described the main differences in the text: " Differences to the setup described in \cite{schraff_2016} include the modeling domain (central Europe covering the Alpine Arc), the grid size of 1.1km and the observation errors tuned to the MeteoSwiss setup."*

- l. 132: "upper (southern)"

  *Done*

- l. 134: does the valley with 1.5 km wide floor continue towards NE or NW?

  *Right, NW direction. Done*

- l. 143: "provide from", check syntax

  *Done : « The campaign provides in-situ observations from the automatic Swiss Measurement Network SwissMetNet (SMN) station at MER and REM observations from MEE facing the Brünig Pass. «*

- l. 161: add one sentence explaining why even though some measurements (ground and remote sensing elsewhere) are assimilated into the model, a later comparison of the model to the same type of measurements still makes sense

  *One of the reasons to compare observed and modeled data at MER is that the MER ground observations are only assimilated if the difference with the modeled data is inside a given threshold. Consequently, the next sentence was modified: "Anyhow, the observations considered as too far from the modeled data are rejected during the assimilation phase, so that a comparison between the observed and modeled data at MER allows making assumption on the models' skills. "*

  *Anyhow, even if the observations were always assimilated, it doesn't ensure to have the same observed and modeled data. Therefore a further sentence was added at the beginning of the results' section to further emphasize this point: "Even if SNM/MER surface observations are assimilated by KENDA-1, the comparison of the modeled and observed data at MER and MEE allows evaluating the impact of the assimilation at MER."*

  l. 177-178: check syntax

*The sentence was modified: «During the Meiringen campaign the training dataset of Payerne on the Swiss plateau was used, leading to potential further bias. «*

- l. 227: can you add the mean ridge height to the plot as horizontal line?

  *The mean ridge height was added to Fig. 3 and 6 that are now merged into Fig. 2.*

- l. 228: it is not simple to understand where the "T rise shortly at sunset" is visible in Fig. 3

  *It is principally visible in April and to a lesser extent in May, but it does not correspond to a main feature and does not lead to an interesting conclusion. We decided then to delete this sentence to shorten the paper and improve the streamline.*

- l. 232: there is no figure D1b (also at l. 234). What are "standard" values?

  *Yes, both positive and negative gradients refer to the same figure (D1). This was correct in the manuscript. Lute and Abatzoglou (2021) did not present standard values for evening T gradient, so that we delete this part of the sentence.*

- l. 233: "(Fig. 3a) near the ground (590-1000 m) for all months..."

  *Done*

- l. 235: "apart" or "onwards"? Please, correct all references

  *Done here and in the whole manuscript*

- Fig. 3: explain the dashed vertical lines in the caption. Change the x marks to, e.g., 6h or submultiple of 24h. Is Fig. 3b mentioned anywhere in the text? Maybe it should be moved next to Fig. 6?

  *As explained now in the figure caption, the dashed lines correspond to sunrise and sunset. Figure 3b was removed and replaced by Fig. 6.*

- l. 251: "station altitude" instead of "real topography"

  *Done*

- l. 257: what does "difference in the effect" stand for?

  *There is effectively not much difference in the effect of the ELR correction at both stations so that this sentence was removed.*

- Fig. 4: the dashed vertical lines should be explained in the figure caption

  *Yes, it's done: "The dotted and dashed lines correspond to the median and the mean, respectively."*

- Fig. 5: T differences between ... and SMN/MER" (subtraction is non-commutative). Also, is "MERn" a typo?

  *The use of "different" in this context if a French mistake. In French, "different" can signify "various" and has nothing to do with a subtraction. The world is not necessary and was suppressed. MERn is a typo.*

- l. 279: explain why "705" m

  *The two mentioned altitudes are now described: "The T profile comparisons describes the differences between the MWR/MEE retrieved T profiles and KENDA-1/MEE modelled T between the first level of the model at 705 m and the top of the good MWR measurements at 2500 m."*

- l. 284: correct missing reference ("??") and similar occurrences

  *All the references were checked and corrected.*

- l. 289: "presents slightly"

  *Done*

- l. 294: "shows"

  *The subject of this verb is plural, so that the third person singular cannot be applied.*

- Fig. 7b: check sign of deltaT. Is it T_SMN  - T_MER (>0) or the opposite (as in the main text)? Use submultiples of 24h on the time axis. Mention why there are data gaps

  *Delta T refers to differences between the ground T and the minimal T of the inversion. They are therefore computed always from the same type of observations.*

  *Multiples of 6h are now used. There is no MWR data from November to January and in August since the MWR was installed current January and not measuring during whole August as stated in the experimental part. No data are represented in Fig. 7b (presently Fig. 5b), when no T inversions were observed.*

- l. 319: "or not of the values" please check this sentence

  *This sentence was modified and merged with the previous one:" During these nights, differences between the model's first guess and observations are mainly around 5 °C and can reach 10 °C in extreme cases (results not shown), so that observations are rejected due to differences exceeding the predefined threshold based on the ensemble first guess, its spread and the observation error."*

- l. 326: what is a "Bise" wind?

  *Bise is a cold dry northern wind in Switzerland. This is now specified in the text.*

- l. 349: "1500 m", is this altitude limit so clear?

  *No, this limit is not clear, moreover if we consider that the E winds can provide from both down-valley and foehn winds. The reference to the vertical extent was removed.*

- l. 355: "between 1300 and 1700 m", not clear from the figure

  *Yes, we do agree that the contribution from N wind is not clearly visible in this plot, so that the sentence was removed.*

- Fig. 8: how do you deal with direction in calm wind conditions? Why are the white boxes (data gaps) similar for all sources? Clearly write the difference between plots b) and c) even in the figure, not only in the caption

  *All the analysis of wind directions were only performed on observations with wind speeds>2 km/h to avoid overinterpretation of the direction during calm winds. This is now mentioned in the experimental section.*

  *White boxes are similar in observations and KENDA-1 data because the modeled data were restricted to time with observations to allow a comparison between the measurements and the model. The indication of the type of data and the site are now also given in the figure.*

- Fig. 9: explain what positive and negative speeds mean in the caption. Clearly state what interpolation/smoothing technique was used in the contour plots

  *Positive speeds correspond to up valley wind as specified in the color of the arrows in Fig. 1. This information is now given under 3.2.2 in the manuscript: "For this analysis, the positive speeds correspond to up valley wind (see Fig. 1 and Fig 1B) and to northern wind from the Brünig Pass (see Fig. 1B) for along and across valley winds, respectively."*

  *The method to interpolate is now clearly station in the figure caption: "a) Diurnal cycle of the hourly T inversion frequency between T at SMN/MER (589 m) and FEDRO/BRU (998 m) ground stations, at the lowest level (640 and 705 m, respectively) and 1000 m of MWR/MEE and KENDA-1/MEE profiles. The 1_D measured values were interpolated using a linear interpolation with 10 m spaced vectors. b) Mean $\Delta$T for the time where an inversion is detected. Sunrise and sunset are represented by dotted lines."*

- l. 403-404: is 800 m "ground level"?

  *800 m corresponds to the DWL first level. 800 m was however falsely reported since the first level is 200 a.g.l. corresponding to 775 m in MEE. This altitude was modified in the whole manuscript.*

- l. 405: "reduced" compared to what?

  *The reduced wind speeds are compared to the ground SMN/MER observations. This is now stated in the manuscript:" The onset of the up valley winds occurs with the same delay to sunrise (~4 h)*

*during the summer months but their speed is of reduced maximum amplitude (10-15 km/h) than at SMN/MER"*

- l. 408: "occurred"

*Done*

- l. 416: "ridge height"

*Done*

- Fig. 10: y label should be "DWL/MEE" instead of "DWL/MER"

*Done*

- l. 453: "reported at 200 m", explain why in the instrument description

*The fact that the DWL first level is at 200 m a.g.l. is now explained in the experimental section. The up valley wind is only measured in DWL/MEE on the 10 July not because of an instrumental problem, but because other types of winds are measured during the rest of the chosen period.*

- Fig. 11: add "FEB"

*Done*

- l. 467-470: unclear, please rephrase. Maybe clearly state that MER is more sheltered from the N wind than MEE? Can BRZ be influenced by katabatic nighttime winds from the slope?

*The sentence was simplified to make a clear causality between the wind observed at the Brünig Pass and its influence at MEE. The next sentences were also slightly modified to underline that the wind from the Brünig pass largely influence the diurnal cycle of the wind direction at BRZ but only slightly at MER:" These winds from the Brünig Pass explains first the N wind observed in MEE during the afternoon, the early evening and even sometimes in the morning (e.g. on the 11 of July). Second, they also strongly influence the diurnal cycle at BRZ leading to the onset of down valley winds in the early afternoon or even by suppressing up valley winds (July, 11). Finally, their influence at MER is however weak with only a slight shift of the wind direction towards N in the late afternoon."*

*The winds from the Brünig appear already in the morning during these clear and warm days in July. Since a complete T analysis at all the ground stations were not performed, these down slope winds are not necessarily katabatic, but could be qualified of drainage wind following the AMS glossary of Meterology. But a complete analysis of the causes of the diurnal cycle of the down slope winds from the Brünig is beyond the scope of this paper.*

- Fig. 12: is BRZ really representative of thermal circulation? Also, use correct colours in the legend of Fig. 12b. Add the indication "(orange)" after "sunshine amount" in the caption

*Fig. 12 and the related section now entitled "Heterogenity of wind patterns in the Haslital valley" aims to highlight that the standard thermal valley wind pattern is not spatially homogeneous even at distances of some km. In that sense, BRZ is not representative of a standard thermal valley circulation (similarly to MER) but its wind pattern is anyhow produced by thermal effect in the complex topography of the Haslital valley.*

*The colors of the legend of Fig 12 b were corrected and the indication that the sunshine amount is given in orange was added.*

- l. 486-487: I cannot fully understand the meaning of this sentence

*This sentence notes that the KENDA-1/MEE grid cell comprises part of the slope leading to the Brünig Pass and should consequently monitor slope winds better than the DWL/MEE situated in the middle of the valley. Anyhow this was already mentioned at several places in the manuscript and brings no important information here. The sentence was then deleted to shorten the manuscript.*

- l. 490: "up to 1500 m during all other months", not so clear in summer. The N component up to 2500 m is difficult to separate

*Yes, this sentence aims to describe only the strongest N winds with speed > 20 km/h. The manuscript was modified to highlight that N wind can also extend up to 2500 m, but with a weaker speed.*

- l. 503: "Foehn is"

*Done*

- l. 512: "measured wind speed"

*Done*

- l. 516: "better agreement … (not shown)", are your referring to the difference between the two red boxes in Fig. 14b? Is this difference significant?

*We do agree that this difference is not clearly significant if we refer to both Fig. 14 a and b. The sentence was removed.*

- Fig. 14: explain the number row at the top of the plots. Correct the intervals on the x axis in Fig. 14b. Correct order of factors in caption (difference should be with reference to SMN/MER)

*The numbers are now explained in the figure caption: "and n is the number of cases in each of the categories". The intervals on the x axis and the legend were also corrected.*

- l. 521: "(mettre…)", correct typo

*Done*

- l. 549: why do you mention three different foehn types, when the selected cases are all of deep foehn?

  *We mentioned them because they exist. However, since only one type of foehn was measured, this description was removed.*

- Fig. 15: correct x labels for 2nd and 3rd columns

  *Labels were corrected.*

- l. 558-560: this sentence is a bit general. Maybe rephrase so that it can be understood that the focus is on "atmosphere over complex topography"

  *This quite general sentence is now more focused on the particularities of the modelling of MoBL and was move under 4 as an introduction to the discussion section:" Complex topography, landscape heterogeneity and specific thermal wind regimes challenge the models' spatial and temporal resolutions, their performances in data assimilation and the parametrization of multi-scale processes."*

- l. 560-561: isn't it the same over flat terrain?

  *Yes it is also the case in heterogeneous terrain and various surface types. The two first sentences of 4.1 were modified and used as an introduction to the discussion section (see also previous remark).*

- l. 569: "As such", please check syntax

  *The expression was modified to "In this regard,…"*

- l. 570-576: most of it has already been written. Please, shorten this part

  *This paragraph was shortened as recommended: "Moreover, the model cell over MEE overlaps the slope towards the Brünig Pass, so that KENDA-1/MEE reports an average of winds from the Brünig Pass and in the Haslital. DWL/MEE, on the other hand, only observes winds in the middle of the Haslital. Consequently, the differences between the modeled T/wind averaged values and the observations cannot be only considered as model errors."*

- Sect. 4: use the space here to anticipate the structure of the discussion, especially Sects. 4.2 and 4.3

  *A first paragraph now summarizes the challenges of MoBL in complex terrain and describes the discussed points: "Complex topography, landscape heterogeneity and specific thermal wind regimes challenge the models' spatial and temporal resolutions, their performances in data assimilation and the parameterization of multi-scale processes. The discussion will consequently focus on three points, the specificity of the terrain around the campaign site, the comparison of the observed wind and T profiles with previous observations in the Alps and the model performances in Meiringen."*

- l. 585-589: this can be shortened by writing that the forecast skills were not the focus of the paper

  *The § was shorten as suggested: "Finally, this study is based on monthly median values, so that the averaging artifacts have to be considered, e.g. for the analysis of maximum wind speed, the onset time of valley wind or wind directions. In that sense, this analysis focused on climatology and not on the forecast skills of COSMO-1."*

- l. 596: "pairs", do you mean at different altitudes?

  *Yes, it is now specified in the text*

- l. 607: "months"

  *Done*

- Sect. 4.2.2: too much detail. Do not repeat all results, just mention the most important similarities and differences

  *The section 4.2.2 was largely shortened, mostly the second § containing the largest number of repetitions about the results in the Haslital.*

- l. 684: "other study" by whom?

  *This not published study was performed by one of the co-author, D. Leuenberger. It is now specified in the manuscript.*

- l. 722: are winds "well modeled" despite "the onset is predicted with a larger inaccuracy" (l. 724)? Does the sentence only stand for the wind velocity?

  *Both the qualification of « very good performance" and "larger inaccuracy" are overrated and have been removed since they are not based on any similar performance indicators.*

- Table A1: is the last column a % of a %? Write the year the anomaly refers to in the caption

  *Yes, the penultimate column is the percentage of maximum sunshine and the last column th corresponding percentage of the norm. The years considered for the climatological norm (1991-2020) are already given in the table caption.*

- Figure A1: a y label for the sunshine duration is missing

  *Done*

- Figure B1: Explain the (black and coloured) arrows in the figure. Caption: "downloaded"

  Fig B1 was removed. The arrows are explained in Fig. 1 that takes over part of the information of Fig. B1 and B2

- Figure B2: red/blue lines are too thin

  Fig. B2 is now part of Fig. 1 and the ridges' heights are not larger.

- Appendix D, E, F, G, H, I, J: titles should precede the figures

  The appendix was converted into a supplement and no titles subsist.

- Figures E1 and E2: y axis label (and measurement unit) is missing. Explain the two lines of plots

  Fig E2 was removed. Y label and unit are now reported. The lines of the plots are already explained in the figure caption.

- Figure F1: add a legend for the purple, blue and red lines. Where are the dashed lines mentioned in the caption?

  Fig. F1 was removed.

- Figure H1: "sunrise"

  The dashed lines are now described in the figure caption.

- Figure I1: the second colour key should report "°", not "°C"

  *Done*

---

## Referee Report (RR1)

**Review of 'Comparison of temperature and wind between ground-based remote sensing observations and NWP model profiles in complex topography: the Meiringen campaign' by Bugnard et al.submitted to Atmospheric Measurement Techniques**

March 27, 2024

The study investigates wind and temperature in the boundary layer of the Haslital in Switzerland using ground-based remote sensing and in situ instruments deployed during the Meiringen campaign from October 2021 to August 2022. It compares the observations of a microwave radiometer, Doppler lidar, and surface meteorological station to the COSMO-1E model analysis (KENDA-1). The valley is rather narrow with 1.5 km width and one of the sites is affected by the low altitude Brünigpass to the north. By means of monthly composites, nighttime inversion and thermally driven wind systems were detected. Large model errors were found for nighttime temperatures on the average as well as on individual days.

This review is of the revised submission of the manuscript and is provided in view of the interactive public discussion. In the revised version, many of the comments of previous reviewers are addressed. For example, the manuscript was shortened and the appendix was moved to a supplemental. The gathered data provide a great opportunity to investigate the boundary layer conditions in a mid-sized Alpine valley and to evaluate the performance of KENDA-1. In my opinion, the issues with partial data assimilation at MER is of particular interest and may be relevant for other locations as well, leading to potential forecast improvements in complex terrain. While I believe that the manuscript is improved compared to the original version, I still have some major concerns about some of the aspects of the study, some new, some already raised by the previous reviewers. For example, the authors strongly focus on thermally driven flows, however, they don't distinguish between thermally and synoptically driven conditions when computing the monthly composites, which, in my opinion, masks many of the relevant features. KENDA-1 has a 1.1 km horizontal resolution, while the Haslital is 1.5 km wide. The authors discuss the difference in valley floor height between reality and the model, but they do not discuss the terrain in the model at all, such as shape and depth of the valley, and to what extent the Brünigpass is resolved or how many grid points are available in the valley. I provide more details on these two aspects as well as many others in my comments below. Overall, I think the manuscripts includes too much description and not enough analysis in its present form. I suggest that the authors consider my comments before the manuscript can be accepted for publication.

**1 General comments**

1. The manuscripts contains a very detailed description of conditions in the results section, but hardly any investigation of the relevant processes. It first presents a lengthy description of the temperature

and wind field using monthly composites with lots of details on values in specific layers etc, followed by a description of a three-day case study when the flow from the Brünigpass affected the conditions in the Haslital and by a description of three foehn events. In my opinion, it currently is a mix between a campaign overview and some preliminary results. It is very descriptive without a clear story line. If the case studies are meant as teasers (for future manuscripts) they could be much shorter. No in-depth analysis of the case studies is presented and there are many open question, which arise to the reader. For example, why should a warm upvalley wind from the Sarneraatal descend into the Haslital? An in-depth analysis would probably be beyond the scope of the manuscript, but such contradictions to theory and open question should at least be mentioned. Also, I don't think that there is enough observational proof presented to conclude that the wind conditions in the Haslital are affected by the flow from the Brünigpass or that there is a vortex present. These findings should be formulated as hypotheses. A more detailed study, possibly using the 3D model output is needed to provide strong evidence for this and to investigate the physical processes.

2. The analysis focuses very much on thermally driven flows, but the monthly composites are not separated for days that are dominated by large-scale conditions (frontal passages, foehn) or are affected by clouds (unfavorable for thermally driven flows) or are dominated by fair weather conditions (favorable for thermally driven flows). Computing composites over all days masks the signal of thermally driven flows (which primarily occur under fair-weather conditions, as correctly mentioned in l. 48). The authors still interpret the weak signals in wind in the composites and conclude that e.g. no thermally driven winds were observed in December and January. However, they might still be there just not in the monthly composites. Furthermore, including everything in the averages makes the comparison to other studies (Discussion section), in which days where filtered for thermally driven flows, not very meaningful. For example, conclusions on thermally driven flows are drawn from the composites of along valley wind component in Figs. 7 and 8. There are several features that are not typical at all for thermally driven flows and differ from theory, like the persistence of upvalley flow near ridge height and the decrease in downvalley wind strength during the night. This could be a result from the projection of the flow on along valley wind direction or a result of sampling over all types of conditions. Either way this discrepancies need to be discussed and attempted to be explained.

Monthly plots for wind direction are separated using a wind speed threshold which seems a bit arbitrary and needs more justification. It is currently not clear at what height this threshold must be met and during what time period. Also, why do the authors not distinguish days for the other composites (along-valley and across-valley wind speed, temperature) as well? Since the focus is on thermally driven flows, the analysis could also be restricted to composites of days with thermally driven flows. This would reduce the number of panels and streamline the manuscript. A better and more physical way to distinguish days with thermally driven flows from days dominated by synoptic flows could be by looking for a wind direction reversal twice per day and/or by considering cloudiness.

3. Composite plots for wind (Figs. 6, 8, 9) are presented with 10 panels per row, which makes it impossible to see any details on the time axis and to follow the detailed description for specific hours. I highly suggest to think about another way of presenting these composites plots. I understand the desire to reduce the number of figures, but this should not be done at the cost of visibility.

4. A discussion of the model terrain is needed, which goes beyond the difference in valley floor height. For the capability of the model to simulate terrain induced features, the shape and depth of the valley and the numbers of grid points is most relevant. This is not discussed at all. The Haslital is 1.5 km wide and KENDA-1 has a 1.1 km resolution. What about the Sarnaraatal? Is the Brünigpass even resolved in the model? How are the lakes resolved? How much terrain smoothing is done? In l. 74, it is stated that the grid resolution should be about 10 to 20 times higher than the relevant topographic scale to fully capture the different exchange processes. This aspect is very important when interpreting

KENDA-1 results, but is currently not considered at all.

5. Some sentences are not very clear and perhaps the authors should consider using a professional editing service to remove these language issues. Examples include, l. 202-203: Layer with higher T develops gradually from sunset to sunrise to reach monthly-related maximal T and height. l. 250-251: Globally, the measured MWR/MEE first level T are closer to the SMN/MER T than the modeled T. l. 256-257: The analysis of the negative ground T difference between MER at 590 m and BRU at 998 m (horizontal distance = 3.7 km) shows that near ground T inversions are common during the night for all months in the study. l. 269-271: The missed T inversions by KENDA-1/MEE leads to both its important overestimation of the T at ground level (Fig. 4) and its slight T underestimation between 850-1200 m (Fig. S5 for detailed examples). l. 296-297: ..., whereas flows from W-NW are measured in the rest of the profile concerned by up valley winds (see further explanation in sect. 3.3). l. 303-305: The good KENDA-1/MEE performances comprise first the influence of the foehn up to 2500 m (ws >20 km/h) as well as the presence of valley wind below 1200 m (ws <20 km/h) in March. l. 345-347: Thermally induced wind height increases with temperature, reaching 1000 m in February, 1800 m in May and up to 2000-2200 m in July and August. l: 371-373: This suggests a circular motion with North updraft winds (median vertical velocity of 1 km/h) that cross the valley at a low altitude, rise against the north facing slope and come back at higher altitude with a South downdraft component l: 599-600: This is especially the case at the end of March, when enhanced night time radiative cooling and important global solar radiation form strong inversions.

**2    Specific comments**

1. l. 64: The classic work of Whiteman and Doran (1993) could be referenced here.

2. l. 73-74: The classic work of Skamarock (2004) Skamarock and Klemp (2008) could be referenced here.

3. l. 84-87: Very complicated sentence. Please rephrase.

4. l. 88: Is the "first objective of the campaign to study the seasonal and diurnal cycles" or is this the first objective of **this study**?

5. l. 93: I think a short introduction on what KENDA-1 is and why its evaluation is important needs to be added to the introduction (possibly where NWP models are mentioned).

6. l. 94: I find the acronyms MER for Meiringen and MEE for Unterbach a bit unfortunate, since they are very similar and easy to mix. Maybe just a matter of taste.

7. l. 104: Why are the times not given in local time? This is advantageous for a study that focuses on thermally driven diurnal wind systems. At a minimum the time difference between local time and UTC needs to be given.

8. l. 105-106: The temporal resolution of the observations is higher than 1 hour. How is that considered? Are these values averaged to 1h values before computing monthly composites or are instantaneous hourly values used? When computing the composites how is data availability considered? Are model data only plotted for times and heights where observational data are available?

9. l. 110-112: The Gadmertal and Rychenbachtal are not shown in the map and not relevant for the study. Please remove or modify the map in Fig. 1 to include them. This was already a comment of a previous reviewer.

10. Fig. 1: Fig. 1b is impossible to see. Please increase line width, axis labels, and legend to make it readable. I suggest flipping the x-axis, so that the location of the sites is consistent with 1a and 1c. Why are the dots in Fig. 1b not at the valley floor? Are the station heights in Fig. 1b from the model? In Fig. 1c, labels along VW and across VW are very hard to read.

11. Section 2.2: This section sounds like COSMO-1E is used: "The NWP model used in the study is the limited-area non-hydrostatic atmospheric model from the Consortium for Small-Scale Modeling Model (COSMO)". I think it would be better to make clear from the beginning that the analysis KENDA-1 is used, which uses 1-h forecasts from COSMO-1D and observations. It would also be helpful to mention here how close to the investigation area the different observations are assimilated. SMN is assimilated at MER (this should already be mentioned here). But what about the sondes and profilers? How far away are they from the investigation area?

12. l. 141ff: Where are the MER and MEE model grid cells located in the model terrain? Are they at the valley floor in the model terrain? This is more meaningful than comparing the grid cells to the real terrain, which the model does not know.

13. l. 150ff: I think the fact that MER data are assimilated in KENDA-1 is critical for the results and needs more attention. If the observations are assimilated during certains conditions (daytime) and not during others (nighttime inversions) this may affect the error distribution. In the response to reviewer 2, the authors included an example demonstrating how often MER data are rejected and argue that 'One of the reasons to compare observed and modeled data at MER is that the MER ground observations are only assimilated if the difference with the modeled data is inside a given threshold." This information as well as the threshold would be helpful to include in the manuscript. Furthermore, I think including statistics on how often MER data are assimilated during the whole campaign and at what times would be most helpful and interesting.

14. Section 2.3.2: Was the RPG neural network retrieval used? As far as I know, HATPRO-G5 comes with a surface met sensor. Are this information used in the retrieval? If not, this surface met measurements at MEE could be an additional observational source to evaluate KENDA-1. Why was this not used? What does 'line of sight of about 10 km' mean? Is this the line of sight in the direction of the low elevation scans? In which direction was this performed (up or down the valley)? Temperature biases in microwave radiometer retrieval are highly instrument dependent, arising from spectral biases or liquid nitrogen calibration. The cold bias found by Hervo et al. (2021) is not necessarily transferable. 'The instrument at MER ..' isn't the MWR installed at MEE?

15. Section 2.3.3: The scan strategy of the Doppler lidar is not clear. It sounds like the lidar did a combianation of vertical stare (every 10 min for two minutes) and DBS (every 5 min). What did it do in between? The scan in Fig. S9 looks like it is an RHI scan, this needs to be mentioned here. According to the Vaisala data sheet the maximum range for Wincube 100S is 3 km. Was this system modified to reach 12 km maximum?

16. l. 195: Heavy **liquid** precipiation ?

17. l. 189ff: None of the information on the general conditions during the campaign are considered when presenting the results on temperature and wind. How do the boundary layer conditions differ over snow covered vs. snow free ground? Does KENDA-1 performance depend on snow cover? Because of the monthly composite and snow cover lasting until mid-December this is all mixed together. How do the heat waves reflect in boundary layer conditions and KENDA-1 performance? If the information on general conditions is included, it should be considered when desribing the results.

18. l. 202-209: The description of temperature changes should be related to the boundary layer evolution. 'Layer with higher T' is the daytime boundary layer. Perhaps the authors could compute convective

boundary layer height using the parcel method. I do not see the value of presenting temporal T gradients (Fig. S4), especially not for monthly composites. It is not surprising that T increases during the day and decreases in the evening. I suggest removing this to streamline the manuscript. In Fig. S4 temporal gradients are shown, in the text vertical T gradients are discussed (C/km).

19. l. 213ff: Fig. 2b show the bias. Did the authors also investigate mean absolute error or root mean square error? Is mean absolute error also small when biases are small or are the small biases related to averaging artefacts (days with very large positive biases vs days with very large negative biases)? How are observed and modeled profiles compared? Are modeled heights interpolated to observed heights or vice versa? Since the temporal resolution of the observations is higher than KENDA-1 are they averaged before comparing to the hourly model output? This information should probably be included in Sect. 2. I assume the authors refer to the study of Hervo et al. (2021) when saying 'The cold bias between the MWR and the radio sounding could however suggests a larger error of KENDA-1.' As mentioned before, biases are instrument specific and I do not think this is a valid conclusion.

20. l. 223: Please explain why observations at MEE are compared to MER (here and later in the manuscript). These are different sites affected by different physical processes.

21. Fig. 2: Please add ridge height to panel a). To better see stablity it would be helpful to add potential temperature isentropes. For example, observed potential temperature isentropes could be added to a) and simulated potential temperature isentropes could be added to b).

22. Fig. 3: Please enhance label and legend size.

23. l. 234-239: What is the justification to use a mean environmental lapse rate of -6.5°C/km during nighttime temperature inversions? The height difference between the model grid point and real world is only one challenge when comparing observations to model output. The terrain in the model needs to be considered as well, i.e. is the grid point at the model valley floor or on the slope? What is the valley depth, etc. (also see my general comment on this).

24. Fig. 4: Are the number of samples the same for all time series, that is, are only time stamps used when all observations and model output was available?

25. l. 252ff: Using ground stations as pseudo-profiles can be affected by local impacts (e.g. solar heating during daytime, slope winds), even more since BRU is located at a pass. A brief discussion on potential error sources should be included (Whiteman and Hoch, 2014). How are temperature inversion determined? Just by computing the temperature difference between the upper and lower height and detecting negative difference? Was a minimum absolute value required? The authors compare temperature difference over layers which are not exactly the same depth and which introduce additional uncertainties. Why not compute gradients per fixed height interval? It is interesting that even though March was dominated by foehn events there is a very clear diurnal cycle in inversion frequency and amplitude. This should be discussed. In Fig. S5, there is absolutely no inversion visible in the KENDA-1 profiles. This is very strange and deserves further discussion, in my opinion. Furthermore, it has to be considered that the MWR profiles also have uncertainties and the smoothed shape of the profile may lead to an overestimation of inversion amplitude.

26. Fig. 5: I find the amplitude plots confusing. I assume that sample size is not constant during the day and some of the spikes during summer daytime in the MWR data are probably caused by averaging over very few days. Perhaps a minimum sample size should be required for showing the amplitudes. The grid lines do not fit to the tick labels on the x-axis. What is meant by '10m spaced vectors'?

27. l. 272ff: As mentioned before, I find this issue with assimilation very interesting. Do the large differences between the observations and the 1h forecasts mean that there is no inversion in the model? What about later forecast hours? Is this maybe related to spin up time?

28. l. 288-289: Wind speed values at which height, time, and station are used to distinguish between days dominated by thermally driven and synoptically driven flows? As mentioned in my general comment, it would be more physical to inspect days for the typical reversal of wind direction twice a day instead of looking at thresholds.

29. l. 295-297: Do the authors have an explanation why the upvalley wind shifts with height? Could this be a result of averaging over a variety of conditions?

30. l. 297ff: What is the purpose of providing this amount of detail on synoptic flows? Consider removing or shortening to streamline the manuscript.

31. Fig. 6: Please add ridge height to the plots. It is really hard to see anything in the panels. Consider limiting y-axis to 2500 m (like for T plots) and rearranging the figures (maybe remove synoptically driven days). What is the temporal resolution of the DWL plots? The same as for the model? Were the winds averaged to 1 hour before computing the monthly composites? What is the point of showing panel c? It is not discussed here.

32. l. 307ff: I don't see a good agreement in November. Also there are quite large difference between the observations and the model in summer/spring. The features described here (e.g. N flows from Brünigpass) are hardly visible in Fig. 6b in its current presentation. The conclusion 'This feature is mostly caused by the KENDA-1/MEE cell overlapping the slope towards the Brünig Pass so that winds at the junction between Haslital and Sarneraatal can influence the median modeled wind compounds.' is not valid in my opinion without a detailed analysis of the model terrain.

33. l. 324-325: Typically, downvalley winds gain in strength throughout the night since the driving horizontal temperature and pressure gradients strengthen. Do the authors have any explanations why this is different here? Could this be a result of clouds forming during the night or a sampling issue? Please discuss this contradiction to theory.

34. l. 328-334: The described upvalley wind characteristics are not typical. For example, why should the upvalley wind near the surface be stronger and more regular than at 200 m above the ground. This should be discussed. Could this be related to sampling or an artifact of the projection to along valley direction? Would the valley wind system show more typical characteristics when synoptically dominated winds were excluded from the composites? The difference in onset time of downvalley wind at MER and MEE could be related to the Brünigpass. However, this needs to be formulated as hypothesis without sound observational evidence (which is not given in Sect. 3.3).

35. l. 335-336: The statement that turbulence is leading to daytime varying wind direction is not obvious and needs to be supported by observational evidence or removed.

36. Fig. 7: Please enlarge axis labels. Replace MWR/MEE in caption with DWR/MEE. What values are plotted? Are these hourly values or monthly values? If monthly why not plot hourly values to show day to day variability?

37. l. 338: 7b instead of 7c.

38. l. 340-343: This atypical behavior of the valley wind (upvalley wind during the night, upvalley wind during all day in winter) in KENDA-1 is possibly a result of averaging over all types of conditions. It is hence not clear if KENDA-1 struggles with thermally driven flows or channeling events or on clear or cloudy days. Filtering and focusing on specific conditions would be beneficial to learn more about model deficiencies.

39. l. 344-354: In my opinion, it is not valid to draw conclusions for up- and downvalley winds and on impacts of synoptic flows from the monthly composites, since the composites are most likely strongly

affected by synoptic winds and clouds. To draw meaningful conclusions on up- and downvalley winds, the days need to be filtered for conditions favorable for thermally driven winds. Also, some of the aspects (decrease of downvalley wind speed with height) could be a result of projecting on the valley axis and should be investigated. Without investigating other factors (clouds, synoptics, sampling size, etc) it is not valid to attribute varying wind direction during daytime to turbulence.

40. Fig. 8: How are the composites computed? Are KENDA-1 data only used when observations are available and valid? If not, the comparison is not fair. Are white gaps in panel a) due to small wind speed or missing data? A different color should be chosen for missing data (e.g. grey).

41. l. 366: What does 'data are scarce' mean? Shouldn't the sample size for across valley wind be the same as for along valley wind?

42. l. 366-367: The northerly flows in January and February are not clearly visible.

43. l. 370: What are north-facing slope winds? Do the authors mean downslope winds from the slopes north of the valley (they are south-facing)? For clarification, it would be helpful to repeat the colors used in Fig. 1 to distinguish southerly and northerly cross valley flow.

44. l. 371-374: In my opinion there is not enough evidence for a cross valley vortex from the example RHI plot in Fig. S9. I can see downslope and upslope components, but no closed circulation.

45. l. 375: What color are the winds that descend from Brünigpass? Are they red? Please specify.

46. l. 380-383: Why is the along valley wind mentioned here? This section is about cross valley. It needs appropriate context and reference.

47. Fig. 9: How are the composites computed? Are KENDA-1 data only used when observations are available and valid? If not the comparison is not fair. Are white gaps in panel a) due to small wind speed or missing data? A different color should be chosen for missing data (e.g. grey). What is the sample size at each point (can some of the noise be explained by varying sample size)?

48. l. 394: On this fair weather day, wind speeds of 25-30km/h are reported. How does this fit to the filter of 20km/h to distinguish thermally from synoptically driven days?

49. l. 399-401: The outflow from Brünigpass cannot be thermally driven. Why should warm air during the day descend from the pass to MEE? If there is upvalley wind in Sarneraatal that reaches the pass and descends on the south side of Brünigpass there must be a dynamic effect driving this (wave, etc). Please explain.

50. l. 409-418: This whole paragraph is based on Fig. S8. If this figure is so important, it needs to be included in the manuscript.

51. l. 419ff: It is essential to include the model terrain in this analysis to understand how the model sees the Sarneraatal and the pass.

52. Fig. 10: Both panels should have the same y-axis range.

53. l. 439-441: Specify where clear weather conditions can be expected. Describe the foehn characteristics at MER (direction, over which ridge it is coming). What stations are used to compute the foehn index?

54. l. 442: No June episode shown.

55. l. 445 and Fig. 11: Unless foehn starts always at the same time of the day, the composites should be shown relative to foehn onset and not for hour of the day.

56. l. 453ff: This whole paragraph is again based on Fig. S11 and S12 from the supplement. If this is discussed in so much detail, it needs to be included. However, related to my comment on adding more focus to the paper, I think the whole discussion on foehn should be much shortened or even removed. I also cannot see a T gradient in the types of plots in the supplement.

57. l. 465-466: I don't see the point in comparing different heights at MEE and MER.

58. l. 467-484: I think this description of wind is way too detailed and distracting and could be much shortened or removed.

59. l. 485-494: Given that KENDA-1 provides 3D output, the foehn cases could instead be investigated in KENDA-1 to understand the spatial differences and the model errors during foehn events.

60. Discussion: The discussion section is too long in my opinion and should be more focused. For example, a lengthy comparison to inversions and thermally driven flows in other studies is shown, but given that the composites in the present study are not filtered for thermally driven flows this comparison is not very meaningful. The evaluation of KENDA-1 was done visually based on time-height sections and not by computing model skills. This would have been more meaningful, instead of the descriptive comparison of the model to the observations. I think saying that 'KENDA-1 proposes good monthly median values' (l. 662) and 'Despite the complex topography around MER and the induced elevation bias, the modeled climatology of ground T is comparable to standard verification results' (l. 595-597) is hence not sufficiently supported by the analysis.

61. l. 574-584: The vortex in the Inn Valley was caused by the valley curvature (Babić et al., 2021). This means that the mechanisms here are likely not comparable and caution is advised.

62. l. 585-587: No skills are computed for KENDA-1 and an objective verification of model skills was not done. This conclusion is hence not supported by the presented results.

63. l. 595-597: The reference to standard verification results is confusing and needs to be explained and justified by results.

64. l. 599-600: Since monthly composites are shown this statement is not supported by results.

65. l. 605-617: RH depends on T. Thus, a warm bias leads to a dry RH bias. The statement on RH does not provide additional information. Bias in terms of specific humidity would be more meaningful. What does 'artifacts from the NWP can be expected under conditions favorable to surface T-inversion' mean? The statement 'Finally, the differences with observations can also originate from a modeled ongoing turbulent mixing whereas in reality a cold pool with a full or partial decoupling from the above flow is present in the valley.' is not supported by results.

66. l. 618ff: MWR liquid nitrogen calibration plays a role in MWR profiles biases and should be discussed. Average differences are discussed, but what about individual profiles?

67. Conclusions: Since many readers only read the Summary some basic information on sites and data should be repeated. I don't think that all conclusions are sufficiently supported by the results (e.g. l. 688-690, l. 693-695). I furthermore do not think it is fair to say that the study 'deepens our consensual knowledge about atmospheric phenomena in complex topography'. It is mostly a description of conditions without any in depth investigation of processes.

**References**

Babić, N., Adler, B., Gohm, A., Kalthoff, N., Haid, M., Lehner, M., Ladstätter, P., and Rotach, M. W.: Cross-valley vortices in the Inn valley, Austria: Structure, evolution and governing force imbalances, Quarterly Journal of the Royal Meteorological Society, 147, 3835–3861, 2021.

Skamarock, W. C.: Evaluating mesoscale NWP models using kinetic energy spectra, Mon. Wea. Rev., 132, 3019–3032, doi:10.1175/MWR2830.1, URL `http://journals.ametsoc.org/doi/10.1175/MWR2830.1`, 2004.

Skamarock, W. C. and Klemp, J. B.: A time-split nonhydrostatic atmospheric model for weather research and forecasting applications, J. Comput. Phys., 227, 3465–3485, doi:10.1016/j.jcp.2007.01.037, URL `https://linkinghub.elsevier.com/retrieve/pii/S0021999107000459`, 2008.

Whiteman, C. D. and Doran, J. C.: The relationship between overlying synoptic-scale flows and winds within a valley, J. Appl. Meteor., 32, 1669–1682, 1993.

Whiteman, C. D. and Hoch, S. W.: Pseudovertical temperature profiles in a broad valley from lines of temperature sensors on sidewalls, J. Appl. Meteor. Climatol., 53, 2430–2437, doi:10.1175/JAMC-D-14-0177.1, 2014.

---

## Referee Report (RR2)

Second review of egusphere-2023-1961

*'Comparison between ground-based remote sensing observations and NWP model profiles in complex topography: the Meiringen campaign"*

by Alexandre Bugnard et al.

Paper in review for Atmospheric Measurement Techniques

**Summary**

The manuscript presents a comparison of campaign observations from the Meiringen Campaign in a narrow Swiss Alpine valley with the high-resolution 1-km KENDA analysis. The comparison focuses on temperature and wind profiles measured by a microwave radiometer and Doppler wind lidar, respectively, for ten months during 2021/2022. It is shown that observed and modeled seasonal climatologies of temperature and wind profiles agree well, although for specific situations, such as for example temperature inversions or foehn events the differences are relatively large. The manuscript also links the complex topography to thermal wind systems and presents cross- and along valley flow systems observed during the campaign period.

The manuscript presents valuable observations from an Alpine site and provides new insights in the quality of the high-resolution analysis in complex terrain and shows examples of how specific terrain-induced flow features can influence the differences between observations and the analysis.

The manuscript has been substantially improved during the first revision phase and the authors have addressed major reviewers' comments, i.e., the general structure was revised and the manuscript was streamlined. I support publication of this manuscript but I have several mostly minor comments that should be clarified and addressed prior to publication.

**1   Comments**

1. KENDA T bias

   I still struggle to fully follow the discussion about KENDA temperature biases discussed in Section 3.1. l. 213 ff states: "The main observed pattern is a general low altitude ($< 1500$ m) T underestimation from KENDA-1/MEE." This cold bias pertains to all hours of the day and all months (Fig. 2b; except June). Subsequently, Fig. 4 shows that "KENDA-1 overestimates the T during nighttime ($+1.5$C) in both cells and underestimates it during the day ($-2$C in MEE and $-1.5$C in MER)." (l. 240 ff). I understand that different data and levels are compared in Fig. 4, however, at nighttime Fig. 4 suggests that KENDA is larger than MWR. The latter is not visible in Fig. 2b. I would ask the authors to elaborate on and clarify the KENDA warm or cold bias, respectively, and streamline this paragraph. If the main reason of the KENDA-SMN bias results from

differences in altitude between KENDA grid box and SMN observation and the frequent presence of inversions, Fig. 4 and the respective text may be misleading. From Fig. S5 I cannot induce if KENDA overall over- or underestimates temperature.

2. Altitude depiction in observation and KENDA data
I would appreciate if the authors could provide information directly in the text, figures, and/or figure captions about the altitude of the used data. It is difficult to remember the altitude of lowest model grid box at MEE/MER as well as of MRW and DWL. I think this would facilitate following the storyline of the manuscript.

3. I would ask the authors to again double-check the manuscript for typos, missing words, and grammar mistakes (e.g., l. 36 "Such inversion**s** are favored in complex topography (Joly and Richard, 2018) and persist**-s** longer in deeper valleys, whereas inversion lifetimes converge to the one over a plain for wide valleys (Colette et al., 2003)."; l. 40 "The quality **of** predictions for", l. 392 "of **a** monthly median value**s**"). Please also consistently adjust the date/time format.

4. l. 6: Please introduce the acronyms COSMO-1E and KENDA-1 as not everyone may be familiar with the terminology.

5. l. 19: "of a model": I would specify this and explicitly mention "KENDA-1".

6. l. 125: "Vertical levels with spacings from 20 m at the surface": What is the height of the lower-most level?

7. l. 176: "from 200 m to 12000 m above ground". Can the DWL measure successfully up to 12 km height?

8. l. 186: "Even if SNM/MER surface observations are assimilated by KENDA-1, the comparison of the modeled and observed data allows evaluating the impact of the assimilation at MER." Please clarify this sentence. From a comparison of the resulting KENDA analysis and the assimilated observations alone, the observation impact cannot be deduced directly, unless first-guess (as mention in l. 272 ll) is available.

9. l. 296: "direction at low altitudes (800-1000 m) is mainly from W-SW": I find this very difficult to see in Fig. 6, among others, as the 800-1000 m layer is very shallow and the colors are not distinct.

10. l. 296: "in the rest of the profile": Please specify.

11. l. 338: "The comparison of the first level of KENDA-1/MER (Fig. 7.c)": Fig. 7c suggests that KENDA-1/MER is shown at 775 m. Which altitude is shown?

12. l. 373: "Plots of radial winds perpendicular to the valley direction clearly present this circulation pattern both in presence of up and down valley winds around sunset (Fig. S9)." Please rephrase, e.g. Figure S9 shows ... .

13. l. 359 ff: "Finally, KENDA-1/MEE overestimates the influence of the synoptic winds leading to the absence of along valley wind in winter replaced by constant slow down valley winds below 1200 m and to higher up valley wind speed in spring and summer." (i) "along valley wind in winter replaced by constant slow down valley"; Do you mean "up valley wind" replaced by down valley wind (as "along valley" wind includes both, up and down valley wind)? I'm not sure if I understand the authors reasoning why the "influence of the synoptic winds" leads to down valley winds in winter and an overestimation of up valley wind in summer in KENDA? Could the authors please explain their reasoning.

14. l. 379 "intense north-facing slope winds": Please rephrase; it is easy to confuse this with "north-facing" "slope winds" (i.e., south to north wind direction).

15. l. 389 ff: Please indicate where this information is shown.

16. .. 391: Please correct "SM/MER".

17. l. 451: "Note that the KENDA-1/MER is in better agreement than KENDA-1/MEE with SMN/MER (not shown), which can indicate significant differences in the foehn influence at the two stations." (i) "not shown" Isn't this information shown in Fig. 11? (ii) Can the difference also be related to differences in locations (as argued for MWR/MEE above)?

18. l. 480 ff: I appreciate the summary paragraph. Out of curiosity, do the authors have any hypotheses about the reasons for KENDA wind speed overestimation and simultaneous temperature underestimation?

19. l. 490: "such wind speeds difference is subject to a discussion about a potential large overestimation of the winds at this location": Do you here refer to an overestimation specifically during foehn events or during all conditions?

20. l. 592 ff: Please replace "daily cycle" by "diurnal cycle".

21. l. 610: "the NWP": Please rephrase.

22. Figure 1: I appreciate the revised map. I would suggest to increase the size of Fig. 1b, and would find it more intuitive if the x-axis were reversed to match panels a and c. In addition, I would find it helpful if the caption would indicate colors for up valley and down valley wind, respevtively (e.g. up valley wind (red), etc.). Similarly for slope winds.

23. Figure captions: I would appreciate if the authors could revise figure captions (e.g. Fig. 3, 4, 7) and make sure to include the altitude of data which is shown.

24. Figs. 8 and 9: Are the same sub-sets of dates/data points used in DWL and KENDA profiles (i.e., are KENDA data points removed from the analysis when no DWL observations are available)? It looks like KENDA includes more data points. In contrast, in Fig. 6 the NAN grid points appear to match.

25. Fig. 12: Please correct date and time in all panels.

26. Fig. 12b: Please correct the colorbar labeling.

27. Fig. S11b,c and Fig. S12: I would ask the authors to increase the label sizes.

28. Supplement: I think Figs. S7 and S10 are not referenced in the manuscript. If they are relevant, please include a reference in the manuscript.

29. Title: Generally, abbreviations such as "NWP" are avoided in the title. Moreover, did the authors consider adding "Switzerland" in the title, as Meiringen is rather unknown?

---

## Referee Report (RR3)

**Review of 'Comparison of temperature and wind between ground-based remote sensing observations and numerical weather prediction model profiles in alpine complex topography: the Meiringen campaign' by Bugnard et al submitted to Atmospheric Measurement Techniques**

September 29, 2024

This review is for the second revision of the manuscript. I appreciate that the authors addressed my and the other reviewer's comments in this revised version. It is improved and clarified in some aspects such as the mechanism for the flow from the Sarneraatal over the Bruenigpass and the arbitrary threshold of 20 m/s to select thermally driven days. However, there all still areas that need improvement in my opinion, especially related to conciseness and and preciseness.

The manuscript contains a lot of description and speculation, making it cumbersome to read. The description of the figures in the manuscript is very detailed with numerous mentions of heights, times, and values. This amount of detail might be appropriate for a report, but is somewhat distracting in a scientific article. The manuscript could be more precise and concise, if it focused on relevant aspects (e,g, instead of describing every little detail of the figures, it would help to focus on the aspects directly relevant for the objectives of the manuscript) and using a clear and precise wording. For example, like in these two sentences 'Concerning KENDA-1 data, the foehn breakthrough is modeled too early on March 11 at both stations, on time on March 20 at both stations and on April 23 at MER and too late on April 23 at MEE. The foehn arrival and end is modeled sometimes on time by KENDA-1, but positive and negative time shifts of up to 4h at both stations' (l. 511-514). This could be combined and be shortened: 'KENDA-1 models the foehn breakthrough 4-h too early at both stations on March 11, on time at both stations on March 20, and 4-h too late at MEE on April 23.' Cases like this are present throughout the manuscript and I strongly suggest that the authors try to use a more precise and concise language.

As mentioned in my previous review, I think a shortening of the case studies and the discussion with a focus on the most relevant aspect would be beneficial. Instead of shortening the case studies, the authors added additional analysis on the responsible mechanisms for the flow descending over the Brueningpass. I think this analysis adds value and helps to understand the observations, but, without shortening other parts of the manuscript (like the discussion or description of the figures), results in a rather long manuscript (of 39 pages with 13 figures in the main manuscript and 13 in the supplemental).

More attention should still be paid to details (correct formatting). For example, the formatting is off in the first paragraph of Sect. 3.1.3. Also, times should have units (e.g. UTC) which are consistently use throughout the manuscript. Sometimes a.m. is used. For all heights which are with respect to mean sea level

'a.s.l.' should be added. For times above ground, a.g.l. should be used. This is currently very inconsistent.

In addition to these general comments that should be addressed, I am giving some specific comments and suggestions below.

**1 Specific comments**

- Title: The placement of 'profiles' in the title sounds a bit off. Perhaps it would be better to say 'Comparison of temperature and wind profiles from ground-based remote sensing observations and a numerical weather prediction model in complex topography: the Meiringen campaign'.

- l. 55: Slope winds are driven by a horizontal temperature gradient between the air adjacent to the slope and the free valley atmosphere'

- l. 62-64: What is meant by wind intensity? The reversal from upvalley to downvalley winds in the evening is driven by the reversal of the along valley temperature and pressure gradient. Please clarify.

- l. 85-87: Please be more specific. Why is precise knowledge essential for NWP? To evaluate and improve the models? And why are REM a solution? Are they assimilated or used for evaluation?

- Fig. 1: Please add that BRZ stand for Brienz. It would also be helpful to add the names for BRU, LUN, BUC, and GIH to the caption.

- l. 109: 'a.s.l.' is already used in l. 94.

- l. 130: 574 m?

- l. 141-142: Is the terrain shown in Fig. S3 filtered with this 2dx filter? It looks very steep with large differences between adjacent cells. To be meaningful, the terrain that is actually used by the model should be shown.

- l. 180: SMN was already introduced.

- l. 183: Where is FRU? Not included in Fig. 1. How is the cloud amount estimation done at FRU?

- l. 185-186: Are only wind observations used at BRU? What about temperature? What is 'similar temporal resolution'? Order of minutes or hours?

- l. 199: Please add information that line of sight of about 10 km is in downvalley direction.

- l. 228: What is 'end of winter'? Was there snow on the ground after mid-December?

- l. 230: I would appreciate if a brief description of how the foehn index works was included.

- l. 250: Radio soundings are not mentioned before. Where and when were they performed? What cold bias?

- l. 257: At what height are the statistics computed?

- l. 283ff: Why are still differences used? In the response, the authors indicate that they now use gradients, but this is not clear. In Fig. 5b, the temperature difference is still shown (unit deg C). What is T inversion amplitude? Is this inversion strength? Amplitude is a bit uncommon in this context. In the response to the reviewers' comments (comment 25), the authors claim that they now include the potential sources of error when comparing ground-bases and free atmospheric observations ('T inversions observed on the ground ...'). However, this explanation does not occur in the manuscript.

- Fig. 2. Please make plots in a) and b) the same size for consistency. How is the monthly diurnal cycle computed, is it mean or median? Please add. Is the difference in b) computed before or after the monthly means/medians for the observations and KENDA are computed? Do sunrise and sunset times account for orography? Are they monthly means/medians? How is the mean ridge height computed? Is this based on the lines in Fig. 1? Please explain.

- l. 364-465: This sentence is not clear.

- l. 367: Not clear, why vertical transport is important for weakening of drainage flows.

- Fig. 7: Do sunrise and sunset take orography into account? They are different from sunrise/sunset in Fig. S8. Better show the one taking orography into account.

- l. 373: 'underestimation of wind speed' is not clear. Upvalley wind is actually stronger (positive values) in KENDA. What is missing are downvalley winds.

- l. 376-377: This sentence implies the stronger presence of upvalley wind is leading to weaker downvalley wind speed, which is not physical. Please rephrase.

- l. 382: The phrase 'onset is anticipated compared to' is not clear.

- l. 384-385: What about moist convection? Surely there are convective clouds and precipitation during the summer months that can affect the flow.

- l. 392-393: How can synoptic winds lead to continuous downvalley wind underneath. Please rephrase.

- Fig. 9: Please add a) and b) to the figure. 'wind speed values'.

- l. 422-424: Weird sentence. Please rephrase.

- l. 443: Figures should appear in the same order as they are mentioned.

- l. 449ff: Annotation $(x, y, \theta)$ is not clear. Is GIH at the same height as BRU? If not, how is pressure at GIH used to compute potential temperature at BRU? How is the valley volume computed?

- l. 482: Foehn is NOT a katabatic wind. Is Haslital on northern side of Alpine ridge?

- Fig. 12a: Are sample sizes similar for each hour? With only 117 hours of foehn detected, this would mean that there are only 5-6 samples per hour. Pretty small sample size to compute distribution and to draw meaningful conclusions. This limitation should be mentioned.

- l. 509-510: This comparison is hindered by the fact that the value at SMN/MER is observed at the surface and the value at DWL/MEE at 800 m a.s.l.

- l. 529: Please formulate as hypothesis, 'can explain'.

- l. 531-532: Bise is mentioned here for the first time. How are Bise situations determined? The enhancement is is not documented.

- l. 546ff: Please explain why monthly medians are used and not temporally resolved values. Is this because of noise? The KENDA-1 analysis is used, i.e. the forecast skill is not investigated.

- l. 566ff: Several studies focused on valley wind in the Inn Valley. A classic one that would be good to include is Vergeiner and Dreiseitl (1987).

- l. 579: Where is Sion?

- l. 602ff: A study focusing specifically on the vortices in the Inn Valley is Babić et al. (2021).

- l. 626: What is 3?

- l. 637: The sentence 'Westerhuis et al. ....' is not clear.

- l. 655: Since the information content from the passive microwave radiometer decreases with height, the vertical resolution decreases and inversions and elevated layers are smoothed with height (e.g. Crewell and Löhnert, 2007).

- l. 670: Where is Visp? Without more detail this comparison does not make sense.

- l. 671: 'four-time shorter length of the Haslital' compared to which valley?

- l. 682-683: Differences in real-word and model valley depth certainly are also important.

- l. 697: 'from November 2021 through August 2022'.

- l. 713: Please rephrase '... simultaneous for the entire the profile.'

**References**

Babić, N., Adler, B., Gohm, A., Kalthoff, N., Haid, M., Lehner, M., Ladstätter, P., and Rotach, M. W.: Cross-valley vortices in the Inn valley, Austria: Structure, evolution and governing force imbalances, Quart. J. Roy. Meteor. Soc., 147, 3835–3861, doi:10.1002/qj.4159, 2021.

Crewell, S. and Löhnert, U.: Accuracy of boundary layer temperature profiles retrieved with multifrequency multiangle microwave radiometry, Geoscience and Remote Sensing, IEEE Transactions on, 45, 2195–2201, doi:10.1109/TGRS.2006.888434, 2007.

Vergeiner, I. and Dreiseitl, E.: Valley winds and slope winds - Observations and elementary thoughts, Meteor. Atmos. Phys., 36, 264–286, doi:10.1007/BF01045154, 1987.

---

## Editor Decision (ED1)

REVIEW OF "COMPARISON OF TEMPERATURE AND WIND PROFILES BETWEEN GROUND-BASED REMOTE SENSING OBSERVATIONS AND NUMERICAL WEATHER PREDICTION MODEL IN THE ALPINE COMPLEX TOPOGRAPHY: THE MEIRINGEN CAMPAIGN.

I read with interest the paper and I notice the great improvement in the text, which I now find easy to follow. In my opinion, you nicely responded and modified the work based on all the comments from the other reviewers. I just suggest some smaller modifications, to facilitate reading.

I suggest publishing after taking care of these final minor corrections:

1) line 89: you introduce the campaign in the Haslital here in the introduction. I suggest to provide here some small details (period, location), instead of postponing them in the methods and data section. It helps to give context, otherwise the reader has no idea about which campaign it is.

2) lines 180-190: please refer to figure 1 when you mention the additional station from which you take data from. In this way the reader can find where they are located.

3) line 193: how do you identify rainy conditions in the MWR? beware of wet radome effects after rain is over. Providing some details here does not hurt.

4) line 240 and around; I would start presenting the results from the results that are visible in the figure included in the paper, and then add the points coming from figures in supplementary material. The sentence "the maximum temporal gradient T.... " refers to figure S5 before figure 2 is introduced and we get a feeling of the situation you describe

5) For all plots: be consistent with the label of the time axis. Sometimes is "time of the day", sometimes "hours", some others have Time UTC. My suggestion, to have it consistent with the axis on the y, is to write "Time [hours UTC]".

6) some plots (fig 4, fig 12 in particular) compare lines in pink with lines in red and lines in blue. I is hard, sometimes impossible to distinguish the red and the pink. Please change colors.

7) lines 319 - 324: do you really need these introductory lines? the section on wind is nicely structured and you could save some text, I think.

8) In section 3.2.3 you talk about N winds. Is it North winds? I would write it extensively. I did not get immediately what you meant.

9) Please remove all sentences like "Figure X shows this is proportional to that", as it is in the sentence at line 419-420. You can rephrase all of them as "this is proportional to that (Fig x). In the example of the line 419-420 it can be written as: Clear warm days with low cloud coverage in July show a peculiar wind pattern along the Haslital (Figure 10a). Another case is in line 441-442.

10) Arriving to the end of the paper, it is clear that you mainly aim for climatological variability. I think it is then worth highlighting it more in the abstract. Also the title does not reflect this point particularly. You mainly show diurnal cycles and means, but in the abstract only the performance for monthly temperatures and wind medians made me think about climatology. Maybe it is worth mentioning some specific keywords there "climatology", "monthly/seasonal means" or similar.

---

## Author Response (AR2)

***"Comparison of temperature and wind between ground-based remote sensing observations and NWP model profiles in complex topography: the Meiringen campaign"***

*First, we thank the reviewer for the valuable, in-depth comments to our manuscript. Further analyses were done to explain that the wind from the Saneraatal can be explained by its bigger volume compared to the Haslital. This result improves the comprehension of the influence of the topography on the thermal wind system of the valley.*

*The answers to the comments and questions are written in italic thereafter. The explanations of this document cite the numbering of the figures in the first revision in accordance with the lines' numbers of the comments.*

**Answers to the reviewer 2 comments**

Second review of egusphere-2023-1961

'Comparison between ground-based remote sensing observations and NWP model profiles in complex topography: the Meiringen campaign" by Alexandre Bugnard et al.

Summary

The manuscript presents a comparison of campaign observations from the Meiringen Campaign in a narrow Swiss Alpine valley with the high-resolution 1-km KENDA analysis. The comparison focuses on temperature and wind profiles measured by a microwave radiometer and Doppler wind lidar, respectively, for ten months during 2021/2022. It is shown that observed and modeled seasonal climatologies of temperature and wind profiles agree well, although for specific situations, such as for example temperature inversions or foehn events the differences are relatively large. The manuscript also links the complex topography to thermal wind systems and presents cross- and along valley flow systems observed during the campaign period.

The manuscript presents valuable observations from an Alpine site and provides new insights in the quality of the high-resolution analysis in complex terrain and shows examples of how specific terrain-induced flow features can influence the differences between observations and the analysis.

The manuscript has been substantially improved during the first revision phase and the authors have addressed major reviewers' comments, i.e., the general structure was revised and the manuscript was streamlined. I support publication of this manuscript but I have several mostly minor comments that should be clarified and addressed prior to publication.

1 Comments

1. KENDA T bias

I still struggle to fully follow the discussion about KENDA temperature biases discussed in Section 3.1. l. 213 ff states: "The main observed pattern is a general low altitude (< 1500 m) T underestimation from KENDA-1/MEE." This cold bias pertains to all hours of the day and all months (Fig. 2b; except June). Subsequently, Fig. 4 shows that "KENDA-1 overestimates the T during nighttime (+1.5C) in both cells and underestimates it during the day (-2C in MEE and -1.5C in MER)." (l. 240 ff). I understand that different data and levels are compared in Fig. 4, however, at nighttime Fig. 4 suggests that KENDA is larger than MWR. The latter is not visible in Fig. 2b. I would ask the authors to elaborate on and clarify the KENDA warm or cold bias, respectively, and streamline this paragraph. If the main reason of the KENDA-SMN bias results from differences in altitude between KENDA grid box and SMN observation and the frequent presence of inversions, Fig. 4 and the respective text may be misleading. From Fig. S5 I cannot induce if KENDA overall over or underestimates temperature.

*The applied principle is to use the nearest data for each comparison. As already specified in the manuscript, the MER is at 574 m, so that the 2 m T data are observed at 576 m and the 10 m wind data at 584 m. MEE is at 589 m. The first MWR/MEE is at 625 m and the first DWL/MEE level at 775 m. The manuscript also specified the difference between KENDA-1 first level and the real topography (109 m at MER and 130 m at MEE) as well as the altitude of KENDA-1 first level (20m a.g.l.). The revised version specifies now in sect 2.2 that the first level is 705 m (574-109-20) for KENDA-1/MER and 739 m (589+130+20) for KENDA-1/MEE. As proposed, the observations' levels are now given in the figure or caption of Figs. 3, 4, 5, 7, 10 and 11.*

*Considering the applied principle, the first level of comparison of MWR/MEE and KENDA-1/MEE (Fig. 2) is 739 m, whereas Fig. 3 and 4 compare always the lowest level with SMN/MER. There is then an underestimation of KENDA-1/MEE compared to MWR/MEE between 739 and 1500 m. KENDA-1/MEE and KENDA-1/MER overestimate SMN/MER T (at 576 m) during night and underestimate it during day. Both Fig. 4 and Fig. 2 show that, during day, KENDA-1/MEE at 725 m modeled lower T than observed by MWR/MEE at 625 m. During night, KENDA-1/MEE often misses the T inversion leading to an overestimation of ground T and an underestimation of T below 1500 m. Fig. S5 presents explicit examples of this phenomenon leading to a negative difference KENDA-1/MEE-MWR/MEE at 705 m and a difference KENDA-1/MEE(at 705 m)-SMN/MER larger than MWR/MEE (at 625 m)-SMN/MER. In case of missed T inversion, KENDA-1/MEE extrapolated at 625 m (dashed red line in Fig. S5) would overestimate the T observed by MWR/MEE at the same altitude.*

*The text was adapted to better explain this apparent discrepancy between Fig. 2 and Fig. 4: "The missed T inversions by KENDA-1/MEE lead to both its important overestimation of the T at ground level (Fig. \ref{fig:boxplot_hr}) and its slight T underestimation between ~850-1200 m (Fig. \ref{fig:T_clim}). Detailed examples of T profiles during a day with missed T inversion by KENDA-1/MEE (Fig. S5) show these opposite T bias at several altitudes with SMN/MER and MWR/MEE observations."*

2. Altitude depiction in observation and KENDA data

I would appreciate if the authors could provide information directly in the text, figures, and/or figure captions about the altitude of the used data. It is difficult to remember the altitude of lowest model grid box at MEE/MER as well as of MRW and DWL. I think this would facilitate following the storyline of the manuscript.

*The altitude of the T and wind measurements were added in the captions of Figs. 3, 4, 5, 7, 10, and 11.*

3. I would ask the authors to again double-check the manuscript for typos, missing words, and grammar mistakes (e.g., l. 36 "Such inversions are favored in complex topography (Joly and Richard, 2018) and persist-s longer in deeper valleys, whereas inversion lifetimes converge to the one over a plain for wide valleys (Colette et al., 2003)."; l. 40 "The quality of predictions for", l. 392 "of a monthly median values"). Please also consistently adjust the date/time format.

*A colleague with high English skills reviewed the manuscript. In particular, the mentioned sentences were modified.*

4. l. 6: Please introduce the acronyms COSMO-1E and KENDA-1 as not everyone may be familiar with the terminology.

*Done*

5. l. 19: "of a model": I would specify this and explicitly mentions "KENDA-1".

*Done*

6. l. 125: "Vertical levels with spacings from 20 m at the surface": What is the height of the lower-most level?

*The lowest most level is 20 above the surface of the model's terrain. It has been inserted in the text.*

7. l. 176: "from 200 m to 12000 m above ground". Can the DWL measure successfully up to 12 km height?

*The DWL can effectively measure at 12km when cirrus clouds are present. This sentence is then correct even if such an altitude is not met in the absence of cirrus clouds.*

8. l. 186: "Even if SNM/MER surface observations are assimilated by KENDA-1, the comparison of the modeled and observed data allows evaluating the impact of the assimilation at MER." Please clarify this sentence. From a comparison of the resulting KENDA analysis and the assimilated observations alone, the observation impact cannot be deduced directly, unless first-guess (as mention in l. 272 ll) is available.

*The referee is right, the real impact of the assimilation cannot be estimated without a comparison with the first guess. The sentence was modified: "The comparison between KENDA-1 and observed data at MER allows evaluating the model's performances at a station, whose SNM surface observations are assimilated."*

9. l. 296: "direction at low altitudes (800-1000 m) is mainly from W-SW": I find this very difficult to see in Fig. 6, among others, as the 800-1000 m layer is very shallow and the colors are not distinct. l. 296: "in the rest of the profile": Please specify.

*First, Fig. 6 was changed and no further distinction between wind speed lower or higher than 20 km/h is made. The text was also adapted considering your comment by mentioning first only the*

*W direction (blue color) and second the "the rest of the profile up to ridge height".  The ridge height is also now added on the plots.*

11. l. 338: "The comparison of the first level of KENDA-1/MER (Fig. 7.c)": Fig. 7c suggests that KENDA-1/MER is shown at 775 m. Which altitude is shown?

*The altitudes given in Fig. 7 are right. To allow the comparison between the remote sensing and KENDA-1 at both sites, the same altitude of 775 m was chosen for all plots apart from SMN/MER.*

12. l. 373: "Plots of radial winds perpendicular to the valley direction clearly present this circulation pattern both in presence of up and down valley winds around sunset (Fig. S9)." Please rephrase, e.g. Figure S9 shows ...

*Done: "Fig. S9 shows radial winds perpendicular to the valley direction that clearly illustrate this circulation pattern observed in presence of both up and down valley winds around sunset."*

13. l. 360 ff: "Finally, KENDA-1/MEE overestimates the influence of the synoptic winds leading to the absence of along valley wind in winter replaced by constant slow down valley winds below 1200 m and to higher up valley wind speed in spring and summer." (i) "along valley wind in winter replaced by constant slow down valley"; Do you mean "up valley wind" replaced by down valley wind (as "along valley" wind includes both, up and down valley wind)? I'm not sure if I understand the authors reasoning why the "influence of the synoptic winds" leads to down valley winds in winter and an overestimation of up valley wind in summer in KENDA? Could the authors please explain their reasoning.

*The referee is right for both points. The observed low down valley wind replaces only the up-valley wind and this has nothing to do with the influence of synoptic winds, which is on the contrary visible at higher altitudes.  The figure was also modified to represent only data present in both time series. The text was consequently modified: "Finally, from November bis February, KENDA-1/MEE overestimates the influence of the synoptic winds leading stronger up-valley wind presence down to 800 m and models continuous down-valley winds below 800 m with shallower diurnal cycle than observed by DWL/MEE. The foehn influence in March up to 2500 m is well modeled."*

14. l. 370 "intense north-facing slope winds": Please rephrase; it is easy to confuse this with "north-facing" "slope winds" (i.e., south to north wind direction).

*Yes, it is confusing. The sentence was modified:" Intense winds from north-facing slope ($>$ 25 km/h) are also observed between 1400 and 2000 m during some hours around sunset with a much lower intensity in May."*

15. l. 389 ff: Please indicate where this information is shown.

*These sentences were added to shortly describe the content of the section to streamline the manuscript. The additional wind observations in the Haslital are described with Fig. 10 and the one in the Sarneraatal in Fig. S8. The differences of the wind system at MEE, MER and the entire valley volume refer to the analysis bounded to Fig. 10. We do agree that the mention " to the entire valley volume" is exaggerated and now we only mention from the lake of Brienz to MER.*

16. .. 391: Please correct "SM/MER".

*Done*

17. l. 451: "Note that the KENDA-1/MER is in better agreement than KENDA-1/MEE with SMN/MER (not shown), which can indicate significant differences in the foehn influence at the two stations." (i) "not shown" Isn't this information shown in Fig. 11? (ii) Can the difference also be related to differences in locations (as argued for MWR/MEE above)?

*i) Yes, Fig. 11 allows to see this affirmation so that "not shown" was replaced by Fig. 11.*

*ii) Yes, we could consider that about 1° difference between KENDA-1/MEE and KENDA-1/MER can be attributed to the difference in location as for observations. Fig. 11a however shows that the difference is much less systematic than between MWR/MEE and SMN/MER. This is then a supposition that we prefer not to discuss in the paper.*

18. l. 480 ff: I appreciate the summary paragraph. Out of curiosity, do the authors have any hypotheses about the reasons for KENDA wind speed overestimation and simultaneous temperature underestimation?

*As probably guessed by the reviewer, we do not have hypotheses about the simultaneous wind speed overestimation and T underestimation by KENDA. We tried to figure out thermodynamic solutions, but without success.*

19. l. 490: "such wind speeds difference is subject to a discussion about a potential large overestimation of the winds at this location": Do you here refer to an overestimation specifically during foehn events or during all conditions?

*We are only referring to foehn events, which is now specified in the sentence.*

20. l. 592 ff: Please replace "daily cycle" by "diurnal cycle".

*Done*

21. l. 610: "the NWP": Please rephrase.

*Done*

22. Figure 1: I appreciate the revised map. I would suggest to increase the size of Fig. 1b, and would find it more intuitive if the x-axis were reversed to match panels a and c. In addition, I would find it helpful if the caption would indicate colors for up valley and down valley wind, respevtively (e.g. up valley wind (red), etc.). Similarly for slope winds.

*The requested modifications were made in Figure 1. The caption was adapted to give the colors for along and cross valley winds: "The two cells of the model used are pink. Arrows representing up/down valley winds and north-facing/south-facing slope winds are colored respectively in red/blue."*

23. Figure captions: I would appreciate if the authors could revise figure captions (e.g. Fig. 3, 4, 7) and make sure to include the altitude of data which is shown.

*The altitude of the data was added either in the figure or in the figure caption.*

24. Figs. 8 and 9: Are the same sub-sets of dates/data points used in DWL and KENDA profiles (i.e., are KENDA data points removed from the analysis when no DWL observations are available)? It looks like KENDA includes more data points. In contrast, in Fig. 6 the NAN grid points appear to match.

*You're right and we modified Fig. 8 and 9 so that only data present in both time series are now plotted.*

25. Fig. 12: Please correct date and time in all panels.

*Done. Sorry for the typo.*

26. Fig. 12b: Please correct the colorbar labeling.

*Done. Sorry for the typo.*

27. Fig. S11b,c and Fig. S12: I would ask the authors to increase the label sizes.

*Done*

28. Supplement: I think Figs. S7 and S10 are not referenced in the manuscript. If they are relevant, please include a reference in the manuscript.

*Figs. S7 and S10 were suppressed.*

29. Title: Generally, abbreviations such as "NWP" are avoided in the title. Moreover, did the authors consider adding "Switzerland" in the title, as Meiringen is rather unknown?

*NWP was spelled and we added the world alpine to situate the geographical area.*

**Answers to the reviewer 3 comments**

Review of 'Comparison of temperature and wind between ground-based remote sensing observations and NWP model profiles in complex topography: the Meiringen campaign' by Bugnard et al.

The study investigates wind and temperature in the boundary layer of the Haslital in Switzerland using ground-based remote sensing and in situ instruments deployed during the Meiringen campaign from October 2021 to August 2022. It compares the observations of a microwave radiometer, Doppler lidar, and surface meteorological station to the COSMO-1E model analysis (KENDA-1). The valley is rather narrow with 1.5 km width and one of the sites is affected by the low altitude Brünigpass to the north. By means of monthly composites, nighttime inversion and thermally driven wind systems were detected. Large model errors were found for nighttime temperatures on the average as well as on individual days.

This review is of the revised submission of the manuscript and is provided in view of the interactive public discussion. In the revised version, many of the comments of previous reviewers are addressed. For example, the manuscript was shortened and the appendix was moved to a supplemental. The gathered data provide a great opportunity to investigate the boundary layer conditions in a mid-sized Alpine valley and to evaluate the performance of KENDA-1. In my opinion, the issues with partial data assimilation at MER is of particular interest and may be relevant for other locations as well, leading to potential forecast improvements in complex terrain. While I believe that the manuscript is improved compared to the original version, I still have some major concerns about some of the aspects of the study, some new, some already raised by the previous reviewers. For example, the authors strongly focus on thermally driven flows, however, they don't distinguish between thermally and synoptically driven conditions when computing the monthly composites, which, in my opinion, masks many of the relevant features. KENDA-1 has a 1.1 km horizontal resolution, while the Haslital is 1.5 km wide. The authors discuss the difference in valley floor height between reality and the model, but they do not discuss the terrain in the model at all, such as shape and depth of the valley, and to what extent the Br¨unigpass is resolved or how many grid points are available in the valley. I provide more details on these two aspects as well as many others in my comments below. Overall, I think the manuscripts includes too much description and not enough analysis in its present form. I suggest that the authors consider my comments before the manuscript can be accepted for publication.

1 General comments

1. The manuscripts contains a very detailed description of conditions in the results section, but hardly any investigation of the relevant processes. It first presents a lengthy description of the temperature and wind field using monthly composites with lots of details on values in specific layers etc, followed by a description of a three-day case study when the flow from the Br¨unigpass affected the conditions in the Haslital and by a description of three foehn events. In my opinion, it currently is a mix between a campaign overview and some preliminary results. It is very descriptive without a clear story line. If the case studies are meant as teasers (for future manuscripts) they could be much shorter. No in-depth analysis of the case studies is presented and there are many open question, which arise to the reader. For example, why should a warm upvalley wind from the Sarneraatal descend into the Haslital? An in-depth analysis would probably be beyond the scope of the manuscript, but such contradictions to theory and open question should at least be mentioned. Also, I don't think that there is enough observational proof presented to conclude that the wind conditions in the Haslital are affected by the flow from the Br¨unigpass or that there is a vortex present. These findings should be formulated as hypotheses. A more detailed study, possibly using the 3D model output is needed to provide strong evidence for this and to investigate the physical processes.

> As mentioned by the reviewer, an in-depth analysis of all the case studies presented in this manuscript is beyond the scope of this publication. Here we aim at providing a board overview of the complexity of wind regimes of a narrow Alpine valley and providing insight into how well this complexity is captured by an operationally used NWP model. Thus, we focus on highlighting challenges of the meteorological model and identifying potential (highly localized) phenomena which in future should be further assessed (e.g., during TEAMX). Thus, we do not claim to provide a complete explanation of all phenomena observed, but rather raise awareness for such local scale phenomena that

*might cause difficulties to operational weather prediction models and thus serve as a baseline for future measurement setups.*

*We agree that for both examples highlighted by the reviewer (upvalley flow from the Sarnertal and the influence of the wind regime in the Meiringen valley), the observational proof that can be underlined with measured wind lidar data is limited. This is a consequence of the setup used in this study. For a 3-dimensional representation of vortices, multiple systems (scanning wind lidars) need to be available to perform multiple doppler analyses that can yield estimations of 3-dimensional wind field information in addition to the orthogonal wind components. However, such 3D data is not available for the campaign presented here. But personal observations on site and various discussions with local farmers provide additional evidence for the hypothesis that the up valley wind from the Sarneraatal influences the wind regime in the Haslital. This condition is met if the synoptic condition is related to a Bisenlage, which corresponds to large scale north to north easterly winds. During such conditions, the wind in the Sarneraatal is directed from NNE to SSW, as indicated by the SMN measurements for the 22.11.2021 in the figure below.*

*As explained in the manuscript and used in this study, three meteorological stations are available for wind observations in the Haslital. One is located close to the village of Meiringen (MER, blue) and thus situated in the upvalley, i.e. to the east of the measurement setup during the campaign. A second station is located close to the runway of the airport and the measurement setup during the campaign (MEE. black), a third station is located to the west close to the lake of Brienz (BRZ, red). The arrows indicate the dominant wind direction during the period from 9:30 to 12:30 UTC (during which all the station have wind speeds of >1.5m/s). In the Hasli valley the wind direction is clearly separated with easterly wind in the western part (MEE & BRZ) and westerly winds in the eastern part (MER). Similarly in the snap shots of the RHI scans of the wind lidar, the regime difference is obvious in the along valley direction (left plot). In the direction of BRZ the radial wind are constantly directed away (red colors) whereas to east (MER) a low level radial winds are generally directed towards the instrument, whereas at slightly higher altitudes the radial wind has the opposite sign (directed away from the instrument). Unfortunately, the range is limited to 4km and thus the station MER (located at 5km) distance is not covered. Nevertheless, from this we conclude that there is an effect of the wind coming from the Brünig during these specific synoptic situations (Bisenlage). In addition, the RHI directed towards the Brünig pass (right plot in the lower figure) shows a distinct pattern of radial winds at low elevations along the RHI transect (from N to S) and a 2D vortex signature above the measurement location (indicated by the arrows), that corresponds to the middle of the valley. Such a structure in the two orthogonal planes of radial velocity do indicate a vortex like structure. Nevertheless, without a 3-dimensional observational dataset (e.g. by multi-Doppler analyses), a detailed analysis of the vortex structure is not straight forward and thus we agree with the reviewer's comment that this is not a proof of, but rather a hint for a*

*vortex structure. However, this illustrates the complexity of wind regimes in narrow valleys in complex terrain and we believe it is worth to be mentioned this in the current manuscript to make other campaigns aware of such highly local phenomena, based on which the strategy of future campaigns could be better defined.*

[Figure]

*Fig. 1: a) map of the valley with wind direction during the event, b) wind direction and speed at MER, MEE and BRZ and c) radial wind compounds measured by DWL/MEE.*

*Furthermore, we investigate the pressure difference between Giswyl (GIH) in the Sarneraatal and MER in the Haslital. We used the pressure reduced at sea level to get rid of the altitude difference. Fig. 2 clearly shows that the mean monthly reduced pressure at GIH is higher than at MER for all months but the winter months. The highest difference (>1 hPa) is observed from mid-day and persists until the late afternoon or*

*even the early evening. The difference between potential T measured by MWR/MEE at the altitude of BRU (1010 m) and at BRU is also positive. Air masses from the Saneraatal are then colder than air masses in the Haslital not only in case of biselage. Such a phenomenon can be explained by the valley volume effect since the volume of the Sarneraatal is 1.7 time bigger than that of the Haslital. The heating of the air masses occurs then more slowly than in the Sarneraatal and induce the observed lower T. The up valley wind passing the Brünig Pass will thermodynamically tend to fall into the Haslital at MEE. This phenomenon is enhanced in the afternoon when the up-valley wind in the Sarneraatal is the strongest but also happens sometimes in the morning. This corresponds to the DWL/MEE measurements and seems to be well modeled by KENDA-1/MEE.*

*A figure will be added to the manuscript and this phenomenon is now explained in section 3.3.*

[Figure]

*Fig. 2: a) Monthly diurnal cycle of the difference in pressure reduced at sea level between GIH and MER, b) Monthly diurnal cycle of the difference in potential temperature at 1010 m above MWR/MEE and at BRU and c) Monthly diurnal cycle of the difference in T at 1010 m above MWR/MEE and at BRU.*

2. The analysis focuses very much on thermally driven flows, but the monthly composites are not separated for days that are dominated by large-scale conditions (frontal passages, foehn) or are affected by clouds (unfavorable for thermally driven flows) or are dominated by fair weather conditions (favorable for thermally driven flows). Computing composites over all days masks the signal of thermally driven flows (which primarily occur under fair-weather conditions, as correctly mentioned in l. 48). The authors still interpret the weak signals in wind in the composites and conclude that e.g. no thermally driven winds were observed in December and January. However, they might still be there just not in the monthly composites. Furthermore, including everything in the averages makes the comparison to other studies (Discussion section), in which days where filtered for thermally driven flows, not very meaningful. For example, conclusions on thermally driven flows are drawn from the composites of along valley wind component in Figs. 7 and 8. There are several features that are not typical at all for thermally driven flows and differ from theory, like the persistence of upvalley flow near ridge height and the decrease in downvalley wind strength during the night. This could be a result from the projection of the

flow on along valley wind direction or a result of sampling over all types of conditions. Either way this discrepancies need to be discussed and attempted to be explained.

> *First, the use of the word "projection" to describe how the along and across valley wind speeds were computed is misleading and we apologized for this misinterpretation.  In fact, along valley wind are selected in a +-15° around the valley axis and cross valley wind in a +- 30° around the perpendicular to the valley axis. This is now described in the experimental section.*

[Figure]

[Figure]

> *The described results are then not a result of the projection since, e.g. along valley does not contain any influence from the N-NE wind from the Brünig Pass.*

> *Second, as answered thereafter (p. 11 of this document), a selection of very good (not shown) or good days (Fig. 3 this document) does not modify the main feature of the thermally induced valley winds. We think then the monthly composite still allows us to describe the main wind feature (first goal of the study) and to compare the modeled with the observed data (second goal). Finally, a comparison of the results of this study with other studies seems to us still worth, even if the selected weather conditions, the instrumentation and period of the year are not always identical.*

Monthly plots for wind direction are separated using a wind speed threshold which seems a bit arbitrary and needs more justification. It is currently not clear at what height this threshold must be met and during what time period. Also, why do the authors not distinguish days for the other composites (along-valley and across-valley wind speed, temperature) as well? Since the focus is on thermally driven flows, the analysis could also be restricted to composites of days with thermally driven flows.

This would reduce the number of panels and streamline the manuscript. A better and more physical way to distinguish days with thermally driven flows from days dominated by synoptic flows could be by looking for a wind direction reversal twice per day and/or by considering cloudiness.

> *The arbitrary threshold of 20 km/h was indeed also questioned by the two first reviewers. We then decided to modify Fig. 6 and to plot and discuss first the monthly wind direction without wind speed threshold. As proposed by the third reviewer, plots of monthly wind direction for good weather conditions are now also available in the supplement. They were not inserted in the manuscript since the method to select days with thermally driven flows relays only on cloud coverage at the nearest station with this parameter (FRU). We also think that the criteria allowing thermal winds should however be studied before to be used. Such a study is beyond the scope of this paper, but the following questions could be adressed: which cloud cover during which period impedes thermally driven flows to occur? What is the influence of the cloud cover*

*during the night? Should we also include the influence of ground-based T inversions? Are these factors influenced by the seasons (i.e. the mean T or the snow cover)? Similar questions (e.g. time and altitude of wind direction reversal) are also raised concerning an automatic detection of thermally driven winds by wind direction reversal, particularly in the described complex terrain presented in this study. It has also to be considered that a selection of only "very clear days" misses information about thermally driven wind in "clear days". The provided new figures provided in the manuscript and in the supplement allow to answer some questions raised by the reviewer, but a complete analysis of the occurrence of thermal valley winds as a function of different synoptic situations, different cloud amount is beyond the scope of this study.*

*Sections 3.2 and 3.2.1 were consequently modified to describe the new Fig. 6.*

[Figure]

*Fig. 3 (new Fig. 6): Monthly median wind direction [°] for a) DWL/MEE, b) KENDA-1/MEE and c) KENDA-1/MER (1.11.2001-23.08.2022).*

[Figure]

*Fig. 4 (inserted in the supplement): Monthly median wind direction [°] for a) DWL/MEE, b) KENDA-1/MEE and c) KENDA-1/MER (1.11.2001-23.08.2022) for clear weather days. The clear weather days are determined by less than 5 octas of cloud cover measured at the SMN station of Frutigen.*

3. Composite plots for wind (Figs. 6, 8, 9) are presented with 10 panels per row, which makes it impossible to see any details on the time axis and to follow the detailed description for specific hours. I highly suggest to think about another way of presenting these composites plots. I understand the desire to reduce the number of figures, but this should not be done at the cost of visibility.

> *Figure 6 was modified (see previous answer) and comprises now only three rows allowing more space for each pannel. The space between the plots was also reduced and we hope that this improves the global readability of the pannels.*

4. A discussion of the model terrain is needed, which goes beyond the difference in valley floor height. For the capability of the model to simulate terrain induced features, the shape and depth of the valley and the numbers of grid points is most relevant. This is not discussed at all. The Haslital is 1.5 km wide and KENDA-1 has a 1.1 km resolution. What about the Sarnaraatal? Is the Br¨unigpass even resolved in the model? How are the lakes resolved? How much terrain smoothing is done? In l. 74, it is stated that the grid resolution should be about 10 to 20 times higher than the relevant topographic scale to fully capture the different exchange processes. This aspect is very important when interpreting KENDA-1 results, but is currently not considered at all.

*The most important information on the model terrain such as the ridge heights, the height of the Brünning Pass and the position of the cell containing the MER and MEE were already given in the manuscript. Some points are further discussed as answer to the comment 12 and corresponding sentences have been added to the methods and discussion sections.*

5. Some sentences are not very clear and perhaps the authors should consider using a professional editing service to remove these language issues. Examples include, l. 202-203: Layer with higher T develops gradually from sunset to sunrise to reach monthly-related maximal T and height. l. 250-251: Globally, the measured MWR/MEE first level T are closer to the SMN/MER T than the modeled T. l. 256-257: The analysis of the negative ground T difference between MER at 590 m and BRU at 998 m (horizontal distance = 3.7 km) shows that near ground T inversions are common during the night for all months in the study. l. 269-271: The missed T inversions by KENDA-1/MEE leads to both its important overestimation of the T at ground level (Fig. 4) and its slight T underestimation between 850-1200 m (Fig. S5 for detailed examples). l. 296-297: ..., whereas flows from W-NW are measured in the rest of the profile concerned by up valley winds (see further explanation in sect. 3.3). l. 303-305: The good KENDA-1/MEE performances comprise first the influence of the foehn up to 2500 m (ws >20 km/h) as well as the presence of valley wind below 1200 m (ws <20 km/h) in March. l. 345-347: Thermally induced wind height increases with temperature, reaching 1000 m in February, 1800 m in May and up to 2000-2200 m in July and August. l: 371-373: This suggests a circular motion with North updraft winds (median vertical velocity of 1 km/h) that cross the valley at a low altitude, rise against the north facing slope and come back at higher altitude with a South downdraft component l: 599-600: This is especially the case at the end of March, when enhanced night time radiative cooling and important global solar radiation form strong inversions.

*A competent English writing person corrected the manuscript. All the mentioned sentences were modified.*

2  Specific comments

1. l. 64: The classic work of Whiteman and Doran (1993) could be referenced here.

*The Whiteman and Doran (1993) reference was introduced with a reference to the forced and the pressure-driven channeling mechanism.*

2. l. 73-74: The classic work of Skamarock (2004) Skamarock and Klemp (2008) could be referenced here.

*The citations have been inserted in the text.*

3. l. 84-87: Very complicated sentence. Please rephrase.

*Done*

4. l. 88: Is the "first objective of the campaign to study the seasonal and diurnal cycles" or is this the first objective of this study?

*Right, this is the first objective of the study and not of the whole campaign comprising instruments ( Radar and ceilometer) that are not used in this analysis. The sentence was consequently modified.*

5. l. 93: I think a short introduction on what KENDA-1 is and why its evaluation is important needs to be added to the introduction (possibly where NWP models are mentioned).

*An introductory sentence has been added.*

6. l. 94: I find the acronyms MER for Meiringen and MEE for Unterbach a bit unfortunate, since they are very similar and easy to mix. Maybe just a matter of taste.

*The acronyms were not chosen by the authors in the context of this study but are defined by the various networks in Switzerland. Even if more distinct acronyms would be helpful for the readers, the designation of the stations by their usual acronyms is a priority.*

7. l. 104: Why are the times not given in local time? This is advantageous for a study that focuses on thermally driven diurnal wind systems. At a minimum the time difference between local time and UTC needs to be given.

*UTC time was used since it corresponds to time used in our databank. The difference with local CET time is one hour (CET=UTC+1) and the difference between UTC and solar time at Meiringen (longitude= 8.1909°E) is about 40 minutes. These small differences have no impact on the comprehension of the diurnal cycles and sunrise and sunset times are most of the time represented in the figures. The difference between CET and UTC is now given in the manuscript: "Local time corresponds to Central European Time (CET), which is one hour ahead of UTC time (UTC+1)."*

8. l. 105-106: The temporal resolution of the observations is higher than 1 hour. How is that considered? Are these values averaged to 1h values before computing monthly composites or are instantaneous hourly values used? When computing the composites how is data availability considered? Are model data only plotted for times and heights where observational data are available?

*The measured data are first aggregated into hourly values, which are then used to resolve the diurnal cycles of the figures. Only time with observations are reposted in Fig. 6, whereas all data from KENDA-1 are reported in Figs 8 and 9. This is now corrected and the text was consequently adapted.*

9. l. 110-112: The Gadmertal and Rychenbachtal are not shown in the map and not relevant for the study. Please remove or modify the map in Fig. 1 to include them. This was already a comment of a previous reviewer.

*The mentions of the Gadmertal and Rychenbachtal have been deleted in the revised version of the manuscript.*

10. Fig. 1: Fig. 1b is impossible to see. Please increase line width, axis labels, and legend to make it readable. I suggest flipping the x-axis, so that the location of the sites is consistent with 1a and 1c. Why are the dots in Fig. 1b not at the valley floor? Are the station heights in Fig. 1b from the model? In Fig. 1c, labels along VW and across VW are very hard to read.

*Fig. 1 was modified as requested by both reviewers.*

11. Section 2.2: This section sounds like COSMO-1E is used:" The NWP model used in the study is the limited-area non-hydrostatic atmospheric model from the Consortium for Small-Scale Modeling Model

(COSMO)". I think it would be better to make clear from the beginning that the analysis KENDA-1 is used, which uses 1-h forecasts from COSMO-1D and observations. It would also be helpful to mention here how close to the investigation area the different observations are assimilated. SMN is assimilated at MER (this should already be mentioned here). But what about the sondes and profilers? How far away are they from the investigation area?

> *The manuscript has been adapted accordingly.*

> *The distance between Meiringen and the assimilated radio-sonding at Payerne is of 94 km, whereas the distances to the three assimilated radar wind profilers are comprised between 75 and 110 km. It must be noted that all these profiling observations are situated on the Swiss plateau whereas Meiringen is in the Alps.*

12. l. 141ff: Where are the MER and MEE model grid cells located in the model terrain? Are they at the valley floor in the model terrain? This is more meaningful than comparing the grid cells to the real terrain, which the model does not know.

> *As can be seen in the next figure, both MEE and MER are located at the valley floor in KENDA-1 DEM, so that the thermal valley wind system should be modeled correctly at the grid cells comprising MEE and MER. The narrow valley floor of about 1.5 km corresponds to one grid cell in KENDA-1 DEM at MEE and MER and enlarges to two grid cells 2 km after MEE in the vicinity of the Lac of Brienz.*

> *The following sentence was added to the manuscript: "In the modeled terrain, both MEE and MER stations are situated in the cell grid corresponding to the valley floor."*

[Figure]

*Figure 5: Comparison between KENDA-1 DEM (left) and the DEM25based on the 1:25'000 Swiss national map for the Haslital valley towards Lake Brienz to the west (bottom) and up the valley (top) to the east. The MEE and MER are represented by the red and green dots, respectively.*

13. l. 150ff: I think the fact that MER data are assimilated in KENDA-1 is critical for the results and needs more attention. If the observations are assimilated during certains conditions (daytime) and not during others (nighttime inversions) this may affect the error distribution. In the response to reviewer 2, the authors included an example demonstrating how often MER data are rejected and argue that 'One of the reasons to compare observed and modeled data at MER is that the MER ground observations are only assimilated if the difference with the modeled data is inside a given threshold." This information as well as the threshold would be helpful to include in the manuscript. Furthermore, I think including statistics on how often MER data are assimilated during the whole campaign and at what times would be most helpful and interesting.

> *We added a description of the first guess check and a statistical evaluation of how often and on what daytime the rejections occurred.*

14. Section 2.3.2.0: Was the RPG neural network retrieval used? As far as I know, HATPRO-G5 comes with a surface met sensor. Are this information used in the retrieval?

> *Yes RPG retrieval was used and the surface measurements are used in the retrieval.*

If not, this surface met measurements at MEE could be an additional observational source to evaluate KENDA-1. Why was this not used?

*Indeed, the Radiometer surface measurements could be used to evaluate Kenda 1 but they were not considered during this study.*

What does 'line of sight of about 10 km' mean? Is this the line of sight in the direction of the low elevation scans? In which direction was this performed (up or down the valley)?

*The line of sight of about 10 km means that there was no obstacle in the 10 km distance from the instrument in the direction of the low elevation scans. It was performed down the valley.*

Temperature biases in microwave radiometer retrieval are highly instrument dependent, arising from spectral biases or liquid nitrogen calibration. The cold bias found by Hervo et al. (2021) is not necessarily transferable.

*Indeed, the conclusion of the report might not be transferable. All mention to this report and the associated conclusions were then removed.*

'The instrument at MER ..' isn't the MWR installed at MEE?

*MER was replaced by MEE*

15. Section 2.3.3: The scan strategy of the Doppler lidar is not clear. It sounds like the lidar did a combination of vertical stare (every 10 min for two minutes) and DBS (every 5 min). What did it do in between? The scan in Fig. S9 looks like it is an RHI scan, this needs to be mentioned here.

*No time is left since the configuration of the Lidar for a 10-minute period was the following:*

- *Starting with DBS for 2 minutes*
- *Fixed vertical scan for 2 minutes*
- *DBS for 3 minutes*
- *RHI parallel and perpendicular to the valley for 2 minutes*
- *DBS 1 minute*

*The manuscript was consequently modified to be more explicit: " There are three measurement modes: 120 second zenith scans performed each 10 min to measure vertical wind speed, Range Height Indicator (RHI) scans for two minutes every 10 minutes to measure radial wind speed along and perpendicular to the valley (not used in this study). The rest of the time the instrument was measuring in Doppler Beam Switching (DBS) scans providing 7 independent wind profiles every 5 min to measure horizontal wind speed. In this analysis the wind profiles were averaged for each 5 minutes interval."*

According to the Vaisala data sheet the maximum range for Wincube 100S is 3 km. Was this system modified to reach 12 km maximum?

*The " Typical maximum operational range" provided by vaisala is limited by the presence of target. It is usually 3km but in case of elevated aerosol layers or clouds the 100S can measure higher. The data sheet also mentions the "Max acquisition range" that is around 12km for 100S.*

16. l. 195: Heavy liquid precipitation ?

*Yes, it is now specified in the text that the precipitation was liquid one.*

17. l. 189ff: None of the information on the general conditions during the campaign are considered when presenting the results on temperature and wind. How do the boundary layer conditions differ over snow covered vs. snow free ground? Does KENDA-1 performance depend on snow cover? Because of the monthly composite and snow cover lasting until mid-December this is all mixed together. How do the heat waves reflect in boundary layer conditions and KENDA-1 performance? If the information on general conditions is included, it should be considered when desribing the results.

*The general conditions are now referenced each time they allow to explain the results in the results and the discussion. As explained thereafter, a complete analysis and description of the ABLH is beyond the sclope of this paper.*

18. l. 202-209: The description of temperature changes should be related to the boundary layer evolution. 'Layer with higher T' is the daytime boundary layer. Perhaps the authors could compute convective boundary layer height using the parcel method. I do not see the value of presenting temporal T gradients (Fig. S4), especially not for monthly composites. It is not surprising that T increases during the day and decreases in the evening. I suggest removing this to streamline the manuscript. In Fig. S4 temporal gradients are shown, in the text vertical T gradients are discussed (C/km).

*The height of the convective boundary layer (CBLH) clearly depends on the T profiles described in Fig. 2 and § 3.1.1. CBLH computed by both the Parcel and the Bulk Richardson methods were computed from the MWR/MEE T and DWL/MEE profiles. During this study, we had not time to compute the Mixing Layer height (MLH) from the aerosol backscattering profile measured by the ceilometer. A complete study of the Mountain Boundary Layer (MOB) involving all potential ABLH computed from the MWR, the DWL (using backscattering and wind information) and the ceilometer during day and night associated with estimation of the stability of the atmosphere and the cloud cover would be very valuable. This study will perhaps be done in the future but is clearly beyond the scoop of this already very long manuscript.*

*The evolution of T in MEE is indeed not surprising but is necessary before to evaluate KENDA-1 performance. As requested, the § was shortened: "The evolution of T in MEE from February to July (Fig. 2.a) presents as expected clear diurnal cycle with a vertical extent depending on the season. Layer with higher T develops gradually from sunset to sunrise, persists during the first half of the night and gradually fades out towards sunrise. The time of the T maximum, the persistence of the warm layer and the extent of the warm layer are all enhanced during summer months. The maximal temporal T gradient usually follows sunrise and sunset (Fig. S4) and are confined below 1500 m with values up to +5°C/h in the morning and between -4 and -6.5°C/h in the evening."*

*Concerning the T gradients of Fig. S4, the units of the paper were falsely attributed to °C/km instead of °C/h. The manuscript is now corrected.*

19. l. 213ff: Fig. 2b show the bias. Did the authors also investigate mean absolute error or root mean square error? Is mean absolute error also small when biases are small or are the small biases related to averaging artefacts (days with very large positive biases vs days with very large negative biases)? How

are observed and modeled profiles compared? Are modeled heights interpolated to observed heights or vice versa? Since the temporal resolution of the observations is higher than KENDA-1 are they averaged before comparing to the hourly model output? This information should probably be included in Sect. 2. I assume the authors refer to the study of Hervo et al. (2021) when saying 'The cold bias between the MWR and the radio sounding could however suggests a larger error of KENDA-1.' As mentioned before, biases are instrument specific and I do not think this is a valid conclusion.

*No complete analysis of the uncertainties was done.*

*To allow the comparison between the modeled and observed data, all profiles were linearly interpolated at a vertical resolution of 10 m. This information was missing and is now given in sect. 2.: "Finally, all profiles were linearly interpolated at a vertical resolution of 10 m to allow comparison between the observed and modeled data."*

*As described at line 139, Kenda-1 data corresponds to instant hourly values, while the observed data are hourly medians as described at lines 105-10. The use of hourly medians for the observation was chosen to decrease the uncertainty of the measurements at the cost of introducing a further difference between the modeled and the observed data.*

*To avoid any confusion, we deleted the sentence "The cold bias between the MWR and the radio sounding could however suggests a larger error of KENDA-1."*

20. l. 223: Please explain why observations at MEE are compared to MER (here and later in the manuscript). These are different sites affected by different physical processes.

*The set-up of the campaign did not allow us to install the REM instruments at the SMN/MER station. The best compromise was to install them at MEE that is only 4 km apart in the same valley, and on the valley floor. A set-up with all instruments at the same place would have simplified the comparisons to only in-situ with REM observations and observations with modeled data. MEE is a station in the near vicinity of MER but the topography of the valley presents also different features at both stations, e.g. the presence of the Brünig Pass at MEE. This forces us to compare 1) in-situ observation at MER with both the REM observations at MEE and the modeled data at both MEE and MER, 2) REM observations at MEE with modeled data at MEE, but also 3) modeled data at MEE and MER in order to estimate if the differences between in-situ and REM observations are due to topographical features or to the instrumentation.*

21. Fig. 2: Please add ridge height to panel a). To better see stablity it would be helpful to add potential temperature isentropes. For example, observed potential temperature isentropes could be added to a) and simulated potential temperature isentropes could be added to b).

*The ridge height was added to Fig. 2a. We think that the addition potential T isentropes to T profiles would complexify the plots and require a further description in the paper.*

22. Fig. 3: Please enhance label and legend size.

*Done.*

23. l. 234-239: What is the justification to use a mean environmental lapse rate of -6.5°C/km during nighttime temperature inversions? The height difference between the model grid point and real world is only one challenge when comparing observations to model output. The terrain in the model needs to be considered as well, i.e. is the grid point at the model valley floor or on the slope? What is the valley depth, etc. (also see my general comment on this).

> *A mean environmental lapse rate was only used to test if the mean discrepancies in ground T estimation could be explained by the altitude differences between the stations of MEE and MER. This seemed to us an important first result, which supports all other comparisons. Obviously, the use of a mean environmental lapse makes no sense for specific cases and particularly in presence of T inversions. A sentence explaining this in the first version of the manuscript was suppressed as required by the reviewers.As described in our answer to comment 12, the grid point of both sites are at the valley floor and not on the slope. This information is now mentionned in the manuscript. Moreover, the mean ridge's height being 2200 m, the mean valley depth is 1600 m. This information is now added to §2.1. The topography of KENDA-1 is already described in the manuscript at § 2.2: "the altitude difference between the valley floor and the crests is thus reduced of several hundred meters and, in particular, the Brünig Pass is only 200 m higher than the valley floor."*

24. Fig. 4: Are the number of samples the same for all time series, that is, are only time stamps used when all observations and model output was available?

> *Yes, the number of samples are the same for all observed and modeled time series. This is now specified in the figure caption.*

25. l. 252ff: Using ground stations as pseudo-profiles can be affected by local impacts (e.g. solar heating during daytime, slope winds), even more since BRU is located at a pass. A brief discussion on potential error sources should be included (Whiteman and Hoch, 2014). How are temperature inversion determined? Just by computing the temperature difference between the upper and lower height and detecting negative difference? Was a minimum absolute value required? The authors compare temperature difference over layers which are not exactly the same depth and which introduce additional uncertainties. Why not compute gradients per fixed height interval? It is interesting that even though March was dominated by foehn events there is a very clear diurnal cycle in inversion frequency and amplitude. This should be discussed. In Fig. S5, there is absolutely no inversion visible in the KENDA-1 profiles. This is very strange and deserves further discussion, in my opinion. Furthermore, it has to be considered that the MWR profiles also have uncertainties and the smoothed shape of the profile may lead to an overestimation of inversion amplitude.

> *The potential sources of error when comparing ground-bases and free atmospheric observations are now mentioned in the paper: "T inversions observed on the ground may present an offset compared to observation by remote sensing in the free atmosphere due to the formation of nightly cold and daily warm surface layers or to different insolation or soil moisture depending on location. \citep{Whiteman_2014} observed differences generally within 1°C (standard deviation= 2-3°C) and a better agreement over steep slopes and during winter. BRU is influenced at least during daytime by colder up valley wind from the Sarneraatal (\ref{heteogenity_wind_valley}), which, however, also affect MWR/MEE and SMN/MER. (...)*

*Colder offset in BRU during night should lead to a higher frequency of T inversions observed from ground stations data, which is not the case ."*

*The temperature inversion is determined by detecting negative between the upper and lower T and no minimum absolute values were required.*

*As answered to comment §19, all the profiles were linearly interpolated with a resolution of 10m. The layers from observed and modeled profiles have then always the same depth. In that sense, the gradients are computed per fixed height intervals as suggested by the reviewer. The linear interpolation is well adapted to T profiles even if some uncertainties are still introduced.*

*In March, the 10 days with foehn corresponds to 96 hours, mostly during daytime (62.5%) so that foehn does not have a large impact on the T inversion frequency during night. The number of T inversions during daytime would have been perhaps larger without foehn events. The following sentence was added to the manuscript:" The foehn influence in March occurred mostly during daytime (8.1 \% of daytime and 4.8 \% of nighttime) and had then no direct influence on the T inversion frequency."*

*There is no inversion in the Kenda-1 profiles presented in Fig. S5 since it is indeed an illustration of a missed T inversion by KENDA-1. I do not see how it deserves the discussion on the weakness of model to predict ground-based T inversions.*

*Finally, the uncertainty of the MWR profile due to the smoothing is quite difficult to quantify and its impact on the detection of the T inversion is difficult to estimate.*

26. Fig. 5: I find the amplitude plots confusing. I assume that sample size is not constant during the day and some of the spikes during summer daytime in the MWR data are probably caused by averaging over very few days. Perhaps a minimum sample size should be required for showing the amplitudes. The grid lines do not fit to the tick labels on the x-axis. What is meant by '10m spaced vectors'?

*Yes, the sample size varies during the day and the number of samples per day can be estimated from Fig. 5a (100%= 28-31 depending on the month). The spikes clearly correspond to only 1-2 T inversions. For coherence between Fig. 5a and Fig. 5b, we will however keep the amplitude for very small sample size.*

*The mention of the 10 m grid lines spacing was removed since the interpolation of all profiles to 10 m vertical resolution is now described in section 2.*

27. l. 272ff: As mentioned before, I find this issue with assimilation very interesting. Do the large differences between the observations and the 1h forecasts mean that there is no inversion in the model? What about later forecast hours? Is this maybe related to spin up time?

*Yes,T inversions are often missed by the model from May to August, while the T (see Fig. 5a). The explanation is now better described: "From November to January, KENDA-1/MEE detect most of the near-ground T inversions, which last all day long in winter. Their amplitude is, however, always underestimated by 1-2°C (Fig. 5.b) by KENDA-1/MEE. From February to August, the presence of T inversion at the end of the night and in the first hours after sunrise is often underestimated by KENDA-1/MEE, which can impact the onset time of up valley winds (section Along_Valley_winds). The missed T inversions by KENDA-1/MEE lead to both its important*

*overestimation of the T at ground level (Fig. 4) and its slight T underestimation between ~850-1200 m (Fig. S5 for detailed examples)."*

*We didn't analyze the later forecast hours.*

28. l. 288-289: Wind speed values at which height, time, and station are used to distinguish between days dominated by thermally driven and synoptically driven flows? As mentioned in my general comment, it would be more physical to inspect days for the typical reversal of wind direction twice a day instead of looking at thresholds.

*The 20 km/h threshold is defined for each height and time from the DWL/MEE time series and then applied both the observed and modeled data. The same amount of data is then used for the MWR/MEE, KENDA-1/MEE and KENDA-1/MER. This applied threshold is arbitrary as explained as an answer to the general comment.*

*As specified as answer to the general comments, Fig. 6 was modified, and the arbitrary threshold of 20 km/h is no longer used. A new figure with a selection of days with fear weather was also added to the supplement. We refer the readers to the answer to the second general comment.*

29. l. 295-297: Do the authors have an explanation why the upvalley wind shifts with height? Could this be a result of averaging over a variety of conditions?

*As further explained in section 3.3, the shift in the wind direction above 1000 m is probably a direct influence of the NE wind from the Brünigpass at 1000 m, namely 400 m above the valley floor. This NE wind is observed by the DWL/MEE for all weather conditions (new Fig. 6a= Fig. 3 of this document) and for fair-weather days (Fig. 4 of this document), modeled by KENDA-1/MEE but below 1000 m since the Brünigpass is only 200 m above the valley floor in the model terrain. Surface measurements show that this NE wind can even suppress the up-valley wind at Brienz (Fig. 10 a). We then do not think that it is an effect of averaging over a variety of conditions.*

30. l. 297ff: What is the purpose of providing this amount of detail on synoptic flows? Consider removing or shortening to streamline the manuscript.

*Since the representation with a wind speed threshold was abandoned, this sentence was removed.*

31. Fig. 6: Please add ridge height to the plots. It is really hard to see anything in the panels. Consider limiting y-axis to 2500 m (like for T plots) and rearranging the figures (maybe remove synoptically driven days). What is the temporal resolution of the DWL plots? The same as for the model? Were the winds averaged to 1 hour before computing the monthly composites? What is the point of showing panel c? It is not discussed here.

*Fig. 6 was modified so that it comprises now only 3 rows. The ridges' height was inserted. Fig. 6c with KENDA-1/MER is discussed in § 3.3 (heterogeneity of wind pattern in the Haslital) but is needed here to allow the comparison with KENDA-1/MEE.*

*The temporal resolution of DWL is 1 hour corresponding the average (sect 2 line 105) of raw data each 5 minutes (Sect 2.3.3, line 176). KENDA-1 produces only one data per hour.*

32. l. 307ff: I don't see a good agreement in November. Also there are quite large difference between the observations and the model in summer/spring. The features described here (e.g. N flows from Br¨unigpass) are hardly visible in Fig. 6b in its current presentation. The conclusion 'This feature is mostly caused by the KENDA-1/MEE cell overlapping the slope towards the Br¨unig Pass so that winds at the junction between Haslital and Sarneraatal can influence the median modeled wind compounds.' is not valid in my opinion without a detailed analysis of the model terrain.

> *Fig. 6 and his description were modified. Considering all wind speeds, the agreement between KENDA-1/MEE and DWL/MEE is poor as described now in section 3.2.1. Since Fig. 6 has now only 3 rows, the N flows in red is visible in Fig. 6b.*

> *The reviewer is right, the conclusion if not well formulated and we modified it: "This feature is caused by the lower altitude difference between the topography (400 m) and the model terrain (200 m) and a smaller horizontal distance due to the 1.1 km cells (see Sect. 2.2)."*

33. l. 324-325: Typically, downvalley winds gain in strength throughout the night since the driving horizontal temperature and pressure gradients strengthen. Do the authors have any explanations why this is different here? Could this be a result of clouds forming during the night or a sampling issue? Please discuss this contradiction to theory.

> *The Haslital is a medium size valley and I do not know the usual used terrain (valley width, depth, length and slope, bending, tributaries, etc) involved in theoretical studies. Anyhow the occurrence of a maximum in down-valley speed 2-3 hours after sunset in not only measured by DWL/MEE, but also modeled by KENDA-1/MEE. It is also interesting to note that, at MER, a constant down-valley wind speed during night is both observed by SMN/MER and modeled by KENDA-1/MER at the ground. Fig. 6 (this document) shows that this constant down-valley wind speed at MER is modeled up to the ridge height.*

> *The pressure difference along the valley was not analyzed. It has however to be noted that constant down-valley wind speed during night was also measured and modeled in the Rhône valley at Sion, Visp and in the narrow Magia Valley at Cevio (Fig.4 and 6 in Schmidli and Fig. 11 in Quimbayo, 2021, Schmid et al., 2000) without a discussion about the difference with theory. It seems however that the valley size is not the main explaining factor.*

[Figure]

Fig. 6: Along valley wind speed modeled by a) KENDA-1/MEE and b) KENDA-1/MER.

34. l. 328-334: The described upvalley wind characteristics are not typical. For example, why should the upvalley wind near the surface be stronger and more regular than at 200 m above the ground. This should be discussed. Could this be related to sampling or an artifact of the projection to along valley direction? Would the valley wind system show more typical characteristics when synoptically dominated winds were excluded from the composites? The difference in onset time of downvalley wind at MER and MEE could be related to the Brünigpass. However, this needs to be formulated as hypothesis without sound observational evidence (which is not given in Sect. 3.3).

> *The first column of Fig. 7 (a and c) presents the SMN/MER observations and the modeled KENDA-1/MER at MER whereas the second column (b and d) presents the DWL/MEE observations and the modeled KENDA-1/MEE at MEE. The SMN/MER observations are done at 10 m a.g.l. whereas the other plots correspond to data from 775 m a.s.l., the first DWL level. KENDA-1 data were taken at the same altitude as the DWL to allow a comparison with DWL/MEE, the first KENDA-1/MER level (739 m) being anyhow much higher than the SMN station. Fig. 7 clearly shows that the main difference in both up-valley and down-valley wind speeds is found between both stations and not between different altitudes. MER presents stronger maxima in up-valley wind speed and MEE stronger maxima in down-valley wind speed for both the observation and the model. Fig. 8 confirms that the valley wind speed is usually rather constant along the profiles, at least in the first 1000 m above ground.*

> *We try to select different angles around the valley axis (from +5° to +-15°). These tests as well as a real projection on the valley axis leads to the same result, with stronger up-valley wind at SMN/MER than at MWR/MEE and KENDA-1/MEE. We do then believe that the described up valley wind characteristics are not measurement artifacts.*

> *Further analysis (see answer to the first general comment) explains now the mechanism of flows from the Brünig Pass. Anyhow this last sentence is now formulated as a hypothesis.*

35. l. 335-336: The statement that turbulence is leading to daytime varying wind direction is not obvious and needs to be supported by observational evidence or removed.

*Yes, the mention of turbulence was removed.*

36. Fig. 7: Please enlarge axis labels. Replace MWR/MEE in caption with DWR/MEE. What values are plotted? Are these hourly values or monthly values? If monthly why not plot hourly values to show day to day variability?

*The axis labels are enlarged and the figure caption corrected. Monthly values are plotted since the noise induced by the daily data masked the main features.*

37. l. 338: 7b instead of 7c.

*Done*

38. l. 340-343: This atypical behavior of the valley wind (upvalley wind during the night, upvalley wind during all day in winter) in KENDA-1 is possibly a result of averaging over all types of conditions. It is hence not clear if KENDA-1 struggles with thermally driven flows or channeling events or on clear or cloudy days. Filtering and focusing on specific conditions would be beneficial to learn more about model deficiencies.

*This comment is right, and the present analysis does not allow us to find the causes of the model deficiencies. We hope that colleagues from the modeling community will use our dataset to further test the models, but this is beyond the scole of this paper.*

39. l. 344-354: In my opinion, it is not valid to draw conclusions for up- and downvalley winds and on impacts of synoptic flows from the monthly composites, since the composites are most likely strongly affected by synoptic winds and clouds. To draw meaningful conclusions on up- and downvalley winds, the days need to be filtered for conditions favorable for thermally driven winds. Also, some of the aspects (decrease of downvalley wind speed with height) could be a result of projecting on the valley axis and should be investigated. Without investigating other factors (clouds, synoptics, sampling size, etc) it is not valid to attribute varying wind direction during daytime to turbulence.

*As answered to the first general comment, no projection but a selection of wind around the valley axis are applied to compute the along valley wind compound. There is consequently no artifact from the projection.*

*Fig. 3 (this document) and further analysis with cloud cover <3 oktas at FRU showed that the described features are similar with a selection of clear and very clear days. Moreover, we think that thermal wind occurs not only during fair weather days but also, with a lower intensity, under less good conditions. A selection of only very good weather situations also restricts the analysis of thermal valley wind.*

*The mention of turbulence to explain varying wind direction during daytime was removed.*

40. Fig. 8: How are the composites computed? Are KENDA-1 data only used when observations are available and valid? If not, the comparison is not fair. Are white gaps in panel a) due to small wind speed or missing data? A different color should be chosen for missing data (e.g. grey).

*Fig. 8 was corrected to consider only data present in both time series. White gaps correspond to missing data and low wind to light green. The text was then revised according to the new Figure.*

41. l. 366: What does 'data are scarce' mean? Shouldn't the sample size for across valley wind be the same as for along valley wind?

> *No, the lower amount of data in Fig. 9 is due to the absence of wind in the cross valley direction. As described in the experimental section, too weak winds (speed < 2km/h) were discarded for wind direction analysis.*

42. l. 366-367: The northerly flows in January and February are not clearly visible.

> *Northerly flows come from the south-facing slope, namely from the Sarneraatal and are depected in red. We think that they are visible on the figure as indicated by the red square on the next Figure:*

[Figure]

*Fig. 7 Evolution of the diurnal cycle of the cross-valley wind component [km/h] as a function of altitude for a) the DWL/MEE measurement and b) the KENDA-1/MEE. Winds coming from the south-facing slopes take a positive value (red), for the north-facing slope wind speeds values are negative (blue). Sunrise and sunset at ground level are given by dotted lines.*

43. l. 370: What are north-facing slope winds? Do the authors mean downslope winds from the slopes north of the valley (they are south-facing)? For clarification, it would be helpful to repeat the colors used in Fig. 1 to distinguish southerly and northerly cross valley flow.

> *As explained by the reviewer, north-facing slopes lie south of the valley. The sentence is perhaps misleading and was modified: "Intense winds from north-facing slope (> 25 km/h) are also observed between 1400 and 2000 m during some hours around sunset with a much lower intensity in May." The colors correspond to the colors used in Fig. 1 and are described in the figure caption of Fig. 9 presenting the cross-valley winds.*

44. l. 371-374: In my opinion there is not enough evidence for a cross valley vortex from the example RHI plot in Fig. S9. I can see downslope and upslope components, but no closed circulation.

*The instrumental set-up does not allow us to obtain a 3D image of the wind compound (see answer to the first general comment). The radial wind depicted on Fig S9 presents however a clear example of a cross valley circulation and we do agree that the vortex nomenclature is misused so that it was not used in the manuscript. We then replaced it with cross valley circulation in the supplement.*

45. l. 375: What color are the winds that descend from Brünigpass? Are they red? Please specify.

*As specified in the caption of Fig. 9, winds from the south facing slope are taken as positive, namely positive. The Brünig is situated on the ridge of the south facing slope (see Fig. 1) so that winds from the Brünig are depicted in red. This is now more precisely specified in the figure caption:*

*"Winds coming from the south-facing slopes, namely from the Brünig Pass, take a positive value (red), for the north-facing slope wind speeds values are negative (blue)."*

46. l. 380-383: Why is the along valley wind mentioned here? This section is about cross valley. It needs appropriate context and reference.

*This sentences and the next ones are not relevant here and were deleted.*

47. Fig. 9: How are the composites computed? Are KENDA-1 data only used when observations are available and valid? If not the comparison is not fair. Are white gaps in panel a) due to small wind speed or missing data? A different color should be chosen for missing data (e.g. grey). What is the sample size at each point (can some of the noise be explained by varying sample size)?

*As explained concerning Fig. 8, Fig. 9 was also modified so that only data present in both time series are plotted. White dots correspond also to missing data of no across valley wind. The text was also adapted to the new figure.*

48. l. 394: On this fair weather day, wind speeds of 25-30km/h are reported. How does this fit to the filter of 20km/h to distinguish thermally from synoptically driven days?

*The threshold of 20 km/h is no longer used in this revised manuscript.*

49. l. 399-401: The outflow from Brünigpass cannot be thermally driven. Why should warm air during the day descend from the pass to MEE? If there is upvalley wind in Sarneraatal that reaches the pass and descends on the south side of Brünigpass there must be a dynamic effect driving this (wave, etc). Please explain.

*As explained to general comment 1, a new analysis showed that the air masses in the Sarneraatal are often colder than in the Haslital over MEE. The outflow from Brünig pass is then thermally driven and descends to MEE because it is colder than air masses above at the altitude of Brünig pass. The valley volume effect can explain the T differences between both valleys. This is now explained in the manuscript.*

50. l. 409-418: This whole paragraph is based on Fig. S8. If this figure is so important, it needs to be included in the manuscript.

*The new analysis of the pressure difference between GIH and MER and the potential T difference between BRU and MWR/MEE at 1010 m deserves a new figure. The inclusion of Fig. S8 would too much lengthen the manuscript.*

51. l. 419ff: It is essential to include the model terrain in this analysis to understand how the model sees the Sarneraatal and the pass.

*The height of the ridge and the altitude of the pass are given in the experimental section. In the model, the Brünig Pass is only 200 higher than the valley floor, enhancing the potential impact of the Sarneraatal on the wind system in the Haslital.*

52. Fig. 10: Both panels should have the same y-axis range.

*In fact, both panels have the same y-axis range. It is not obvious since the wind direction stripes are pasted on the figure and that Fig. 10a provides 3 wind direction stripes and Fig. 10b only two.*

53. l. 439-441: Specify where clear weather conditions can be expected. Describe the foehn characteristics at MER (direction, over which ridge it is coming). What stations are used to compute the foehn index?

*Clear weather conditions are expected during the whole foehn event on the northern side of the Alps' ridges. The foehn at MER comes from the Grimsel Pass and follows then the Haslital. This is now specified in the manuscript.*

*The main prerequisite for the occurrence of foehn on the northern slopes of the Alps is a southerly wind on the main Alpine ridge, which is measured at the Gütsch station, Andermatt, (GUE). Conversely, on the southern slope of the Alps, the wind on the main Alpine ridge must come from the north for foehn to occur. The other parameters (average speed, wind gust, wind direction, relative humidity and potential T are taken from SMN/MER as described in sect. 2.3.1.*

54. l. 442: No June episode shown.

*The foehn event in June is considered in the analysis of the T, while only three events (10-16 March 2022/19-22 March 2022/26-24 April 2022) are described in the analysis of the wind. The sentence was modified.*

55. l. 445 and Fig. 11: Unless foehn starts always at the same time of the day, the composites should be shown relative to foehn onset and not for hour of the day.

*This figure intends to show that the diurnal cycle of the T difference between KENDA and the measurements is not found during foehn events and that the T is always overestimated by KENDA-1.*

56. l. 453ff: This whole paragraph is again based on Fig. S11 and S12 from the supplement. If this is discussed in so much detail, it needs to be included. However, related to my comment on adding more focus to the paper, I think the whole discussion on foehn should be much shortened or even removed. I also cannot see a T gradient in the types of plots in the supplement.

*We consider that the modeling of foehn events is still challenging so that the description of foehn events should not be removed but the description of the wind during foehn event was shortened. Figure S11 and S12 are of clear interest and are not really discussed in detail, so that they must remain in the supplement not to lengthen the paper.*

57. l. 465-466: I don't see the point in comparing different heights at MEE and MER.

*The sentence was modified. As it was intended, the point is now focused on the comparison of the maximum wind speed: "The maximal wind speed (60-75 km/h) of DWL/MEE is observed at 800 m and is much higher than at the SMN/MER (45 km/h), especially for the event of March 11."*

58. l. 467-484: I think this description of wind is way too detailed and distracting and could be much shortened or removed.

*Sect. 3.4.2 was largely shortened and consists now of one § to describe the SNM/MER and DWL/MEE observations and one § to describe the differences between the observations and the modeled data.*

59. l. 485-494: Given that KENDA-1 provides 3D output, the foehn cases could instead be investigated in KENDA-1 to understand the spatial differences and the model errors during foehn events.

*This is out of the scope of this paper. The aim of the paper is to compare REM measurements with KENDA-1 and not to study the foehn event in the Haslital. Since a poor agreement is found between KENDA and the measurements, we cannot rely on the 3D KENDA data to study to have a realistic picture of the foehn event in the Haslital.*

60. Discussion: The discussion section is too long in my opinion and should be more focused. For example, a lengthy comparison to inversions and thermally driven flows in other studies is shown, but given that the composites in the present study are not filtered for thermally driven flows this comparison is not very meaningful. The evaluation of KENDA-1 was done visually based on time-height sections and not by computing model skills. This would have been more meaningful, instead of the descriptive comparison of the model to the observations. I think saying that 'KENDA-1 proposes good monthly median values' (l. 662) and 'Despite the complex topography around MER and the induced elevation bias, the modeled climatology of ground T is comparable to standard verification results' (l. 595-597) is hence not sufficiently supported by the analysis.

*As already mentioned before, thermally driven wind occurs not only during fair-weather days. We think that a comparison with the three cited studies is necessary. The discussion was a little shortened.*

*The evaluation of KENDA-1 with computed skills would have been strengthened. The visual evaluation is still valuable. The mention of standard verification results was removed.*

61. l. 574-584: The vortex in the Inn Valley was caused by the valley curvature (Babi´c et al., 2021). This means that the mechanisms here are likely not comparable and caution is advised.

*Yes, the vortex described by Babic et al, 2021, is caused by the valley curvature. The valley curvature between MEE and MER is situated between both stations and not exactly at the DWL site. Secondly, the*

*observed cross valley circulation is certainly influenced by the winds from the Brünig Pass. The § was modified to enhance the differences between both environments: "However, contrary to the CROSSIN campaign's results, valley winds from the Sarneraatal are probably the main drivers of this cross valley circulation in MEE."*

62. l. 585-587: No skills are computed for KENDA-1 and an objective verification of model skills was not done. This conclusion is hence not supported by the presented results.

> *Since no skills were computed for the used locations, we omit this sentence.*

63. l. 595-597: The reference to standard verification results is confusing and needs to be explained and justified by results.

> *This sentence has been rephrased and a reference has been added.*

64. l. 599-600: Since monthly composites are shown this statement is not supported by results.

> *This statement comes from an in-deep analysis of the cold event at the end of March that is not reported in the manuscript. "Result not shown" is now added to the sentence.*

65. l. 605-617: RH depends on T. Thus, a warm bias leads to a dry RH bias. The statement on RH does not provide additional information. Bias in terms of specific humidity would be more meaningful.

What does 'artifacts from the NWP can be expected under conditions favorable to surface T-inversion' mean? The statement 'Finally, the differences with observations can also originate from a modeled ongoing turbulent mixing whereas in reality a cold pool with a full or partial decoupling from the above flow is present in the valley.' is not supported by results.

> *The sentence has been reformulated as a hypothesis.*

66. l. 618ff: MWR liquid nitrogen calibration plays a role in MWR profiles biases and should be discussed. Average differences are discussed, but what about individual profiles?

> *In our experience, after a successful calibration with liquid nitrogen, the biases become negligible.*

> *Some individual profiles are shown in the supplement. However, the goal of the paper is to perform statistics, not to discuss individual profiles.*

67. Conclusions: Since many readers only read the Summary some basic information on sites and data should be repeated. I don't think that all conclusions are sufficiently supported by the results (e.g. l. 688-690, l. 693-695). I furthermore do not think it is fair to say that the study 'deepens our consensual knowledge about atmospheric phenomena in complex topography'. It is mostly a description of conditions without any in depth investigation of processes.

> *The conclusion was modified following the recommendations. Basic informations on sites and instruments were added. The sentence on cross valley circulation was modified and the words "vortex" and "closed circulation" were removed. We consider however that the measurements showed a cross valley circulation. The last sentence was also modified.*

---

## Author Response (AR3)

*Answers to the reviewer 2 technical corrections on*
*"Comparison of temperature and wind between ground-based remote sensing observations and NWP model profiles in complex topography: the Meiringen campaign"*

*We thank the reviewer for the hopefully last technical corrections to our manuscript.*

Technical corrections:

- L. 43 Subscale processes → Do the authors mean subgrid-scale processes?
  *Yes, this was corrected*
- l. 136 Kilometre-scale Ensemble Data Assimilation --> Please replace by 'KENDA' as it is already introduced
  *Done*
- Section 3.1.3 There seems to be a latex error. Please correct.
  *Corrected*
- l. 445 at GIH and MER (10.a) --> at GIH and MER (Fig. 10.a)
  *Done*
- l. 449 10.b shows the difference between ... --> Figure 10.b shows the difference between ...
  *Done*
- l. 529 As described in 3.3 --> As described in Section 3.3
  *Done*
- l. 681
  According to (Schmidli et al., 2018) -> Please remove brackets
  *Done at l. 690*
- l. 710
  This diurnal flow pattern develop --> This diurnal flow pattern develops
  *Done*
- l. 713
  simultaneous for the entire the profile --> Please remove the second 'the'
  *Done*
- Figure 7.
  The lower axis label is cut off.
  *Done*

*Answers to the reviewer 3 comments on*
*"Comparison of temperature and wind between ground-based remote sensing observations and NWP model profiles in complex topography: the Meiringen campaign"*

*We thank the reviewer for the in-depth comments to our manuscript.*

*The answers to the comments and questions are written in italic thereafter. The explanations of this document cite the numbering of the figures in the first revision in accordance with the lines' numbers of the comments.*

This review is for the second revision of the manuscript. I appreciate that the authors addressed my and the other reviewer's comments in this revised version. It is improved and clarified in some aspects such as the mechanism for the flow from the Sarneraatal over the Bruenigpass and the arbitrary threshold of 20 m/s to select thermally driven days. However, there all still areas that need improvement in my opinion, especially related to conciseness and preciseness.

The manuscript contains a lot of description and speculation, making it cumbersome to read. The description of the figures in the manuscript is very detailed with numerous mentions of heights, times, and values. This amount of detail might be appropriate for a report, but is somewhat distracting in a scientific article. The manuscript could be more precise and concise, if it focused on relevant aspects (e,g, instead of describing every little detail of the figures, it would help to focus on the aspects directly relevant for the objectives of the manuscript) and using a clear and precise wording. For example, like in these two sentences 'Concerning KENDA-1 data, the foehn breakthrough is modeled too early on March 11 at both stations, on time on March 20 at both stations and on April 23 at MER and too late on April 23 at MEE. The foehn arrival and end is modeled sometimes on time by KENDA-1, but positive and negative time shifts of up to 4h at both stations' (l. 511-514). This could be combined and be shortened: 'KENDA-1 models the foehn breakthrough 4-h too early at both stations on March 11, on time at both stations on March 20, and 4-h too late at MEE on April 23.' Cases like this are present throughout the manuscript and I strongly suggest that the authors try to use a more precise and concise language.

As mentioned in my previous review, I think a shortening of the case studies and the discussion with a focus on the most relevant aspect would be beneficial. Instead of shortening the case studies, the authors added additional analysis on the responsible mechanisms for the flow descending over the Brueningpass. I think this analysis adds value and helps to understand the observations, but, without shortening other parts of the manuscript (like the discussion or description of the figures), results in a rather long manuscript (of 39 pages with 13 figures in the main manuscript and 13 in the supplemental).

More attention should still be paid to details (correct formatting). For example, the formatting is off in the first paragraph of Sect. 3.1.3. Also, times should have units (e.g. UTC) which are consistently use throughout the manuscript. Sometimes a.m. is used. For all heights which are with respect to mean sea level 1 'a.s.l.' should be added. For times above ground, a.g.l. should be used. This is currently very inconsistent.

In addition to these general comments that should be addressed, I am giving some specific comments and suggestions below.

*The modifications of l. 511-514 were done and adaptation to of the manuscript to more precise and concise language have been done throughout the results section leading to a reduction of ~10% of the text. The authors think that a further shortening of the manuscript would be at the detriment of a correct description of the complexity of the described phenomena.*

*Concerning the time and altitude units, the authors chose the option to mention at the beginning of Sect. 2 that times are in UTC and, if not specified, altitude in m a.s.l. In our opinion, these options make the text lighter, easier to read and allowed by AMT. The authors also paid further attention to correct details such as formatting or mis-spelling.*

**1 Specific comments**

• Title: The placement of 'profiles' in the title sounds a bit off. Perhaps it would be better to say 'Comparison of temperature and wind profiles from ground-based remote sensing observations and a numerical weather prediction model in complex topography: the Meiringen campaign'.

> *Done*

• l. 55: Slope winds are driven by a horizontal temperature gradient between the air adjacent to the slope and the free valley atmosphere'

*The authors checked several reference papers and books on slope winds. If buoyancy generated by temperature gradients are systematically mentioned as driver for slope winds, the direct mention of the direction of the T gradient is almost never mentioned. Convection processes and gravity linked to higher air density (namely corresponding to vertical processes) are often mentioned for anabatic and katabatic winds, respectively. The sentence at line 55 was then modified to solely mention "by temperature gradients" without specifying the direction of the gradients.*

• l. 62-64: What is meant by wind intensity? The reversal from upvalley to downvalley winds in the evening is driven by the reversal of the along valley temperature and pressure gradient. Please clarify.
*Wind intensity was replaced by wind speed.*

*The reversal from up-valley to down-valley winds is now correctly attributed to T and P gradients:" In the evening, as soon as the surface radiative balance becomes negative, the cold air forming at the surface moves down the slope and converges in the valley floor. After the reversal of the along-valley T and pressure gradients, the flow direction shifts from up-valley to down-valley winds."*

• l. 85-87: Please be more specific. Why is precise knowledge essential for NWP? To evaluate and improve the models? And why are REM a solution? Are they assimilated or used for evaluation?
*The sentence was modified: "However, the spatiotemporal heterogeneity of T in complex terrain is challenging for NWP models and the use of REM observations is a solution to evaluate the models and improve them by the assimilation of observed profiles."*

• Fig. 1: Please add that BRZ stand for Brienz. It would also be helpful to add the names for BRU, LUN, BUC, and GIH to the caption.
*Done*

• l. 109: 'a.s.l.' is already used in l. 94.
*Yes, the use of 'a.s.l.' in l. 94 is necessary to estimate the altitude of the topographic features at the site of the campaign and the mention at l. 109 allows to specify the rule applied throughout the paper. The abbreviation a.s.l. is now introduced at l. 94.*

• l. 130: 574 m?
*Thanks, the altitude of MEE is 589 m but the one of MEE is 574m. This is now corrected.*

• l. 141-142: Is the terrain shown in Fig. S3 filtered with this 2dx filter? It looks very steep with large differences between adjacent cells. To be meaningful, the terrain that is actually used by the model should be shown.
*Yes, the 2dx filter has already been applied to the surface topography represented in Fig. S3 and this is exactly the terrain used by KENDA-1/COSMO-1 model.*

• l. 180: SMN was already introduced.
*Done*

• l. 183: Where is FRU? Not included in Fig. 1. How is the cloud amount estimation done at FRU?
*The coordinates of FRU are given in the manuscript. It is now further mentioned that FRU is situated south of Lake of Thun. The cloud amount is estimated by measurements of longwave downward radiation, temperature and relative humidity with a time resolution of 10 min (Automatic Partial Cloud Amount Detection Algorithm, APCADA, Dürr and Philipona, 2004).*

• l. 185-186: Are only wind observations used at BRU? What about temperature? What is 'similar temporal resolution'? Order of minutes or hours?

*No. Lines 179-181 mentioned that wind observations are also performed at all SMN stations including MER, BRZ and GIH and l. 185-186 clearly mention wind observations from FEDRO also at LUN und BUC with similar temporal resolution, namely hourly observations.*

• l. 199: Please add information that line of sight of about 10 km is in downvalley direction.
*Done*

• l. 228: What is 'end of winter'? Was there snow on the ground after mid-December?
*There was no further snow cover higher than 15 cm (that corresponds to a homogeneous snow coverge) after mid-December at MER. "end of the winter" was then replaced by "end of the spring" to be more precise.*

• l. 230: I would appreciate if a brief description of how the foehn index works was included.
*The main prerequisite for the occurrence of foehn on the northern slopes of the Alps is a southerly wind on the main Alpine ridge, which is measured at the Gütsch station, Andermatt, (GUE). Conversely, on the southern slope of the Alps, the wind on the main Alpine ridge must come from the north for foehn to occur. The foehn index is calculated individually for each station using different measurements. There are fixed threshold values for each parameter. These enable the index to be objectively calculated for each station, depending on where it is located and how prone it is to foehn. The parameters used (measured every ten minutes) are average speed, wind gusts, wind direction, relative humidity and potential temperature. The potential temperature is a hypothetical value that the air would have if it were measured at sea level. It is calculated from the temperature and the air pressure.*
*A complete description on how the foehn index is calculated in the Alpine valleys can be found at https://www.meteoswiss.admin.ch/dam/jcr:3ed2aec8-0901-417a-acc3-8be11cce440a/Foehnindex_Arbeitsbericht_223_Automatisiertes_Verfahren_zur_Bestimmung_von_Foehn_in_Alpentaelern_de.pdf*
*Since a detailed description is quite long, the authors prefer not to include it in the manuscript.*

• l. 250: Radio soundings are not mentioned before. Where and when were they performed? What cold bias?
*The radio-sounding is already mentioned in Sect. 2.2 since it is assimilated by KENDA-1. In Switzerland, radio-sounding are only performed operationally at Payerne on the Swiss plateau. Since a MWR is also available at PAY, comparison between RS allows evaluating the MWR performance. The first version of the manuscript had a small section on three radio-sounding performed in November 2023 at MEE, but is was suppressed to shorten the manuscript.*

• l. 257: At what height are the statistics computed?
*The heights used for the statistics of Fig. 3 are described in the figure caption: "The lowest level corresponds to 576 m for SMN/MER, 625 m for MWR/MEE and 705 m for KENDA-1/MER and 739 m for KENDA-1/MER."*

• l. 283ff: Why are still differences used? In the response, the authors indicate that they now use gradients, but this is not clear. In Fig. 5b, the temperature difference is still shown (unit deg C). What is T inversion amplitude? Is this inversion strength? Amplitude is a bit uncommon in this context. In the response to the reviewers' comments (comment 25), the authors claim that they now include the potential sources of error when comparing ground-bases and free atmospheric observations ('T inversions observed on the ground ...'). However, this explanation does not occur in the manuscript.
*The word' amplitude' was replaced by 'strength'.*
*The answer to the previous comments were perhaps misleading. Fig. 2 a presents a T gradient (a temperature difference divided by an altitude difference) whereas Fig. 5 present the frequency of T inversion computed from the T difference and the strength of the T inversion given as the T difference. The measurement campaign set-up involves different lowest level altitudes for the ground observations, the REM observations and the model. The use of gradients does not allow to get ride of these differences, because the gradients will be computed on difference height differences. KENDA-1 has the greatest height*

*difference of the lowest level that corresponds effectively to the presence of the ground (associated with its physical effect) at this altitude. We can see from Fig. S6 that KENDA-1 T profiles differs from MWR profiles from the lowest level up to ~1500 m in case of missed T inversion. The underestimated strength of the T inversion modeled by KENDA-1 would appears smaller on a figure with a gradient due to the division by a smaller height difference than for MWR/MEE or the pair of ground stations. However, the discussion on the differences between the different set-up cannot be avoided. A new sentence warns the reader of this difference: "The higher altitude of KENDA-1/MEE lowest level results in a lower inversion strength but explains only 30\% and 40\% of the difference with MWR/MEE and BRU-MER pair, respectively.".*
*In our response, we say that the potential sources of errors are described, which was done in Sect 3.1.3 but was not readable due to an overleaf error. They are however not taken into account in Fig. 5.*

• Fig. 2. Please make plots in a) and b) the same size for consistency. How is the monthly diurnal cycle computed, is it mean or median? Please add. Is the difference in b) computed before or after the monthly means/medians for the observations and KENDA are computed? Do sunrise and sunset times account for orography? Are they monthly means/medians? How is the mean ridge height computed?
Is this based on the lines in Fig. 1? Please explain.

*The plots of Fig. 2 have now the same size.*
*As already explained in Sect 2, the monthly averages are aggregated according to the median hourly values. An addition of the way of averaging each time they are used would make the text too heavy.*
*The median of the hourly T differences is presented in Fig. 2b. We tried both solutions. The difference of the monthly T median leads to larger values but with similar monthly diurnal cycles.*
*In this case, the sunrise and sunset time does not take the shading into account. In the first version of the manuscript, Fig. 9b presented sunrise and sunset including the shading at MER that was computed from the real topography and not from the lines in Fig. 1. The shading effects affect mostly sunset time between October and March. Sunrise time is almost not affected. We decided not to use systematically the shading to compute sunrise and sunset because 1) they are different at MER and MEE as well as at the different altitudes and 2) phenomena as T inversion of thermal winds depends not only on the shading at the station site but on the effect of solar radiation in the whole valley.*
*The monthly sunrise and sunset correspond to the monthly mean that is approximately similar to the median since the variation is quasi linear without any outliers.*
*The mean ridge height corresponds to the mean of the ridge from one km upstream to on km downstream from MEE station.*

• l. 364-465: This sentence is not clear.
*The sentence is not relevant and was deleted.*

• l. 367: Not clear, why vertical transport is important for weakening of drainage flows.
*You're right, this explanation is not relevant and was removed.*

• Fig. 7: Do sunrise and sunset take orography into account? They are different from sunrise/sunset in Fig. S8. Better show the one taking orography into account.
*No, the orography is not taken into account (see previous explanation on Fig. 2). The sunrise and sunset hours are the same as in Fig. S8, which however presents a complete seasonal cycle (Jan-Dec) whereas Fig. 7 presents only the months of the Meiringen Campaign.*

• l. 373: 'underestimation of wind speed' is not clear. Upvalley wind is actually stronger (positive values) in KENDA. What is missing are downvalley winds.
*The sentence was modified since it relates to the downvalley wind speed in November and December: "A comparison of the first level of KENDA-1/MER and SMN/MER (Fig. 7 b and a) indicates an underestimation of downvalley wind speed by KENDA-1/MER, leading to the absence of a diurnal cycle in November and December."*

• l. 376-377: This sentence implies the stronger presence of upvalley wind is leading to weaker downvalley wind speed, which is not physical. Please rephrase.

> *The sentence was modified: "The modeled data at MER and MEE also show distinct differences, a stronger up-valley wind speed in MER, a weaker down-valley wind speed and the presence of weak up-valley wind during the entire days in winter."*

• l. 382: The phrase 'onset is anticipated compared to' is not clear.

> *The sentence was modified: "The onset of down-valley winds near the ground happens earlier than at higher altitudes so that up-valley winds can persist until 1-3 h after sunset above 1500 m."*

• l. 384-385: What about moist convection? Surely there are convective clouds and precipitation during the summer months that can affect the flow.

> *There is surely moist convection, a development of convective clouds and precipitations, but there is no reason why they should affect the wind direction only between 1000 and 1500 m. I would expect an influence on the whole profile of the wind direction.*

• l. 392-393: How can synoptic winds lead to continuous downvalley wind underneath. Please rephrase.

> *The sentence was modified: "Finally, in winter, KENDA-1/MEE overestimates the influence of the synoptic winds, which leads to the presence of homogeneous up-valley winds down to 1000 m, and models continuous down-valley winds underneath."*

• Fig. 9: Please add a) and b) to the figure. 'wind speed values'.

> *Done*

• l. 422-424: Weird sentence. Please rephrase.

> *The sentence was modified: "The observed wind speeds during a series of clear warm days in July with low cloud coverage (Fig. 11) present a wind pattern in the Haslital, which is undetectable in the analysis based on monthly medians."*

• l. 443: Figures should appear in the same order as they are mentioned.

> *Done*

• l. 449ff: Annotation (x, y, θ) is not clear. Is GIH at the same height as BRU? If not, how is pressure at GIH used to compute potential temperature at BRU? How is the valley volume computed?

> *The annotation (x, y, ϑ) was replaced by (ϑ). GIH is at a lower altitude than BRU. The pressure is not measured at BRU, so that the pressure at the BRU altitude was computed from GIH ground pressure with the barometric formula. This is now mentioned in the manuscript.*
>
> *The valley volume is computed from the integral of the difference between the ground and the mean ridge height for each pixel of the topography. It is specified that the volume is an approximation.*

• l. 482: Foehn is NOT a katabatic wind. Is Haslital on northern side of Alpine ridge?

> *Yes, foehn is not a katabatic wind. The Haslital is influenced by South foehn and is then considered to be on the northern side of Alpine ridge (see next figure for the influence of south and north foehn, from https://www.meteoswiss.admin.ch/weather/weather-and-climate-from-a-to-z/foehn.html). The sentence was consequently modified: "South alpine foehn is a strong wind that brings warm and dry down-valley wind and leads to clear weather conditions on the northern side of the Alpine ridge."*

[Figure]

Typical valleys of Switzerland with southerly foehn (red) and northerly foehn (blue). The areas in which foehn is rarely observed are marked with a dashed line. (© MeteoSwiss)

• Fig. 12a: Are sample sizes similar for each hour? With only 117 hours of foehn detected, this would mean that there are only 5-6 samples per hour. Pretty small sample size to compute distribution and to draw meaningful conclusions. This limitation should be mentioned.

*The numbers n at the top of each figure correspond to the number of cases in each category. Fig. 12a has effectively only 3-5 cases per hour whereas Fig. 12b has between 13 and 31 cases per wind speed category. It is now specified in the figure caption that "The limited number of cases per hour in a) involves a higher uncertainty in the results.".*

• l. 509-510: This comparison is hindered by the fact that the value at SMN/MER is observed at the surface and the value at DWL/MEE at 800 m a.s.l.

*Yes, but the SMN/MER is at 574 m a.s.l., namely 200 m below the DWL/MEE first level at 775 m a.s.l. It seems that the different sites (4 km in horizontal distance and 30° difference in the orientation of the valley) is as important as the height difference. We think that the differences in height, horizontal distance and terrain between MEE and MER are sufficiently mentioned in the paper, so that it is not needed to repeat it here. Moreover the next § describing KENDA-1 results mentions these differences.*

• l. 529: Please formulate as hypothesis, 'can explain'.
*Done*

• l. 531-532: Bise is mentioned here for the first time. How are Bise situations determined? The enhancement is not documented.

*Bise situation are determined by the speed of the north-, northeast- or east-wind over the northern Alpine foreland which normally occurs in the presence of northeast-southwest 850 hPa pressure gradient in the order of 1 hPa/100 km.*
*Wanner and Furger, Meteorology and Atmospheric Physics, 43(1):105-115, DOI:10.1007/BF01028113).*
*The sentence was modified :" This phenomenon can be enhanced in case of Bise situation, a N-NE synoptic winds that occurred on 35 days in the January-August 2022 period."*

• l. 546ff: Please explain why monthly medians are used and not temporally resolved values. Is this because of noise? The KENDA-1 analysis is used, i.e. the forecast skill is not investigated.

*As explained in the next sentence, this analysis focused on climatology so that monthly medians were principally used. Temporally resolved values are still used 1) for case studies such as the foehn events or the peculiar wind pattern during hot summer days and 2) statistical analysis such as the diurnal cycle of T differences (Fig. 3-5).*

• l. 566ff: Several studies focused on valley wind in the Inn Valley. A classic one that would be good to include is Vergeiner and Dreiseitl (1987).

> *It is now specified that only studies in the Alps and using REM instruments are cited in Sect. 4.2.2 since the main feature of this paper is the use of REM technology in a medium size alpine valley. Since the referee find important the citation of this paper, it is now cited in the introduction (l. 65 of the new manuscript).*

• l. 579: Where is Sion?

> *Sion is in the Rhone valley. Since the precise location of the campaign is not cited, we removed the mention of Sion and replace it with "the Rhone valley".*

• l. 602ff: A study focusing specifically on the vortices in the Inn Valley is Babi´c et al. (2021).

> *Yes, I know quite well the excellent publication of Babic et all, 2021. The measurement program during the CROSSINN campaign allowed to evidence cross-valley vortex. The modest set-up in Meiringen does not allow such a detailed analysis of the circulation in the Haslital. Babic's paper is not cited since its results cannot be directly compared with the results presented in this paper.*

• l. 626: What is 3?

> *The mention of "Sect." was accidently omitted. It is now corrected.*

• l. 637: The sentence 'Westerhuis et al. ....' is not clear.

> *The sentence was merged in the next one: "Westerhuis et al., 2021} showed, that, in complex topography, numerical artifacts may originate from the intersection between T inversions and the surface of the vertical grid used by the model."*

• l. 655: Since the information content from the passive microwave radiometer decreases with height, the vertical resolution decreases and inversions and elevated layers are smoothed with height (e.g. Crewell and L¨ohnert, 2007).

> *Yes, this is right and the paper of Crewell and Löhnert, 2007 is now cited.*

• l. 670: Where is Visp? Without more detail this comparison does not make sense.

> *The manuscript already specify that Visp is in the Rhone valley. The exact coordinates are now given.*

• l. 671: 'four-time shorter length of the Haslital' compared to which valley?

> *It is now specified that the comparison is done with the Rhone valley.*

• l. 682-683: Differences in real-word and model valley depth certainly are also important.

> *Yes, the authors completely agree with this statement.*

• l. 697: 'from November 2021 through August 2022'.

> *Done*

• l. 713: Please rephrase '... simultaneous for the entire the profile.'

> *Done*

**References**

Babi´c, N., Adler, B., Gohm, A., Kalthoff, N., Haid, M., Lehner, M., Ladst¨atter, P., and Rotach, M. W.: Cross-valley vortices in the Inn valley, Austria: Structure, evolution and governing force imbalances, Quart. J. Roy. Meteor. Soc., 147, 3835–3861, doi:10.1002/qj.4159, 2021.

Crewell, S. and L¨ohnert, U.: Accuracy of boundary layer temperature profiles retrieved with multifrequency multiangle microwave radiometry, Geoscience and Remote Sensing, IEEE Transactions on, 45, 2195–2201, doi:10.1109/TGRS.2006.888434, 2007.

Vergeiner, I. and Dreiseitl, E.: Valley winds and slope winds - Observations and elementary thoughts, Meteor. Atmos. Phys., 36, 264–286, doi:10.1007/BF01045154, 1987.

---

## Author Response (AR4)

Answer to the editor comments on "COMPARISON OF TEMPERATURE AND WIND PROFILES BETWEEN GROUND-BASED REMOTE SENSING OBSERVATIONS AND NUMERICAL WEATHER PREDICTION MODEL IN THE ALPINE COMPLEX TOPOGRAPHY: THE MEIRINGEN CAMPAIGN.

*The authors thank the editor for accepting the publication of the paper and for taking the time to review the paper. The answers to the comments are written in italic thereafter.*

I read with interest the paper and I notice the great improvement in the text, which I now find easy to follow. In my opinion, you nicely responded and modified the work based on all the comments from the other reviewers. I just suggest some smaller modifications, to facilitate reading.
I suggest publishing after taking care of these final minor corrections:

1) line 89: you introduce the campaign in the Haslital here in the introduction. I suggest to provide here some small details (period, location), instead of postponing them in the methods and data section. It helps to give context, otherwise the reader has no idea about which campaign it is.

> *The following sentence was added to the introduction :" The measurement campaign took place from November 2021 to August 2022 at Meiringen, a small alpine village in the Haslital."*

2) lines 180-190: please refer to figure 1 when you mention the additional station from which you take data from. In this way the reader can find where they are located.

> *Done*

3) line 193: how do you identify rainy conditions in the MWR? beware of wet radome effects after rain is over. Providing some details here does not hurt.

> *The producer of RPG Radiometer equipped the with precipitation sensor so that the data are flagged in case of rain. At the end of the precipitation event, a ventilator removes the remaining water droplets. The radome is moreover change two time a year to ensure a good removal of droplets. The following sentence was added to the manuscript: "Precipitation is detected by the MWR and the radome is ventilation thereafter. The data acquired during rainy conditions are consequently discarded."*

4) line 240 and around; I would start presenting the results from the results that are visible in the figure included in the paper, and then add the points coming from figures in supplementary material. The sentence "the maximum temporal gradient T.... " refers to figure S5 before figure 2 is introduced and we get a feeling of the situation you describe

*I do not understand your remark since the two first sentences of the paragraph describe Fig. 2a and Fig. 2a is mentioned in the first sentence. There is then a sentence describing a result visible in the supplement to introduce T gradients. Finally, T inversions, which are a peculiar sort of gradient visible in Fig. 2a, are further described. The requirement of presenting first the results of the figure in the manuscript is therefore fulfilled.*

5) For all plots: be consistent with the label of the time axis. Sometimes is "time of the day", sometimes "hours", some others have Time UTC. My suggestion, to have it consistent with the axis on the y, is to write "Time [hours UTC]".

*The label of the time axis was replaced by "Time [h UTC]" in Fig. 2, 4, 5, 6, 7, 8, 9, 12a, S5, S7, S8 and S10. We chose to use "h" instead of "hours" since the Copernicus rules specify that standard abbreviation should be used for units.*

6) some plots (fig 4, fig 12 in particular) compare lines in pink with lines in red and lines in blue. I is hard, sometimes impossible to distinguish the red and the pink. Please change colors.

*The mangenta color was changed in dark green in Fig. 4 and 12.*

7) lines 319 - 324: do you really need these introductory lines? the section on wind is nicely structured and you could save some text, I think.

*These introductory lines are effectively not absolutely necessary. They were added to manuscript during the first iteration of the review process to facilitate the reading of the manuscript as recommended by the second referee. We think that it is still important to mention at this point the location of the REM and in-situ instruments as well as to mention the use of observations from other stations. The last sentence was also added to answer the comment of a referee about the necessity to compare the results in MEE and MER. In an effort of coherence with the modifications required by the three referees during the review process, only the sentence about KENDA-1 analysis is now removed.*

8) In section 3.2.3 you talk about N winds. Is it North winds? I would write it extensively. I did not get immediately what you meant.

*North wind is now written extensively through the whole manuscript.*

9) Please remove all sentences like "Figure X shows this is proportional to that", as it is in the sentence at line 419-420. You can rephrase all of them as "this is proportional to that (Fig x). In the example of the line 419-420 it can be written as: Clear warm days with low cloud coverage in July show a peculiar wind pattern along the Haslital (Figure 10a). Another case is in line 441-442.

*The sentences beginning with "Fig. xx " were suppressed at lines 241, 288, 357, 404 and 419. This kind of sentences allows anyhow to simplify the reading. They were consequently not modified when the merging of two sentences in one lead to long and complex sentences as at lines 396 and 441.*

10) Arriving to the end of the paper, it is clear that you mainly aim for climatological variability. I think it is then worth highlighting it more in the abstract. Also the title does not reflect this point particularly. You mainly show diurnal cycles and means, but in the abstract only the performance for monthly temperatures and wind medians made me think about climatology. Maybe it is worth mentioning some specific keywords there "climatology", "monthly/seasonal means" or similar.

*We add to remove the word "climatology" from the manuscript as judiciously requested by the second referee in the first review round. His remark was right and we will not re-introduce the notion of climatology for a 10-months campaign. The mention of a monthly analysis has been added to the abstract: "The findings of the present study mostly based  on monthly averages allow to better understand the temperature distributions, the thermally driven wind system in a medium size valley, the interactions with tributary valley flows, as well as the performances and limitations of KENDA-1 in such complex topography." Concerning the title, it seems us inappropriate to add the notion of "monthly average" first not to lengthen an already extensive title and second to acknowledge the fact that single profiles and hourly observations are also reported for case studies (see e.g. Fig. 10, 13, s6, s9, s11, s12 and s13)I.*